

**Title :**
**ORCHIDEE MICT-LEAK (r5459), a global model for the production, transport and**
**transformation of dissolved organic carbon from Arctic permafrost regions, Part**
**2: Model evaluation over the Lena River basin.**
**Authors:**
**Simon P.K. Bowring[1], Ronny Lauerwald[2], Bertrand Guenet[1], Dan Zhu[1], Matthieu**
**Guimberteau[1,3], Pierre Regnier[2], Ardalan Tootchi[3], Agnès Ducharne[3], Philippe**
**Ciais[1]**
**Affiliations:**
[1] Laboratoire des Sciences du Climat et de l'Environnement, LSCE, CEA, CNRS, UVSQ,
91191 Gif Sur Yvette, France
[2] Department of Geoscience, Environment & Society, Université Libre de Bruxelles,
Bruxelles, Belgium
[3] Sorbonne Université, CNRS, EPHE, Milieux environnementaux, transferts et
interaction dans les hydrosystèmes et les sols, Metis, 75005 Paris, France
**Abstract**
In this second part of a two-part study, we perform a simulation of the carbon and water
budget of the Lena catchment with the land surface model ORCHIDEE MICT-LEAK,
enabled to simulate dissolved organic carbon (DOC) production in soils and its transport
and fate in high latitudes inland waters. The model results are evaluated in their ability
to reproduce the fluxes of DOC and carbon dioxide ($CO_2$) along the soil-inland water
continuum, and the exchange of $CO_2$ with the atmosphere, including the evasion
outgassing of $CO_2$ from inland waters. We present simulation results over years 1901-
2007, and show that the model is able to broadly reproduce observed state variables
and their emergent properties across a range of interacting physical and biogeochemical
processes, including: 1) Net primary production (NPP), respiration and riverine
hydrologic amplitude, seasonality and inter-annual variation; 2) DOC concentrations,
bulk annual flow and their volumetric attribution at the sub-catchment level; 3) High
headwater versus downstream $CO_2$ evasion, an emergent phenomenon consistent with
observations over a spectrum of high latitude observational studies. (4) These quantities
obey emergent relationships with environmental variables like air temperature and
topographic slope that have been described in the literature. This gives us confidence in
reporting the following additional findings: (5) Of the ~34TgC yr$^{-1}$ left over as input to
terrestrial and aquatic systems after NPP is diminished by heterotrophic respiration, 7
TgC yr$^{-1}$ is leached and transported into the aquatic system. Of this, over half (3.6 TgC yr$^{-1}$
$^{-1}$) is evaded from the inland water surface back into the atmosphere and the remainder
(3.4 TgC yr$^{-1}$) flushed out into the Arctic Ocean, proportions in keeping with other,
empirically derived studies. (6) DOC exported from the floodplains is dominantly
sourced from recent, more 'labile' terrestrial production, in contrast to DOC leached
from the rest of the watershed with runoff and drainage, which is mostly sourced from
recalcitrant soil and litter. (7) All else equal, both historical climate change (a
spring/summer warming of 1.8°C over the catchment) and rising atmospheric $CO_2$
(+85.6ppm) are diagnosed from factorial simulations to contribute similar, significant
increases in DOC transport via primary production, although this similarity may not
hold in the future.



**1 Introduction**
A new branch of the high latitude-specific land surface component of the IPSL Earth
System model, ORCHIDEE MICT-LEAK (r5459), was enabled to simulate new model
processes of soil dissolved organic carbon (DOC) and $CO_2$ production, and their
advective/diffusive vertical transport within a discretized soil column as well as their
transport and transformation within the inland water network, in addition to improved
representation of hydrological and carbon processes in floodplains. These additions,
processes first coded in the model ORCHILEAK (Lauerwald et al., 2017) and
implemented within the high latitude base model ORCHIDEE-MICT v8.4.1 (Guimberteau
et al., 2018), were described in detail in Part 1 of this study.  This second part of our
study deals with the validation and application of our model. We validate simulation
outputs against observation for present-day and run transient simulations over the
historial period (1901-2007) using the Lena River basin as test case. The simulation
setup and rationale for choice of simulation basin are outlined below.
**2 Simulation Rationale**
The Lena river basin, which is bounded by the region 52-72°N; 102-142°E, was chosen
as the basin for model evaluation because it is the largest DOC discharge contribution
amongst the Arctic rivers, according to some estimates (Raymond et al., 2007; Holmes et
al., 2012), with its 2.5 million $km^2$ area (befitting our coarse-grid resolution) discharging
almost 20% of the summed discharge of the largest six Arctic rivers, its large areal
coverage by Podzols (DeLuca and Boisvenue, 2012), and the dominance of DOC versus
particulate organic carbon (POC) with 3-6Tg DOC-C $yr^{-1}$ vs. 0.03-0.04 Tg POC-C $yr^{-1}$
(Semiletov et al., 2011) in the total OC discharge load –factors all broadly representative
of the Eurasian Arctic rivers. Compared to other Eurasian  rivers, the Lena is relatively
well studied, which provides data across the range of soil, hydrologic, geochemical and
ecological domains over space and time, that enable us to perform adequate model
evaluation.
Climatological forcing is input from the Global Soil Wetness Project  Phase 3 (GSWP3)
v.0 data at a 1 degree 3-hourly resolution covering the period 1901 to 2007
(Supplement, Table S1), which is then interpolated to a 30 minute timestep to comply
with the timestep of the model's surface water and energy balance calculation period.
This dataset was chosen for its suitability as input for reproducing the amplitude and
seasonality of Northern Hemisphere high latitude riverine discharge in ORCHIDEE-
MICT, as compared to other datasets (Guimberteau et al., 2018). An improved
floodplains area input file for the Lena basin (Tootchi et al., 2019)  was used to drive the
simulation of floodplain dynamics (Supplement, Table S1).
Simulations were run over the Lena river basin  (Fig. 3a) at a 1 degree resolution (Fig. 1)
for the historical period between 1901 and 2007 to evaluate the simulated output of
relevant carbon fluxes and hydrologic variables against their observed values, as well as
those of emergent phenomena arising from their interplay (Fig. 1), at both the grid and
basin scale. We evaluate at the basin scale because the isolation of a single geographic
unit allows for a more refined alaysis of simulated variables than doing the same over
the global Pan-Arctic, much of which remains poorly accounted for in empirical
databases and literature.




**3 Simulation Setup**


As detailed in Part 1 (Section 3.1), the soil carbon stock used by our model was reconstituted from the soil carbon spinup of an ORCHIDEE-MICT run from Guimberteau et al. (2018) and run to quasi-steady state equilibrium for the Active and Slow carbon pools (Supplement, Fig. S1) under the new soil carbon scheme used in the model configuration of the present study (Fig. 1). After some adjustment runs to account for different data read/write norms between ORCHIDEE-MICT and this model version, the model was then run in transient mode under historical climate, land cover and atmospheric $CO_2$ concentrations. A summary of the step-wise procedure for simulation setup described above is detailed graphically in Fig. 1. The model was forced with and run over the climate, $CO_2$ and vegetation input forcing data for the period spanning 1901-2007 (Supplement, Table S1).

113

In order to derive an understanding of the environmental drivers of carbon cycling in the Lena watershed and analyze the model sensititivity to the corresponding forcing data, alternative simulations were run with constant climate and $CO_2$ conditions (Table 1, and Supplement Table S1). Thus a factorial simulation was devised, consisting of 2 factors and 3 simulations whose inputs were otherwise identical but for the investigated factor (Table 1).

120

121

**4 Results and Discussion**

123

We refer to different simulations performed in this study according to the sensitivity factors to which they are subjected. The 'Control' (CTRL) simulation is that for which transient climate and atmospheric $CO_2$ forcings are used. CLIM and $CO_2$ are those simulations for which climate variability and atmospheric $CO_2$ were held constant at their pre-industriela levels, respectively (Table 1). The following evaluation sections compare observations solely against the CTRL. The subsequent section will evaluate this comparison against the factorial simulations described above.

131

The overall carbon budgets and their fluxes as generated by each of the simulations are shown in Figs. 2 and 11 and discussed in detail at the end of the evaluation. Below, we examine that budget's component parts, in the following sequential order: In section 4.1 we briefly look through the overall carbon budget of the entire basin, discussing component fluxes of the budget, their values and what they mean. Section 4.2 evaluates DOC discharge, followed by DOC concentrations in export (4.3), dissolved $CO_2$ transport in rivers and its evasion from the river surface (4.4), emergent phenomena with respect to $CO_2$ evasion compared to river size (4.5.1) and DOC concentrations and slope (4.5.2), followed by DOC reactivity pools (4.6) and NPP and soil respiration (4.7). Wherever possible, model output are compared with available in situ observations, while emergent relationships between fluxes or concentrations and environmental controls found in observatons are also drawn from the model output, to provide a 'process oriented' evaluation of the model. In Section 4.8 we discuss the overall drivers of the fluxes simulated by our model with respect to the two CLIM and $CO_2$ factorial simulations and the implications of these for the future.

147

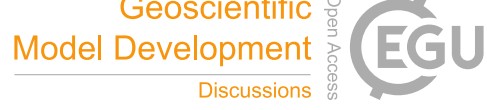



### 4.1  Carbon Budget: Simulated yearly fluxes

Fig. 2 summarises the components of the carbon cycle across the Lena basin, averaged over the decade 1998-2007. All units are in TgC yr$^{-1}$ and the errors are derived from average yearly standard deviations for each of these fluxes. Modelled carbon inputs to terrestrial ecosystems are dominated by photosynthetic input (GPP). GPP assimilates (875 TgC yr$^{-1}$) are either used as metabolic substrate by plants and lost as $CO_2$ by plant respiration processes (376 TgC yr$^{-1}$) or soil respiration processes (465 TgC yr$^{-1}$), leaving behind annual terrestrial carbon storage in living biomass and soil, known as net biome productivity (NBP, a sink of atmospheric $CO_2$ of 34 TgC yr$^{-1}$). Further carbon inputs are delivered to the terrestrial surface via a combination of atmospheric deposition, rainwater dissolved carbon, and the leaching of canopy carbon compounds, all of which summing up to a flux transported to the soil surface (4.6 TgC yr$^{-1}$) by throughfall (see Part 1, Section 2.5).

In the soil, DOC is produced by the decomposition of litter and soil organic carbon (SOC) pools (see Part 1, Section 2.4 and Fig. 2) and can be ad- or de- sorbed to solid particles (see Part 1, Section 2.11), while there is a continuous exchange  of  DOC with (solid) soil organic carbon. The interplay between decomposition and sorption leads to DOC concentration changes in the soil solution. DOC in the soil solution as well as a fraction of dissolved $CO_2$ produced in the root zone from root and microbial respiration is exported to rivers along the model's two hydrological export vectors, surface runoff and deep drainage (Part 1, Section 2.6). For the Lena basin simulations, these fluxes of C exported from soils amount to 5.1 and 0.2  TgC yr$^{-1}$, for DOC and $CO_2$ respectively.  Three water pools, representing streams, rivers and groundwater and each containing dissolved $CO_2$ and well as DOC of different reactivity, are routed through the landscape and between grid cells following the river network in the catchment (Part 1, Section 2.7). In addition, seasonally flooded soils located in low, flat grid cells next to the river network (see Part 1, Section 2.8) export DOC (0.57 TgC yr$^{-1}$) and $CO_2$ (1.54 TgC yr$^{-1}$) to the river network when their inundation occurs. Part of this leached inundated material is reinfiltrated back into the soil from the water column during floodplain recession ('Return' flux, 0.45 TgC yr$^{-1}$).  During its transport through inland waters, DOC can be decomposed into $CO_2$ (2.1 TgC yr$^{-1}$) and a fraction of river $CO_2$ produced from DOC and transferred from soil escapes to the atmosphere (3.6TgC yr$^{-1}$) through gas exhange kinetics (Part 1, Section 2.10). This flux is termed '$CO_2$ evasion' in Fig. 2 of this study. Carbon that 'survives' the inland water reactor is exported to the coastal ocean in the form of DOC (3.16 TgC yr$^{-1}$) and $CO_2$ (0.26 TgC yr$^{-1}$).  These fluxes and their interpretation within the context of the Land-Ocean-Aquatic Continuum (LOAC) are returned to in Section 4.8 of this study.

### 4.2 Discharge and DOC flux to the ocean

Simulated river water discharge captures the key feature of Arctic river discharge  – that of a massive increase in flow to  ~80,000 m$^3$s$^{-1}$ in April-June caused by melting snow and ice, otherwise known as ice-out or spring freshet, but underestimates observed river discharge in August to October by around 70% which is in the range of ~15,000-28,000 m$^3$s$^{-1}$ (Figs. 3c, 4b). Given that DOC fluxes are almost directly proportional to river discharge in the Lena basin (Fig. 3d), this sub-optimal performance with regard to hydrology during August to October seeming to be the main cause of a substantial underestimation in simulated bulk DOC outflow.  Another cause may simply be the lack



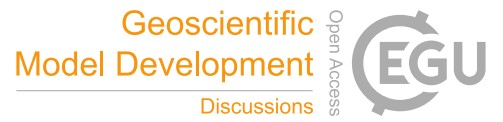

of peat representation in the model, for which DOC flux concentrations in outflowing
fluvial water can be very high (e.g. Frey et al., 2005; 2009: see Section 4.5.1).
In addition, the mean spring (June) discharge peak flows are slightly underestimated or
out of phase in simulations (Figs. 3c, 4b) compared to observations (Ye et al., 2009): this
is caused by a large amount of water throughput being simulated in May ($\sim$10,000 m$^3$ s$^{-1}$)
in excess of observed rates. Finally, during the winter low-flow period, it seems that
the model consistently under-estimates water flow-through volumes reaching the river
main stem (see Fig. 3c, winter months). Although this underestimate is not severe
relative to annual bulk flows, the divergence is large as a percentage of observations
(see right-hand axis, Fig. 3c), and may point to an issue in how ice is represented in the
model, such as the fact that solid ice inclusions in the soil column are not represented, or
the possibility that much slower groundwater dynamics than those represented in the
model are feeding discharge.
In addition to this, the presence of a dam on the Vilui tributary of the Lena has been
shown to reduce main stem winter low-flow rates by up to 90% (Ye et al., 2003), similar
to the discrepancy of our low-flow rates: given that our model only simulates 'natural'
hydrological flows and thus does not include dams, we expect that this effect is also at
play. Evaluating these considerations, if presently possible, remains beyond the scope of
this paper. We note that discharge simulations with ORCHIDEE MICT (Fig. 12 of
Guimberteau et al. (2018)) performed with the same climate forcing over the basin are
comparable with those from ORCHIDEE MICT-L, with similar overall seasonality and
discharge peaks of $\sim$60,000m$^3$ s$^{-1}$ in the former over the period 1981-2007. This
indicates that the modifications made in Bowring et al. (Part 1) focussing on the DOC
cycle did not degrade the hydrological performance of the model in this regard.
Our CTRL simulation shows that the yearly sum of DOC output to the Arctic Ocean has
increased steadily over course of the 20$^{th}$ Century, from $\sim$1.4Tg DOC-C yr$^{-1}$ in 1901 to
$\sim$4Tg DOC-C yr$^{-1}$ in 2007 (Fig. 4a). Smoothing the DOC discharge over a 30-year
running mean shows that the increasing trend (Fig. 4a) over this averaging scale is
almost linear, at $\sim$0.11TgC per decade, or a net increase of 40% using this averaging
scale. Empirically based estimates of total contemporary DOC entering the Laptev Sea
from Lena river discharge vary around $\sim$2.5-5.8 TgC-DOC (Cauwet and Sidorov, 1996;
Dolman et al., 2012; Holmes et al., 2012; Lara et al., 1998; Raymond et al., 2007;
Semiletov et al., 2011).
The red bar in Fig. 4a shows the average simulated DOC discharge of the last decade
(1998-2007) of 3.2 TgC yr$^{-1}$, to be compared with estimates of 3.6 TgC yr$^{-1}$ (black bar)
from Lara et al. (1998) and 5.8 TgC yr$^{-1}$ (orange bar) from Raymond et al. (2007) and 5.7
TgC yr$^{-1}$ from Holmes et al. (2012). These estimates are based on different years,
different data and different scaling approaches, whose veracity or accuracy are beyond
the scope of this study to address or assess.
Nonetheless, the most recent and elaborate of those estimates is that of Holmes et al.
(2012) who used a rating curve approach based on 17 samples collected from 2003 to
2006 and covering the full seasonal cycle, which was then applied to 10 years of daily
discharge data (1999-2008) for extrapolation. Given that their estimate is also based on
Arctic-GRO-1/PARTNERS data (https://www.arcticgreatrivers.org/data), which stands



as the highest temporal resolution dataset to date, we presume that their estimate can
be taken to be the most accurate of the actual riverine discharge of DOC from the Lena
basin.  Compared to their average annual estimate of 5.7 Tg C yr$^{-1}$ then, our simulated
DOC export is somewhat low, which can be due to multiple causes:
Firstly, as noted above, the model underestimates observed river discharge. We plot
seasonal DOC discharge against river discharge for the Lena outflow grid cell (Kusur
station –see Fig. 3a) over 1901-2007 in Fig. 3d, which shows a quasi-linear positive
relationship between the two. This dependence is particular to the Arctic rivers, in
which the DOC yield of rivers experiences disproportionately large increases in output
with increases in discharge yield (Fig. 4, Raymond et al., 2007), relative to the same
relationship in e.g. temperate rivers like the Mississippi (Fig. 3, Raymond et al., 2007),
owing largely to the 'flushing' out of terrestrially fixed carbon from the previous year's
production by the massive runoff generated by ice and snow melt during the spring
thaw.
Average river discharge almost doubled between the first and last decades of our
simulation (Fig 4c), giving further credence to the relationship between DOC and water
discharge. Comparing simulated annual mean discharge rate (m$^3$ s$^{-1}$) with long-term
observations (Ye et al. 2003) over years 1940-2000 (Fig. 4c) shows that though absolute
discharge rates are underestimated by simulations, their interannual variation
reasonably tracks the direction and magnitude of observations. Linear regressions
through each trend yield very similar yearly increases of 29 vs 38 m$^3$ s$^{-1}$ yr$^{-1}$ for
simulations and observations, respectively, while the mean annual water discharge
differential hovers at 30% (Fig. 4c), a fraction similar to that of the simulated and
observed (Raymond et al., 2007; Holmes et al., 2012) bulk annual DOC discharge
discrepancy (Fig. 4a).
Figure 4b plots discharge over the first (1901-1910) and last (1998-2007) decades of
simulated monthly DOC and river discharge with observed river discharge.  The bulk of
the DOC outflow occurs during the spring freshet or snow/ice-melting period of
increased discharge, accounting for ~50-70% of the year's total Lena outflow to the
Arctic (Lammers et al., 2001; Ye et al., 2009), with peak river discharge rates in June of
~80,000 m$^3$ s$^{-1}$. DOC concentrations increase immensely at this time, as meltwater
flushes out DOC accumulated from the previous year's litter and SOC generation
(Raymond et al., 2007; Kutscher et al., 2017).
This is reproduced in our simulations given that DOC discharge peak occurs at the onset
of the growing season, meaning that outflow DOC is generated from a temporally prior
stock of organic carbon. Simulation of the hydrological dynamic is presented in maps of
river discharge through the basin in Fig. 3b, which show low-flows in April with
substantial hydrographic flow from upstream mountainous headwaters and Lake Baikal
inflow in the south, peak flow in June with substantial headwater input in the northern
portion and a moderate flow through the mainstem with little headwater input in
September.
In Fig. 4b we observe that (i) DOC discharge fluxes closely track hydrological fluxes
(solid versus dashed lines); (ii) the simulated modern river discharge peak is very close
to the historical observed discharge peak, however it slightly overestimates spring





fluxes and substantially underestimates fluxes in the Autumn (dashed red versus black
lines).  Thus the discrepancy between simulated bulk DOC discharge fluxes and
empirical estimates may largely be found in the simulated hydrology. (iii) The curve
shape of discharge fluxes differs greatly between the first and last decades of simulation.
The difference between the first and last decades of the simulation in Fig. 4b is mostly
attributable to a large increase in the DOC flux mobilised by spring freshet waters that
culminate in the early summer outflow of DOC to the ocean, which generate the peaks in
DOC flux. This suggests both greater peaks in simulated DOC flux and a shift to earlier
peak timing, owing to an increase in river discharge indicative of an earlier spring and a
progressively warmer environment. (iv) The maximum modeled modern monthly DOC
flux rate of ~1.3 TgC month$^{-1}$ (Fig. 4b, solid red line) is comparable to the mean
maximum DOC flux rate measured in a recent study, which showed that the aggregate
carbon discharge flux of the Lena River over its 2-month peak period in 2013 was 3.5
TgC, giving a mean flux of 1.75 TgC month$^{-1}$ (Kutscher et al., 2017, Fig. 2).
The monthly pattern of DOC discharge approximates the seasonal pattern found in an
empirical Pan-Arctic DOC discharge study by Raymond et al. (2007), which they take to
represent total Lena river DOC discharge. The latter study, which looks at Pan-Arctic
DOC discharge rates, finding them to be 15-20% higher than in prior estimates, gives
discharge maxima in May, whereas our simulated maxima are in June.  We compare the
Raymond et al. (2007) modern DOC outflow (Fig. 4d, solid black line) from the Lena
river at Zhigansk (Raymond et al., 2007) against simulated DOC outflow from the
Zhigansk site as well as from the river outflow site (Kusur) 500km downstream (Fig. 4d,
solid blue and solid red lines, respectively).
Simulated DOC flux is underestimated for both sites. Peakflow at Zhigansk seems to be
attenuated over May and June in simulations, as opposed to May peakflow in
observations, while peakflow at Kusur is definitively in June. This suggests that
simulated outflow timing at Zhigansk may slightly delayed, causing a split in peak
discharge when averaged in the model output.  Thus the aggregation of model output to
monthly averages from calculated daily and 30 minute timesteps can result in the
artificial imposition of a normative temporal boundary (i.e. month) on a continuous
series. This may cause the less distinctive 'sharp' peak seen in Fig. 4d, ,which is instead
simulated at the downstream Kusur site, whose distance some 500km away from
Zhigansk more clearly explains the delay difference in seasonality.
We further evaluate our DOC discharge at the sub-basin scale, to see if the simulated
aggregate flux exiting the Lena river mouth is composed of a coarsely realistic
breakdown of source matter geography. In other words, whether the fractional
contribution of different DOC flows from rivers draining the simulated Lena basin
correspond to those in the observed basin.  This comparison is depicted in Fig. 5, where,
again using data from Kutscher et al., (2017), the observed and simulated percentage
DOC contributions of the Aldan, Vilui, and Upper and Lower Lena sub-basins to total flux
rates are 19 (24)%,  20(10%), 33 (38%) and 30 (28)% in simulations (observations) for
the four basins, respectively.



While deviations between simulation and observation can be expected given the
difference in magnitude and timing of DOC discharge previously discussed, in addition
to interannual variability, the nearly twofold value mismatch of the Vilui basin likely has
its roots in the fact in its real-word damming, not represented here. On the other hand,
we cannot explain the ~5% discrepancies in other sub-basin fluxes, particularly for the
Aldan.
Of the shortcomings in our model with respect to observations, year-on-year variations
over the decade 1998-2007 may be of significance, given that the Holmes et al. (2012)
and Raymond et al. (2007) DOC discharge values are significantly higher than total
organic carbon (DOC+POC) outflow estimates (~5.0-5.4 TgC yr$^{-1}$, Fig. 4a blue boundary)
as presented in Lara et al. (1998). To this we can add scale-related inaccuracies in the
routing protocol that can lead to small geographic inconsistencies in simulated versus
observed phenomena, as well as the exclusion of explicit peatland formation and related
dynamics in this model, which is the subject of further model developments within the
ORCHIDEE-MICT envelope (Qiu et al., 2018) that have yet to be included in this
iteration. With peatlands thought to cover ~17% of the Arctic land surface (Tarnocai et
al., 2009), and with substantially higher leaching concentrations, this may be a
significant omission from our model's representation of high latitude DOC dynamics.

**363 4.3 DOC Concentrations in lateral transport**

The range of simulated riverine DOC concentrations approximates those found in the
literature for the Lena and other Eurasian high-latitude river basins (e.g. Arctic-GRO 1
(https://www.arcticgreatrivers.org/data); Denfeld et al., 2013; Mann et al., 2015;
Raymond et al., 2007; Semiletov et al., 2011). In those for the Lena, observed average
DOC concentrations hover at ~10mgC L$^{-1}$. Likewise, simulated DOC concentrations
mostly lie in the range of 0-10 mgC L$^{-1}$, with monthly grid cell maxima of 1-200 mgC L$^{-1}$,
and on flow-weighted average exhibit the observed seasonal range and amplitude.
Figure 6 summarises some of this simulated output, showing maps of mean monthly
DOC concentration for stream water, river water and groundwater (Fig. 6a,b,c,
respectively) in April, June and September –the beginning, middle and end of the non-
frozen period in the basin, respectively, over 1998-2007.
For both the stream and river water reservoirs, DOC concentrations appear to have
spatio-temporal gradients correlated with the flux of water over the basin during the
thaw period, with high concentrations of 10-15 mgC L$^{-1}$ as the snow and ice melts in
April in the upstream portions of the basin, these high concentrations moving
northward to the coldest downstream regions of the basin in June. Lower DOC
concentrations of ~5 mgC L$^{-1}$ dominate the basin in September when the bulk of
simulated lateral flux of DOC has dissipated into the Laptev Sea, bearing in mind that we
underestimate the river discharge flux in the Autumn. In contrast, groundwater DOC
concentrations are generally stable with time, although some pixels appear to
experience some 'recharge' in their concentrations during the first two of the three
displayed thaw months. Significantly, highest groundwater DOC concentrations of up to
20 mgC L$^{-1}$ are focussed on the highest elevation areas of the Lena basin on its Eastern
boundary, which are characterized by a dominance of Podzols (SI, Fig. 2b).
This region, the Verkhoyansk range, is clearly visible as the high groundwater DOC

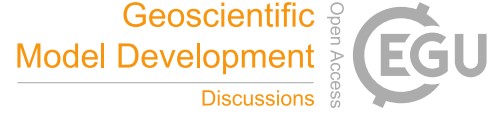



concentration(2-20mgC L$^{-1}$) arc (in red) in Fig. 6a, as well as other high elevation areas
in the south-western portion of the basin (see Fig. 3a for the basin grid cell mean
topographic slope), while the central basin of very low mean topographic slope exhibits
much smaller groundwater DOC concentrations (0-2mgC L$^{-1}$).  The range of simulated
groundwater DOC concentration comes close to those aggregated from the empirical
literature by Shvartsev (2008) in his seminal review of global groundwater
geochemistry, which finds from >9000 observations that groundwater in permafrost
regions exhibit a mean concentration of ~10 mgC L$^{-1}$ after peatlands and swamps (not
simulated here) are removed (Table 2).
The high groundwater reservoir DOC concentrations simulated in high altitude regions
by ORCHIDEE MICT-L is related to the fact that, in the model, DOC is rapidly produced
and infiltrated deep into soil above the permafrost table, to the point that it reaches the
simulated groundwater pool relatively quickly, allowing it to enter this reservoir before
being metabolised through the soil column –hence allowing for the relatively high
groundwater concentrations found in mountain areas. Because of the prevailing low
temperatures, this DOC is not quickly decomposed by microbes and instead feed the
groundwater DOC pool.
**4.4 In-Stream $CO_2$ Production, Transport, Evasion**
In our model, the fate of DOC once it enters the fluvial system is either to remain as DOC
and be exported to the ocean, or to be degraded to dissolved $CO_2$ ($CO_{2(aq.)}$), which is
itself either also transported to the marine system or outgassed from the fluvial surface
to the atmosphere (see Part 1, Section 2.10).  The latter two outcomes also apply to
$CO_{2(aq.)}$ produced in the soil by organic matter degradation and subsequently
transported by runoff and drainage flows to the water column.  As shown in Fig. 2, a
large proportion of DOC (38%, 2.1 TgC yr$^{-1}$) that enters the water column is degraded to
$CO_{2(aq.)}$ during transport, which adds to the 1.65 TgC yr$^{-1}$ of direct $CO_{2(aq.)}$ input from the
terrestrial land surface.  Of this bulk $CO_2$ exported into and generated within the water
column, 3.6 TgC yr$^{-1}$ evades from the water surface to the atmosphere before reaching
the river delta. In what follows, we evaluate first inputs of $CO_{2(aq.)}$ to the water column in
terms of their seasonality, before evaluating $CO_2$ evasion rates and the relation of this to
smaller and larger water bodies (river versus stream).
The seasonality of riverine dissolved $CO_2$ concentrations ($CO_{2(aq.)}$, mgC L$^{-1}$) is evaluated
in Fig. 4d to compare $CO_{2(aq.)}$ concentrations with DOC bulk flows, since $CO_{2(aq.)}$
concentrations follow an inverse seasonal pattern to those of DOC, being highest during
the winter baseflow period and lowest in summer due to dilution during its high
discharge phase (Semiletov et al., 2011).  The simulated flow of $CO_{2(aq.)}$ at Kusur (Fig. 4d,
dashed red) reproduces the seasonality of observations from Cauwet and Sidorov
(1996), who sampled the Lower Lena (ship-board, several sites in river delta region (see
Fig. 3a)), but somewhat underestimates concentrations, this perhaps due to the absence
of peat representation in our model, in combination with underestimated hydrological
discharge.  Also included in Fig. 4d is the basin average for all non-zero values, whose
shape also tracks that of observations.  Thus the model represents on the one hand
increasing hydrological flow mobilising increasing quantities and concentrations of DOC
while on the other hand those same increasing hydrological flows increasing the flux,
but decreasing the concentration, of $CO_{2(aq.)}$ throughput.



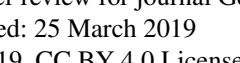


To our knowledge, no direct measurements for $CO_2$ evasion from the surface of the Lena
river are available in the literature, presumably owing to the notorious difficulty in
successfully obtaining such data. We refer to Denfeld et al. (2013) for evaluating our
evasion flux results, since their basin of study, the Kolyma River, is the most
geographically proximate existing dataset to the Lena, despite biogeographical
differences between the two basins –namely that the Kolyma is almost entirely
underlain by continuous permafrost. The Kolyma River $CO_2$ evasion study measured
evasion at 29 different sites along the river basin ($\sim$158-163°E; 68-69.5°N), with these
sites distinguished from one another as 'main stem', 'inflowing river' or 'stream' on the
basis of reach length.  The study showed that during the summer low-flow period
(August), areal river mainstem $CO_2$ evasion fluxes were $\sim$0.35 gC m$^{-2}$ d$^{-1}$, whereas for
streams of stream order 1-3 (widths 1-19m), evasion fluxes were up to $\sim$7 gC m$^{-2}$ d$^{-1}$,
and for non-mainstem rivers (widths 20-400m) mean net fluxes were roughly zero
(Table 3 of Denfeld et al., 2013).   Thus, while small streams have been observed to
contribute to roughly 2% of the Kolyma basin surface area, their measured percentage
contribution to total basin-wide $CO_2$ evasion $\sim$40%, whereas for the main stem the
surface area and evasion fractions were $\sim$80% and 60%, respectively.

Results such as these, in addition to permafrost soil incubation experiments (e.g. Drake
et al., 2015; Vonk et al., 2013, 2015b, 2015a) suggest that small streams, which
represent the initial (headwater) drainage sites of these basins, rapidly process
hydrologically leached carbon to the atmosphere, and that this high-reactivity carbon is
a mix of recently thawed ancient permafrost material, as well as decomposing matter
from the previous growth year.  This is given as evidence that the total carbon
processing of high-latitude rivers is significantly underestimated if only mainstem
carbon concentrations are used in the accounting framework, since a large amount of
carbon is metabolised to the atmosphere before reaching the site of measurement.

Figure 7 summarises some of the results from the simulated water body $CO_2$ outgassing
flux.  Year-on-year variation in basin-wide evasion from river, stream and floodplain
sources combined exhibits a marked increasing trend over the course of the 20$^{th}$
Century, increasing from a minimum of $\sim$1.6 TgCO$_2$-C yr$^{-1}$ in 1901 to a maximum of $\sim$4.4
TgCO$_2$-C yr$^{-1}$ in 2007, an increase of almost 300% (Fig. 7a). Smoothing the data over a 30
year running average yields a dampened net increase in basin-wide evasion of $\sim$30%
over the historical period on this averaging scale (Fig. 7a). Thus yearly evasion flux is
some 105% of yearly DOC discharge to the coast from the Lena basin and 51% of C
exported from soils to headwaters as $CO_2$ or DOC. If we compare the mean yearly rate of
increase in absolute (TgC yr$^{-1}$) $CO_2$ evasion and DOC discharge based on linear
regression over the whole simulation period, it appears that the rate of increase of both
fluxes has been strikingly similar over the simulated 20$^{th}$ Century, with mean increases
of 11.1 GgC yr$^{-1}$ and 11.5 GgC yr$^{-1}$ per year for evasion and export, respectively.

The heterogeneity of $CO_2$ evasion from different sources in the model is most evident in
terms of their geographic distribution and relative intensity, as shown in the evasion
flux rate maps (tons grid cell$^{-1}$ d$^{-1}$) over floodplain, stream and river areas in April, June
and September (Fig. 8a-c).  Whereas floodplains (Fig. 8a) tend to have some of the
highest evasion rates in the basin, their limited geographic extent means that their
contribution to basinwide evasion is limited for the whole Lena. Stream evasion



meanwhile (Fig.8b), tends to be broadly distributed over the whole basin, representing
the fact that small streams and their evasion are the main hydrologic connectors outside
of the main river and tributary grid cells, whereas river evasion (Fig. 8c) is clearly linked
to the hydrographic representation of the Lena main stem itself, with higher total
quantities in some individual grid cells than for the stream reservoir, yet distributed
amongst a substantially smaller number of grid cells. Whereas the stream reservoir has
greatest absolute evasion flux rates earlier in the year (April-May), maximum evasion
rates occur later in the year and further downstream for the river reservoir, reflecting
the fact that headwaters are first-order integrators of soil-water carbon connectivity,
whereas the river mainstem and tributaries are of a secondary order. Note that the
September values must be interpreted with caution, given the underestimation in our
simulations of the river discharge during the Autumn period.
The spatio-temporal pattern of increasing evasion over the simulation period is shown
in Fig. 7b as a Hovmöller difference plot, between the last and first decade, of log-scale
average monthly evasion rates per latitudinal band. This shows that the vast majority of
outgassing increase occurs between March and June, corresponding to the progressive
onset of the thaw period moving northwards over this timespan. Although relatively
small, outgassing increases are apparent for most of the year, particularly at lower
latitudes. This would suggest that the change is driven most acutely by relatively greater
temperature increases at higher latitudes ('Arctic amplification' of climate warming, e.g.
Bekryaev et al., 2010) while less acute but more temporally homogenous evasion is
driven by seasonal warming at lower latitudes.
As previously discussed, the proportion of total basin-wide $CO_2$ evasion attributable to
headwater streams and rivers is substantially greater than their proportion of total
basin surface area. Figure 7c represents the mean monthly fractional contribution of
each surface hydrological water pool to the total evasion flux (unitless) over the period
1998-2007. This shows that over the entirety of the thaw period, the stream water pool
takes over from the river water pool as the dominant evasion source, particularly at the
height of the freshet period, where its fractional contribution rises to >75%.
The stream fraction of August outgassing is roughly 57% of the annual total, which is
higher than the ~40% found for streams in the Denfeld et al. (2013) study. However,
the values between the two studies are not directly comparable, different basins
notwithstanding. This is because in ORCHIDEE MICT-L, the 'stream' water reservoir is
water routed to the river network for all hydrologic flows calculated to not cross a 0.5
degree grid cell boundary (the resolution of the routing module, explained in Part 1,
Section 2.6), which may not be commensurate with long, <20m wideth streams in the
real-world, that were used in the Denfeld et al. (2013) study. In addition, this 'stream'
water reservoir in the model does not include any values for width or area in the model,
so we cannot directly compare our stream reservoir to the <20m width criterion
employed by Denfeld et al. (2013) in their definition of an observed stream. Thus our
'stream' water reservoir encompasses substantially greater surface area and hydrologic
throughput than that in the Denfeld et al. study. We also add the qualification that
because of its coarse-scale routing scheme, ORCHIDEE isn't able to simulate stream
orders lower than 4 or 5 thus missing a potentially substantial vector for the water-
surface evasion of $CO_2$.



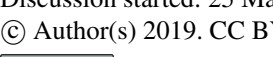

Significantly, also shown in Fig. 7c, is the gradual onset of evasion from the floodplain
reservoir in April, as the meltwater driven surge in river outflow leads to soil inundation
and the gradual increase of proportional evasion from these flooded areas over the
course of the summer, with peaks in June-August as water temperatures over these
flooded areas likewise peak. We stress the importance of these simulation results as
they concur with large numbers of observational studies (cited above) which show
smaller headwater streams' disproportionately large contribution to total outgassing
(Fig. 7c), this being due to their comparatively high outgassing rates (Fig. 7e). In
addition, the contribution of floodplains to evasion, an otherwise rarely studied feature
of high latitude biomes, is shown here to be significant.
A Hovmöller plot (Fig. 7d) of the monthly longitude-averaged stream reservoir fraction
of total evasion, gives some indication as to the spatio-temporal pattern under which
evasion from this hydrological pool evolves over the course of the year. From this we
can infer that: (i) The dominance of stream evasion begins in the most southern
upstream headwaters in the lower latitude thaw period (April-May), and trickles
northward over the course of the next two months, following the riverflow. (ii) The
intensity of stream water evasion is greatest in the lower latitude regions of the basin,
which we speculate is the result of higher temperatures causing a greater proliferation
of small thaw water-driven flows and evasion. (iii) Areas where the stream fraction is
not dominant or only briefly dominant during the summer (58-60°N, 63-64°N, 70-71°N)
are all areas where floodplain $CO_2$ evasion plays a prominent role at that latitudinal
band.
Although not directly comparable due to the previously mentioned issues arising from
our model-derived representation of 'stream' water versus those in the real world, we
evaluate the approximate rate of areal $CO_2$ efflux from the water surface against
obervations from Denfeld et al. (2013). The 'approximate' caveat refers to the fact that
model output doesn't define a precise surface area for the stream water reservoir, which
is instead bundled into a single value representing the riverine fraction of a grid cell's
total surface area. Thus, in order to break down the areal outgassing for the stream
versus river water reservoirs, we derive an approximate value for the fractional area
taken up by rivers and streams in a simple manner: we weight the total non-floodplain
inundated area of each grid cell by the relative total water mass of each of the two
hydrological pools, then divide the total daily $CO_2$ flux simulated by the model by this
value. The per-pool areal estimate is an approximation since it assumes that rivers and
streams have the same surface area: volume relationship. This is clearly not the case,
since streams are generally shallow, tending to have greater surface area per increment
increase in depth than rivers. Thus, our areal approximations are likely underestimated
(overestimated) for streams (rivers), respectively.
The comparison of simulated results with those from Denfeld et al. (2013) are displayed
in Fig. 7e, which shows boxplots for simulated $CO_2$ evasion (gC m$^{-2}$ d$^{-1}$) from the stream
water reservoir and river water reservoir averaged over 1998-2007. The empirical
(Kolyma river) analog of this data, from which this plot is inspired (Fig. 4d in Denfeld et
al., 2013), is shown inset in the figure, with whiskers in their case denoting measured
maxima and minima. Median efflux was 1.1 (6) versus 0.4 (0.8) for stream and river,
respectively, in simulations (observations). Like the observations, simulated stream
efflux had a substantially greater interquartile range, mean (24.6) and standard



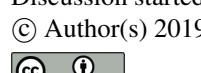

deviation (73) than total river efflux (1.3 and 7.2, respectively). Note that from ~700
non-zero simulation datapoints, 7 were omitted as 'outliers' from the stream reservoir
efflux statistics described below, because very low stream:river reservoir values skewed
the estimation of total approximate stream surface area values very low, leading to
extreme efflux rate values of 1-3000gC m$^{-2}$ d$^{-1}$ and are thus considered numerical
artefacts of the areal approximation approach used here.
**4.5 Emergent Phenomena**
**4.5.1 DOC and mean annual air temperature**
A key emergent property of DOC concentrations in soils and inland waters should be
their positive partial determination by the temperature of the environment under which
their rates of production occur, as has been shown in the literature on permafrost
regions, most notably in Frey & Smith (2005) and Frey & McClelland (2009).
Increasing temperatures should lead to greater primary production, thaw,
decomposition and microbial mobilisation rates, and hence DOC production rates,
leading to (dilution effects notwithstanding) higher concentrations of DOC in thaw and
so stream waters. Looking at this emergent property allows us to evaluate the soil-level
production of both DOC and thaw water at the appropriate biogeographic and temporal
scale in our model. This provides a further constraint on model effectiveness at
simulating existing phenomena at greater process-resolution.
Figure 9 compares three datasets (simulated and two observational) of riverine DOC
concentration (in mgC L$^{-1}$) plotted against mean annual air temperature (MAAT). The
simulated grid-scale DOC versus MAAT averaged over July and August (for
comparability of DOC with observational sampling period) of 1998-2007 is shown in
red, and observed data compiled by Laudon et al. (2012) and Frey and Smith (2005) for
sites in temperate/cold regions globally and peatland-dominated Western Siberia,
respectively. The Laudon et al. (2012) data are taken from 49 observations including
MAAT over the period 1997-2011 from catchments north of 43°N, and aggregated to 10
regional biogeographies, along with datapoints from their own sampling; those in the
Frey and Smith study are from 55-68°N and ~65-85°E (for site locations, see Laudon et
al. (2012), Table 1 and 2; Frey and Smith (2005), Fig. 1).
Fig. 9 can be interpreted in a number of ways. First, this MAAT continuum spans the
range of areas that are both highly and moderately permafrost affected and permafrost
free (Fig. 9, blue and green versus orange shading, respectively), potentially allowing us
a glimpse of the behaviour of DOC concentration as the environment transitions from
the former to the latter. Simulated Lena DOC concentrations, all in pixels with MAAT < -
2°C and hence all bearing continuous or discontinuous permafrost ('permafrost-
affected' in the figure), only exhibit a weakly positive response to MAAT on the scale
used ($y=6.05e^{0.03MAAT}$), although the consistent increase in DOC minima with MAAT is
clearly visible.
Second, the Laudon et al. (2012) data exhibit an increasing then decreasing trend over
the range of MAAT (-2°C to 10°C) in their dataset, which they propose reflects an
'optimal' MAAT range for the production and transport of DOC, occupying the 0°C to 3°C
range (Fig. 9, red shading). Below this optimum range, DOC concentrations may be



limited by transport due to freezing, and above this, smaller soil carbon pools and
temperature-driven decomposition would suppress the amount of DOC within rivers.
Third, the lower end of the Laudon et al. (2012) MAAT values correspond to a DOC
concentration roughly in line with DOC concentrations simulated by our model at those
temperatures.
Fourth, DOC concentrations in the Frey and Smith (2005) data exhibit a broad scattering
in permafrost-affected sites, with concentrations overlapping those of our simulations
(Fig. 9, green shading), before rapidly increasing to very high concentrations relative to
the Laudon et al. (2012) data, as sites transition to permafrost-free (red shading,
$y=3.6_{MAAT}+29.4$). Their data highlight the difference in DOC concentration regime
between areas of high (Frey and Smith, 2005) and low (Laudon et al., 2012) peatland
coverage and the different response of these to temperature changes.  Fifth, because our
simulation results largely correspond with the observed data where the MAAT ranges
overlap (green shading), and because  our model does not include peatland specific
processes, we should expect our model to broadly follow the polynomial regression
plotted for the Laudon et al. (2012) data as temperature inputs to the model increase.
Figure 9 implies that this increase should be on the order of a doubling of DOC
concentration as a system evolves from a MAAT of -2°C to 2°C.  In addition, as the Arctic
environment warms we should expect the response of DOC concentrations as a whole to
reflect a mix of both observationally-derived curves, as a function of peatland coverage.

### 4.5.2 DOC and topographic slope
Subsurface water infiltration fluxes and transformations of dissolved matter represent
an  important,  if  poorly  understood  and  observationally  under-represented
biogeochemical pathway of DOC export to river main stems, involving the complex
interplay of slope, parent material, temperature, permafrost material age and soil
physical-chemical processes, such as adsorption and priming.
In the Lena basin, as in other permafrost catchments, topographic slope has been shown
to be a powerful predictor for water infiltration depth, and concentration and age of
dissolved organic carbon (Jasechko et al., 2016; Kutscher et al., 2017; McGuire et al.,
2005), with deeper flow paths and older, lower DOC-concentrated waters found as the
topographic slope increases. This relationship was shown in Fig. 4 of Kutscher et al.
(2017) who surveyed DOC concentrations across a broad range of slope angle values in
the Lena basin and found a distinct negative relationship between the two.  We compare
the Kutscher et al. (2017) values with our model output, by plotting stream and river
DOC concentrations averaged per gridpoint over 1998-2007 against the topographic
map used in the routing scheme, versus their empirically derived data (Fig. 10).   As
shown therein, a similar negative relationship between the two variables is clearly
apparent.
A similar relationship was found in temperate rivers by Lauerwald et al. (2012), and a
recent paper by Connolly et al. (2018), based on their observational data and a synthesis
of Pan-Arctic empirical literature. They showed that for Arctic catchments in general, the
relationship of DOC concentration in fluvial waters scaled in a consistent and strongly
negative manner against topographic slope.  This was found for all Arctic catchments,
globally, prompting Connolly et al. to argue that topographic slope may be a type of





'master variable' for estimating fluvial DOC concentrations in the absence of viable *in*
*situ* measurement programs.

The reasoning for the negative slope-DOC concentration relationship is that as elevation
increases, temperature and primary production decreases. This leads to a thinner
organic soil layer, meaning that mineral soil plays a stronger role in shallow hydrologic
flowpaths, allowing for deeper infiltration and shorter residence time in a given soil
layer. In addition, steeper terrain leads to a lower soil water residence time and lower
moisture than in flat areas.

As a result, a given patch of soil matter will be exposed to leaching for less (residence)
time, while the organic matter that is leached is thought to be adsorbed more readily to
mineral soil particles, leading to either their re-stabilisation in the soil column or
shallow retention and subsequent heterotrophic respiration in situ, cumulatively
resulting in lower DOC concentrations in the hydrologic export (Kaiser and Kalbitz,
2012; Klaminder et al., 2011). This line of reasoning was recently shown to apply also to
deep organic permafrost soils (Zhang et al., 2017), although the degree to which this is
the case in comparison to mineral soils is as yet unknown.

In addition, and as described in Part 1 (Section 2.5) of this study, MICT-L contains a
provision for increased soil column infiltration and lower decomposition rates in areas
underlain by Podzols and Arenosols. The map from the Harmonized World Soil Database
(Nachtergaele, 2010), which is used as the input to this criterion, shows areas underlain
by these soils in the Lena basin to also be co-incident with areas of high topographic
slope (Fig. 3a, SI, Fig S2b). Their Podzol effect is to increase the rate of decomposition
and infiltration of DOC, relative to all other soil types, thus also increasing the rate of
DOC flux into groundwater (see Part 1 of this study, Section 2.5).

Our modelling framework explicitly resolves the processes involved in these
documented dynamics –soil thermodynamics, solid vertical flow (turbation), infiltration
as a function of soil textures and types, adsorption as a function of soil parameters (see
Part 1 of this study, Section 2.11), DOC respiration as a function of soil temperature and
hence depth (Part 1, Section 2.12), lagging of DOC vertical flow behind hydrological
drainage flow (summary Figure in Part 1, Fig. 1). We thus have some confidence in
reporting that the simulated negative relationship of DOC concentration with
topographic slope may indeed emerge from the model. If generalisable to permafrost
basins as a whole, this relationship may be an emergent process-based signal with
which to evaluate the biogeographic performance of permafrost-region DOC modelling
initiatives in the future, as was recently suggested by Connolly et al. (2018).

**4.6 DOC Reactivity Pools**

Here we examine the reactivity of DOC leached from the soil and litter to different
hydrological export pools. Surface runoff DOC export is dominated by refractory carbon
(Fig. 11a), with export rates largely following discharge rates as they drain the basin
with an increasing delay when latitude increases. As the thaw period gets underway
(April), the fraction of labile carbon in surface runoff DOC increases substantially from
south to north, reflecting the hydrologic uptake of the previous year's undecomposed
high-reactivity organic matter, as well as the addition of new inputs from the onset of

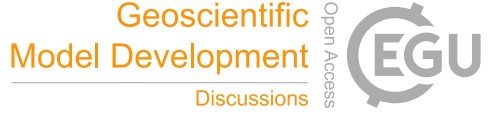

the current year's growing season.

Refractory carbon–dominated drainage DOC export rates (Fig. 11a) are centered on the
months June through October, with refractory export rate intensities per latitudinal
band during this period largely consistent with the fraction of inundated area (Fig. S1)
experienced by these bands during the course of the year, these centering on the areas
bounded by 52-65N and 70-72N.  The high refractory proportion of drainage flow is
expected, as drainage leaches older, relict soil and litter matter.  Because of its longer
water residence time, labile DOC carried vertically downward through the soil
infiltration flux will tend to be metabolised in situ before it can be exported to the
hydrological network, thus further increasing the proportion of refractory carbon.

By contrast floodplain DOC export (Fig. 11a) is dominated by the labile carbon pool but
is composed of more nuanced mix of both reactivity classes, reflecting its relatively
greater dependence on the current year's 'fresh' biomass as source material (62% labile
DOC versus 38% refractory DOC, year-averaged) for carbon leaching.  This can be
expected, since DOC and $CO_2$ production that would normally occur first in soil free DOC
concentrations before being gradually exported into surface runoff and drainage inputs
to the hydrological network are instead directly supplied to the water column as they
are generated, meaning that there is less of a time lag for the rapid decomposition of the
labile portion than through the other two hydrological export pathways.

For both the river and stream pool, mean DOC concentrations are also dominated by
refractory carbon sources. Interestingly, very high concentrations in the stream
reservoir are maintained year-round in the northernmost reaches of the Lena basin,  the
causes of which are not directly deducible from our data.  Likely, very high stream
concentrations are obtained from the confluence of relatively low volumetric water
fluxes in these regions that owe themselves to the freezing temperatures, with these low
temperatures likewise retarding direct heterotrophic respiration of contemporary plant
litter and favouring instead their environmental mobilisation by hydrological leaching,
when liquid water is available for matter dissolution.

When averaged over the year,  the dominance of the refractory DOC carbon pool over its
labile counterpart is also evident for all DOC inputs to the hydrological routing except
for floodplain inputs, as well as within the 'flowing' stream and river pools themselves.
This is shown in Table 2, where the year-averaged percentage of each carbon
component of the total input or reservoir is subdivided between the 'North' and 'South'
of the basin, these splits being arbitrarily imposed as the latitudinal mid-point of the
basin itself (63N).  This reinforces the generalised finding from our simulations that
refractory carbon dominates runoff and drainage inflows to rivers (89% refractory, on
average),  while floodplains export mostly labile DOC to the basin (64%), these values
being effectively independent of this latitudinal sub-division (Table 2).  This may be
expected, given that almost the entire basin is underlain by continuous permafrost,
whereas in areas with discontinuous or sporadic permafrost, the combination of higher
primary productivity and so litter input, with seasonal thaw of labile permafrost soil
matter may be expected to substantially increase the labile portion of the overall sum of
these quantities.  Nonetheless, there appears to be a small consistent difference between
North and South in the stream and river water DOC makeup, in that the labile portion
decreases between North and South ; this may be an attenuated reflection of the portion

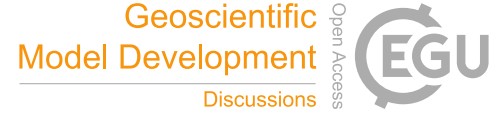



of labile DOC that is decomposed to $CO_2$ within the water column during its transport
northward, affecting the bulk average proportions contained within the water in each
'hemisphere'.
**4.7 NPP and Soil Respiration**
Rates of yearly net primary production (NPP) for Russian and Siberian forests have been
inferred in situ from eddy flux and inventory techniques to range from 123-250 gC m$^{-2}$
yr$^{-1}$ (Beer et al., 2006; Lloyd et al., 2002; Roser et al., 2002; Schulze et al., 1999;
Shvidenko and Nilsson, 2003).  We likewise simulate a broad range of NPP carbon
uptake rates, of 61-469 gC m$^{-2}$ yr$^{-1}$ averaged per grid cell over the Lena basin, with a
mean value of 210 gC m$^{-2}$ yr$^{-1}$.  NPP is heterogeneously distributed over space and
between PFTs (SI, Fig. S4c), with forests averaging 90 gC m$^{-2}$ yr$^{-1}$ and grasslands
averaging 104  gC m$^{-2}$ yr$^{-1}$ over the basin as a whole.  Low values tended to originate in
basin grid cells with elevated topography or high mean slope, while the maximum value
was standalone, exceeding the next greatest by ~100 gC m$^{-2}$ yr$^{-1}$, and is most likely
caused by the edge effects of upscaling a coastal gridcell's small fraction of terrestrial
area where high productivity occurs in a small plot, to the grid cell as a whole.  By
evaluating NPP we are also evaluating at a secondary level litter production, which is at
a third level a major component of DOC production.
Taken as a whole, gross primary production (GPP) was performed under simulations by
four PFT groups, with the largest basin-wide bulk contributions coming from boreal
needleleaf summer-green trees and C3 grasses (SI, Fig. S4a), the highest GPP uptake
rates (3 TgC pixel$^{-1}$ yr$^{-1}$) generated by boreal needleleaf evergreen trees, and the
remainder of GPP contributed by Boreal broad-leaved summer-green trees (SI, Fig. S4a).
Soil respiration rates, of combined soil heterotroph and plant root respiration in our
Control simulation, averaged 208 gC m$^{-2}$ yr$^{-1}$ (0.57 gC m$^{-2}$ d$^{-1}$) over the Lena basin over
the period 1990-2000, which is somewhat higher than those found by Elberling (2007)
in forest soils over Svalbard, of 103-176 gC m$^{-2}$ yr$^{-1}$ (0.28-0.48 gC m$^{-2}$ d$^{-1}$). Sawamoto, et
al. (2000) measured in situ summertime soil respiration over the central Lena basin and
found rates of 1.6-34 gC m$^{-2}$ d$^{-1}$,  while Sommerkorn (2008) observed rates of 0.1-3.9 gC
m$^{-2}$ d$^{-1}$ at higher latitudes, these appearing to vary with vegetation and fire history,
water table depth and temperature.  Mean heterotrophic respiration rates of 1.6 gC m$^{-2}$
d$^{-1}$ are simulated here during July and August, in the range 0.0.5-2.2 gC m$^{-2}$ d$^{-1}$ for each of
the above PFT groups.  The spatial distribution of, and difference in respiration rates
between PFT groups largely mirrors  those for NPP (SI Fig. S4c), with maximum rates of
1.4 gC m$^2$ d$^{-1}$ over forested sites, versus a maximum of 2.2 gC m$^2$ d$^{-1}$ over
grassland/tundra sites (SI, Fig. S4b).
Aggregated over the basin, results show that increases over the course of the 20[th]
Century were simulated for NPP, GPP, River Discharge, DOC, $CO_{2(aq.)}$, autotrophic and
heterotrophic respiration and $CO_2$ evasion, with percentage changes in the last versus
first decade of +25%, +27%, 38%, +73%, +60%, +30%, +33% and +63%, respectively.
(Fig. 11b).   It thus appears that rising temperatures and $CO_2$ concentrations
disproportionately favoured the metabolisation of carbon within the soil and its
transport and mineralisation within the water column, fed by higher rates of primary
production and litter formation as well as an accelerated hydrological cycle (see Fig. 4b

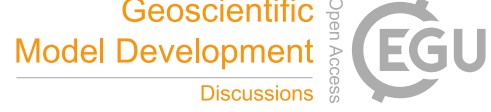



and 13a).

In Figure 11c we run linear regressions through scatter-plots of yearly DOC and $CO_2$
export and $CO_2$ evasion fluxes, versus rates of NPP (TgC yr$^{-1}$). These show that whereas
bulk DOC flux appears most sensitive (steeper slope) to increases in NPP, it is also least
coupled to it (more scattered, $R^2$=0.42). $CO_2$ evasion is least sensitive yet most tightly
coupled to NPP ($R^2$=0.52), while $CO_2$ export is intermediate between the two for both
($R^2$=0.43) –this is expected given that $CO_2$ export is also the intermediate state between
DOC export and $CO_2$ evasion. The greater scattering of DOC:NPP compared to
evasion:NPP is understandable, given that the initial of leaching is a covariate of both
primary production and runoff, whereas the actual evasion flux is largely dependent on
organic inputs (production) and temperature.

**4.8 Land-Ocean Aquatic Continuum (LOAC)**

**4.8.1 LOAC Fluxes**

Overall, our simulation results show that dissolved carbon entering the Lena river
system is significantly transformed during its transport to the ocean. Taking the average
throughput of carbon into the system over the last ten years of our simulation, our
results show that whereas 7 TgC yr$^{-1}$ (after reinfiltration following flooding of 0.45 TgC
yr$^{-1}$; see Fig. 2 'Return' flux) of carbon enters the Lena from terrestrial sources as
dissolved carbon and $CO_2$, only 3.4 TgC yr$^{-1}$  is discharged into the Laptev Sea and
beyond from the river mouth. The remainder (3.6TgC yr$^{-1}$) is metabolised in the water
column during transport and evaded to the atmosphere (bottom panel, Fig. 12a).  The
terrestrial DOC inflow estimate is comparable to that made by Kicklighter et al. (2013),
who estimated in a modelling study terrestrial dissolved carbon loading of the Lena is
~7.7 TgC yr$^{-1}$.

The relative quantities of carbon inflow, evasion and outflow in the river system that are
presented for the Lena in Fig. 12a can be compared to the same relative quantities –that
is, the ratios of evasion:in and out:in, where 'in' refers to dissolved terrestrial input, –
from the global study by Cole et al. (2007), who estimated these fluxes from empirical or
empirically-derived data at the global scale.   This is shown in the top panel of Fig. 12a,
where we simplify the Cole et al. (2007) data to exclude global groundwater $CO_2$ flux
from the coast to the ocean  (because our basin mask has a single coastal pixel whereas
coastal groundwater seepage is distributed along the entire continental boundary) and
the POC fraction of in-river transport and sedimentation (since ORCHIDEE MICT lacks a
POC erosion/sedimentation module) from their budget.

This gives global terrestrial dissolved carbon input of 1.45 PgC yr$^{-1}$, 0.7 PgC of which is
discharged to the ocean, and the other 0.75 PgC evaded to the atmosphere.  Taking the
previously mentioned [evasion:in] and [out:in] ratios as a percentage, the outflow and
evasion fluxes for the Lena versus the global aggregate are remarkably similar, at 48.6
vs. 48.3% and 51.4 vs 51.7%, for the two respective flows. Thus our results agree with
the proposition that the riverine portion of the 'land-ocean aquatic continuum' (Regnier
et al., 2013) or 'boundless carbon cycle' (Battin et al., 2009) is indeed a substantial
reactor for matter transported along it.



### 4.8.2 LOAC drivers

The constant climate (CLIM) and constant $CO_2$ (CO2) simulations were undertaken to assess the extent –and the extent of the difference –to which these two factors are drivers of model processes and fluxes. These differences are summarised in Figs. 12(b-c), in which we show the same 1998-2007 –averaged yearly variable fluxes as in the CTRL simulation, expressed as percentages of the CTRL values given in Fig. 2. A number of conclusions can be drawn from these diagrams.

First, all fluxes are lower in the factorial simulations, which can be expected due to lower carbon input to vegetation from the atmosphere (constant $CO_2$) and colder temperatures (constant climate) inhibiting more vigorous growth and carbon cycling. Second, broadly speaking, both climate and $CO_2$ appear to have similar effects on all fluxes, at least within the range of climatic and $CO_2$ values to which they have subjected the model in these historical runs. With regard to lateral export fluxes in isolation, variable climate (temperature increase) is a more powerful driver than $CO_2$ increase (see below). Third, the greatest difference between the constant climate and $CO_2$ simulation carbon fluxes appear to be those associated with terrestrial inflow of dissolved matter to the aquatic network, these being more sensitive to climatic than $CO_2$ variability. This is evidenced by a 49% and 32% decline in $CO_2$ and DOC export, respectively, from the land to rivers in the constant climate simulation, versus a 27% and 23% decline in these same variables in the constant $CO_2$ simulation. Given that the decline in primary production and respiration in both factorial simulations was roughly the same, this difference in terrestrial dissolved input is attributable to the effect of climate (increased temperatures) on the hydrological cycle, driving changes in lateral export fluxes.

This would imply that at these carbon dioxide and climatic ranges, the modelled DOC inputs are slightly more sensitive to changes in the climate rather than to changes in atmospheric carbon dioxide concentration and the first order biospheric response to this. However, while the model biospheric response to carbon dioxide concentration may be linear, thresholds in environmental variables such as MAAT may prove to be tipping points in the system's emergent response to change, as implied by Fig. 9, meaning that the Lena, as with the Arctic in general, may soon become much more temperature-dominated, with regard to the drivers of its own change.

### 4.8.3 LOAC export flux considerations

Despite our simulations' agreement with observations regarding the proportional fate of terrestrial DOC inputs as evasion and marine export (Section 4.8.1, Fig. 12a), our results suggest substantial and meaningful differences in the magnitude of those fluxes relative to NPP in the Lena, compared to those estimated by other studies in temperate or tropical biomes. Our simulations' cumulative DOC and $CO_2$ export from the terrestrial realm into inland waters is equivalent to ~1.5 % of NPP.

This is considerably lower than Cole et al. (2007) and Regnier et al. (2013) who find lateral transfer to approximate ~5% (1.9PgC yr$^{-1}$) of NPP at the global scale, while Lauerwald et al. (2017) found similar rates for the Amazon. The cause of this discrepancy with our results is beyond the scope of this study to definitively address,



given the lack of tracers for carbon source and age in our model. Nonetheless, our analysis leads us to hypothesise the following.

Temperature limitation of soil microbial respiration at the end of the growing season (approaching zero by October, SI Fig. S4d) makes this flux neglible from November through May (SI Fig. S4d). In late spring, mobilisation of organic carbon is performed by both microbial respiration and leaching of DOC via runoff and drainage water fluxes. However, because the latter are controlled by the initial spring meltwater flux period, which occurs before the growing season has had time to produce litter or new soil carbon (May-June, Fig. 4b), aggregate yearly DOC transport reactivity is characterised by the available plant matter from the previous year, which is overwhelmingly derived from recalcitrant soil matter (Fig. 11a) and is itself less available for leaching based on soil carbon residence times.

This causes relatively low leaching rates and riverine DOC concentration s(e.g. Fig. 9), as compared to the case of leaching from the same year's biological production. Highlighting this point are floodplains' domination by labile carbon sourced from that year's production with a mean DOC concentration of 12.4 mgC $L^{-1}$ (1998-2007 average), with mean riverine DOC concentrations around half that value (6.9 mgC $L^{-1}$). Nonetheless the May-June meltwater pulse period dominates aggregate DOC discharge. As this pulse rapidly subsides by late July, so does the leaching and transport of organic matter. Warmer temperatures come in conjunction with increased primary production and the temperature driven soil heterotrophic degradation of contemporary and older matter (via active layer deepening). These all indicate that transported dissolved matter in rivers, at least at peak outflow, is dominated by sources originating in the previous year's primary production, that was literally 'frozen out' of more complete decomposition by soil heterotrophs.

Further, we infer from the fact that all of our simulation grid cells fall within areas of low (<-2°C) MAAT, far below the threshold MAAT (>3°C) proposed by Laudon et al. (2012) for soil respiration-dominated carbon cycling systems (Fig. 9), that the Lena is hydrologically-limited with respect to DOC concentration and its lateral flux. Indeed, the seasonal discharge trend of the Lena –massive snowmelt-driven hydrological and absolute DOC flux, coupled with relatively low DOC concentrations at the river mouth (Fig. 4b, simulation data of Fig. 9), are in line with the Laudon et al. (2012) typology.

We therefore suggest that relatively low lateral transport relative to primary production rates (e.g. as a percentage of net primary production, (%NPP)) in our simulations compared to the lateral transport : NPP percentages reported from the literature in other biomes is driven by meltwater (vs. precipitation) dominated DOC mobilisation, which occurs during a largely pre-litter deposition period of the growing season. DOC is then less readily mobilised by being sourced from recalcitrant matter, leading to low leaching concentrations relative to those from labile material. As discharge rates decline, the growing season reaches its peak, leaving carbon mobilisation of fresh organic matter to be overwhelmingly driven by in situ heterotrophic respiration.

While we have shown that bulk DOC fluxes scale linearly to bulk discharge flows (Fig. 3d), DOC concentrations (mgC $L^{-1}$) hold a more complex and weaker positive relationship with discharge rates, with correlation coefficients ($R^2$) of 0.05 and 0.25 for



river and stream DOC concentrations, respectively (Fig. 13). This implies that while
increasing discharge reflects increasing runoff and an increasing vector for DOC
leaching, particularly in smaller tributary streams, by the time this higher input of
carbon reaches the river main stem there is a confounding effect of dilution by increased
water fluxes which reduces DOC concentrations, explaining the difference between
stream and river discharge vs. DOC concentration regressions in the Figure. Thus, and
as a broad generalisation, with increasing discharge rates we can also expect somewhat
higher concentrations of terrestrial DOC input to streams and rivers. Over the
floodplains, DOC concentrations hold no linear relationship with discharge rates
($R^2=0.003$ , SI Fig. S5), largely reflecting the fact that DOC leaching is here limited by
terrestrial primary production rates more than by hydrology. To the extent that
floodplains fundamentally require flooding and hence do depend on floodwater inputs
at a primary level, we hypothesise that DOC leaching rates are not limited by that water
input, at least over the simulated Lena basin.
As discussed above simulated DOC and $CO_2$ export as a percentage of simulated NPP
over the Lena basin was 1.5% over 1998-2007. However, this proportion appears to be
highly dynamic at the decadal timescale. As shown in Fig. 11b, all lateral flux
components in our simulations increased their relative throughput at a rate double to
triple that of NPP or respiration fluxes over the 20th century, also doing so at a rate
substantially higher than the rate increase in discharge. In addition, differentials of
these lateral flux rates with the rates of their drivers (discharge, primary production)
have on average increased over the century (Fig. 11b). This suggests that there are
potential additive effects of the production and discharge drivers of lateral fluxes that
could lead to non-linear reponses to changes in these drivers as the Arctic environment
transforms, as suggested by the Laudon et al. (2012) data plotted in Fig. 4. Acceleration
of the hydrological cycle compounded by temperature and $CO_2$ -driven increases in
primary production could therefore increase the amount of matter available for
leaching, increase the carbon concentration of leachate, and increase the aggregate
generation of runoff to be used as a DOC transport vector. Given that these causal
dynamics apply generally to permafrost regions, both low lateral flux as %NPP and the
hypothesised response of those fluxes to future warming may be a feature particular to
most high latitude river basins.

**5. Conclusion**

This study has shown that the new DOC-representing high latitude model version of
ORCHIDEE, ORCHIDEE MICT-LEAK, is able to reproduce with reasonable accuracy
modern concentrations, rates and absolute fluxes of carbon in dissolved form, as well as
the relative seasonality of these quantities through the year. When combined with a
reasonable reproduction of real-world stream, river and floodplain dynamics, we
demonstrate that this model is a potentially powerful new tool for diagnosing and
reproducing past, present and potentially future states of the Arctic carbon cycle. Our
simulations show that of the 34 TgC yr$^{-1}$ remaining after GPP is respired autotrophically
and heterotrophically in the Lena basin, over one-fifth of this captured carbon is
removed into the aquatic system. Of this, over half is released to the atmosphere from
the river surface during its period of transport to the ocean, in agreement with previous
empirically-derived global-scale studies. Both this transport and its transformation are
therefore non-trivial components of the carbon system at these latitudes that we have

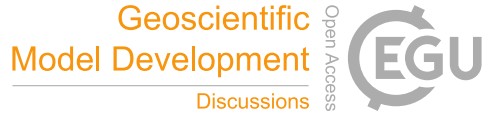

shown are sensitive to changes in temperature, precipitation and atmospheric $CO_2$
concentration.  Our results, in combination with empirical data, further suggest that
changes to these drivers –in particular climate –may provoke non-linear responses in
the transport and transformation of carbon across the terrestrial-aquatic system's
interface as change progresses in an Arctic environment increasingly characterised by
amplified warming.
**Code and data availability**
The source code for ORCHIDEE MICT-LEAK revision 5459 is available via
http://forge.ipsl.jussieu.fr/orchidee/wiki/GroupActivities/CodeAvalaibilityPublication/
ORCHIDEE_gmd-2018-MICT-LEAK_r5459
Primary data and scripts used in the analysis and other supplementary information that
may be useful in reproducing the author's work can be obtained by contacting the
corresponding author.
This software is governed by the CeCILL license under French law and abiding by the
rules of distribution of free software. You can use, modify and/or redistribute the
software under the terms of the CeCILL license as circulated by CEA, CNRS and INRIA at
the following URL: http://www.cecill.info.
**Authors' contribution**
SB coded this model version, conducted the simulations and wrote the main body of the
paper. RL gave consistent input to the coding process and made numerous code
improvements and bug fixes.  BG advised on the inclusion of priming processes in the
model and advised on the study design and model configuration; DZ gave input on the
modelled soil carbon processes and model configuration. PR contributed to the
interpretation of results and made substantial contributions to the manuscript text.  MG,
AT and AD contributed to improvements in hydrological representation and floodplain
forcing data. PC oversaw all developments leading to the publication of this study. All
authors contributed to suggestions regarding the final content of the study.
**Competing interests**
The authors declare no competing financial interests.

**Acknowledgements**
Simon Bowring acknowledges funding from the European Union's Horizon 2020
research and innovation program under the Marie Sklodowska-Curie grant agreement
No. 643052, 'C-CASCADES' program.  Simon Bowring received a PhD grant. Matthieu
Guimberteau acknowledges funding from the European Research Council Synergy grant
ERC-2013-SyG-610028 IMBALANCE-P. RL acknowledges funding from the European
Union's Horizon 2020 research and innovation program under grant agreement
no.703813 for the Marie Sklodowska-Curie European Individual Fellowship "C-Leak".

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

**Tables and Figures:**
**Table 1:** Summary describing of the factorial simulations undertaken to examine the
relative drivers of lateral fluxes in our model.

| Simulation Name | Abbreviation | Historical Input Data | Input* Held Constant |
|---|---|---|---|
| Control | CTRL | Climate, CO2, Vegetation | None |
| Constant Climate | CLIM | CO2, Vegetation | Climate |
| Constant CO2 | CO2 | Climate, Vegetation | CO2 (Pre-industrial) |

*Historically-variable input

**Table 2:** Mean observed groundwater $CO_2$ and DOC concentrations for global
permafrost regions subdivided by biogeographic province and compiled by Shvartsev
(2008) from over 9000 observations.

| | Permafrost Groundwater Provinces | | | | |
|---|---|---|---|---|---|
| | Swamp | Tundra | Taiga | Average | Average (-Swamp) |
| $CO_2$ (mgC $L^{-1}$) | 12.3 | 14 | 10.8 | 12.4 | 12.4 |
| DOC (mgC $L^{-1}$) | 17.6 | 10.1 | 9.3 | 12.3 | 9.7 |

**Table 3**: Summary of the average carbon reactivity types comprising the hydrological
inputs to rivers and streams (runoff, drainage and floodplain inputs), and within the
rivers and streams themselves, subdivided between the 'North' and 'South' of the Lena
basin (greater or less than 63N, respectively).

| Hydrological Source | Model Carbon Reactivity Pool | North | South |
|---|---|---|---|
| **Runoff Input** | Refractory | 81% | 83% |
| | Labile | 19% | 17% |



| Drainage Input | Refractory | 96% | 94% |
|---|---|---|---|
| | Labile | 4% | 6% |
| Flood Input | Refractory | 36% | 37% |
| | Labile | 64% | 63% |
| Streams | Refractory | 91% | 89% |
| | Labile | 9% | 11% |
| Rivers | Refractory | 92% | 90% |
| | Labile | 8% | 10% |


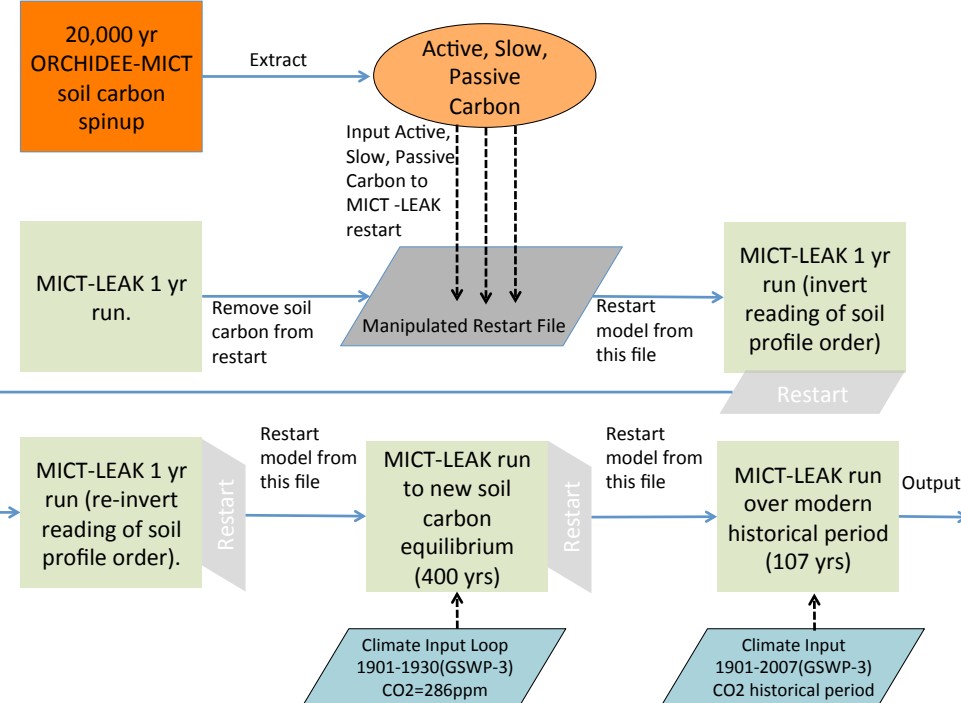

**Figure 1:** Flow diagram illustrating the step-wise stages required to set up the model,
up to and including the historical period. The two stages that refer to the inverted
reading of restart soil profile order point to the fact that the restart inputs from
ORCHIDEE-MICT are read by our model in inverse order, so that one year must be run in
which an activated flag reads it properly, before the reading of soil profile restarts is re-
inverted for all subsequent years.





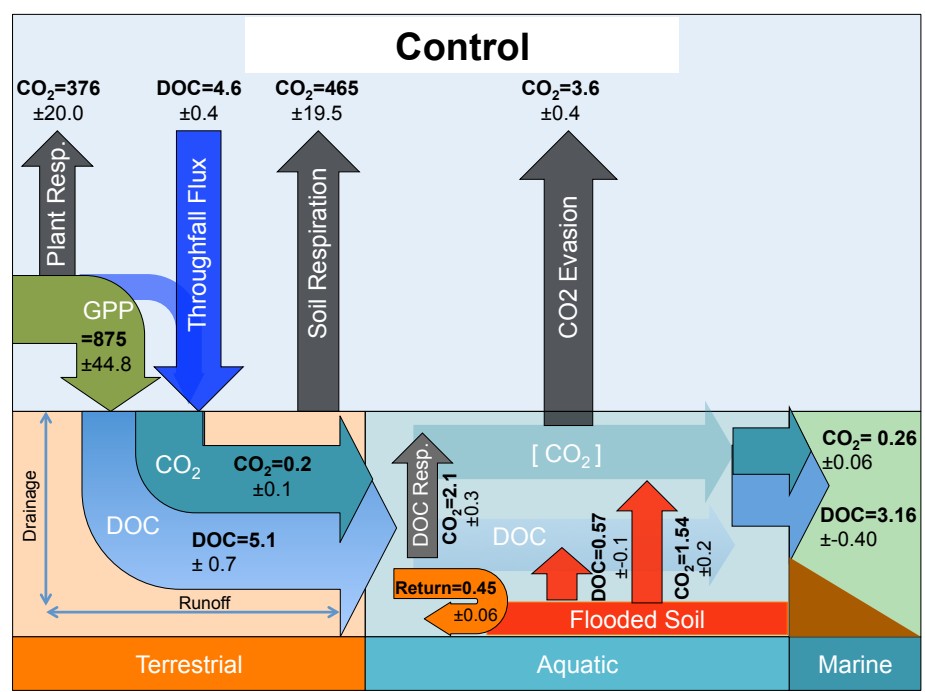

**Figure 2:** Schematic diagrams detailing the major yearly carbon flux outputs (TgC yr[-1]) from the Control simulation averaged over the period 1998-2007 as they are transformed and transported across the land-aquatic continuum.





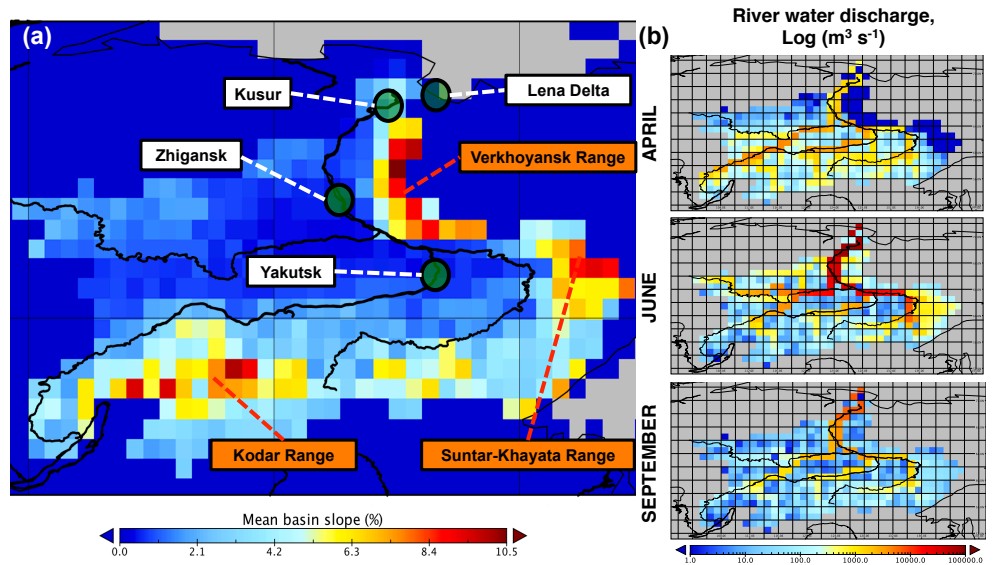






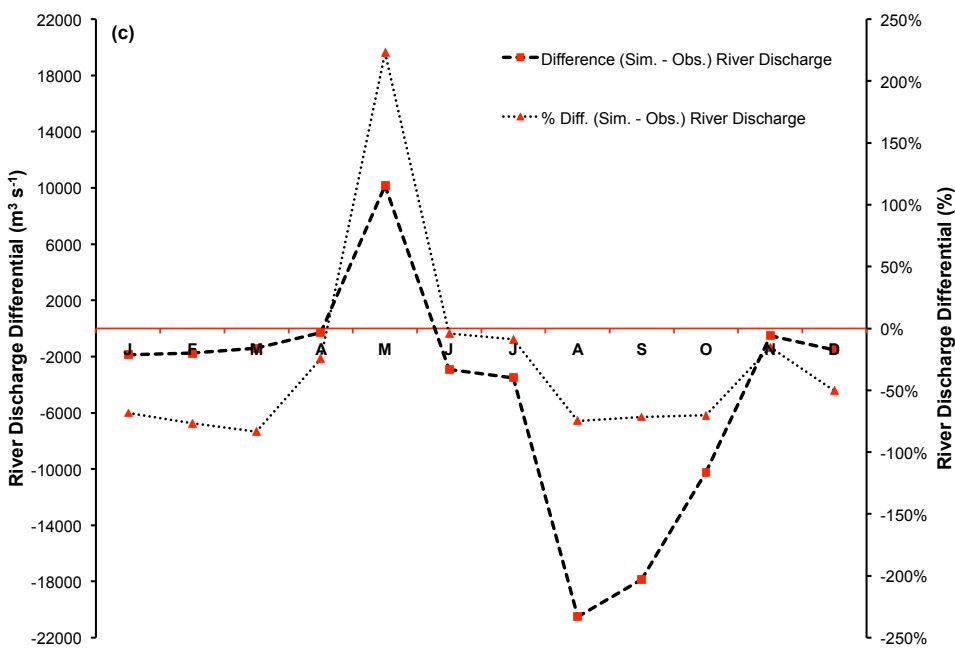


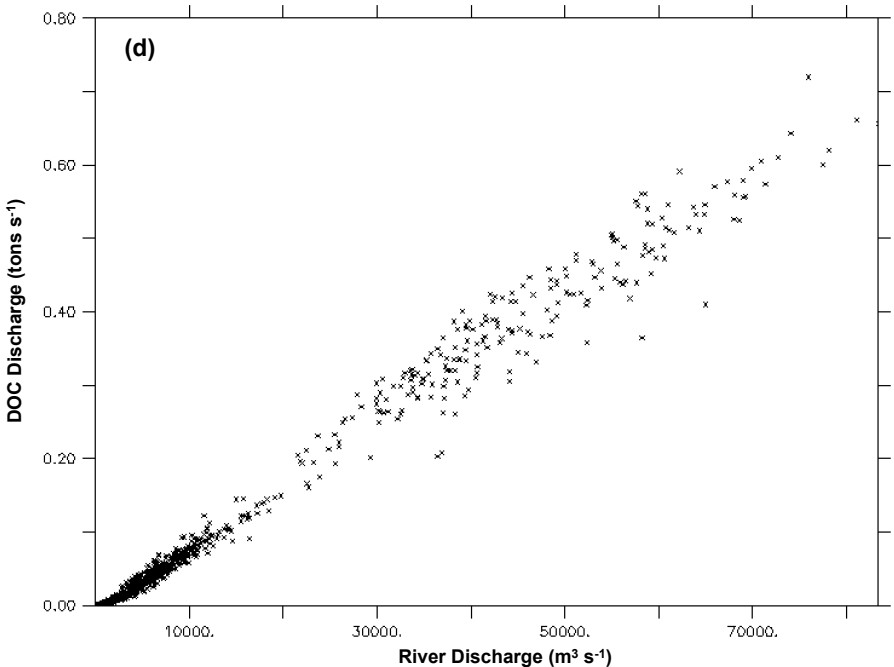






**Figure 3**: Map of the Lena **(a)** with the scale bar showing the mean grid cell topographic
slope from the simulation, and the black line the satellite-derived overlay of the river
main stem and sub-basins. Mountain ranges of the Lena basin are shown in orange.
Green circles denote the outflow gridcell (Kusur) from which our simulation outflow
data are derived, as well as the Zhigansk site, from which out evaluation against data
from Raymond et al. (2007) are assessed. The regional capital (Yakutsk) is also included
for geographic reference. **(b)** Maps of river water discharge (log(m$^3$ s$^{-1}$)) in April, June
and September, averaged over 1998-2007.   **(c)** The mean monthly river discharge
differential between observed discharge for the Lena (Ye et al., 2009) and simulated
discharge averaged over 1998-2007, in absolute (m$^3$ s$^{-1}$) and percentage terms.   **(d)**
Regression of simulated monthly DOC discharge versus simulated river discharge at the
river mouth (Kusur) over the entire simulation period (1901-2007).
**(a)**

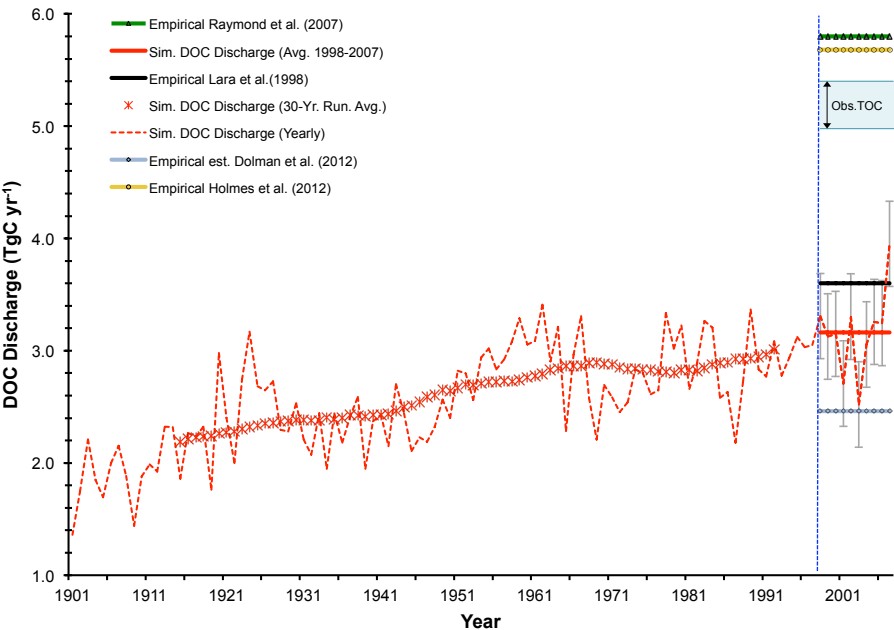

**(b)**



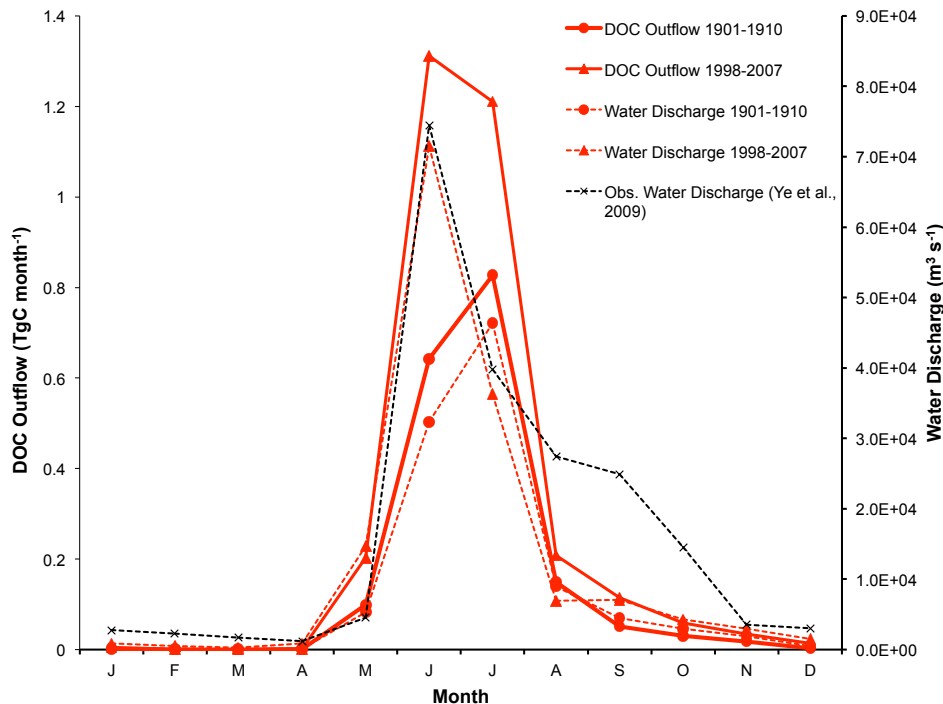

**(c)**

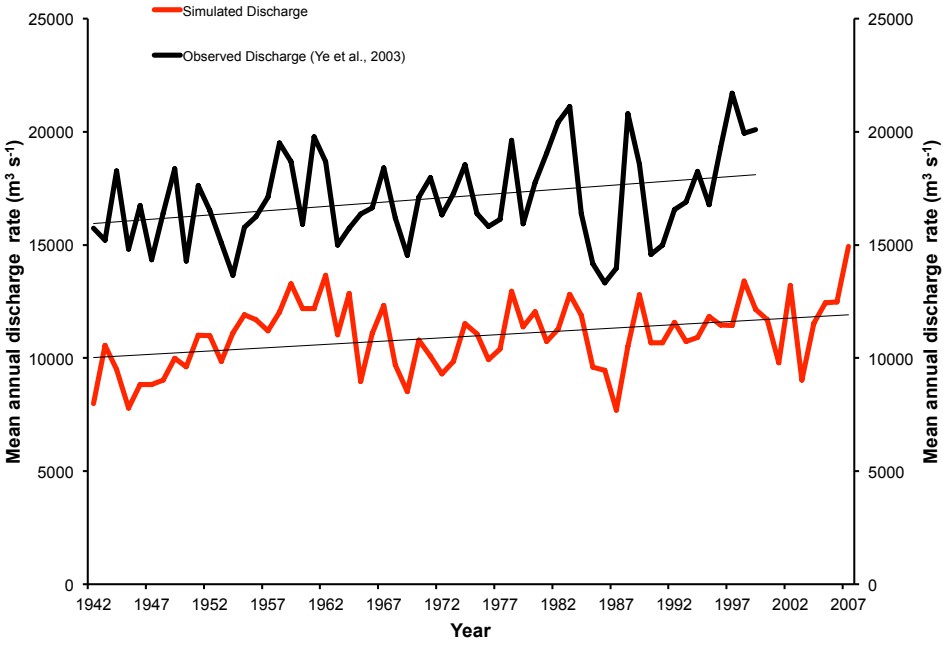






**(d)**

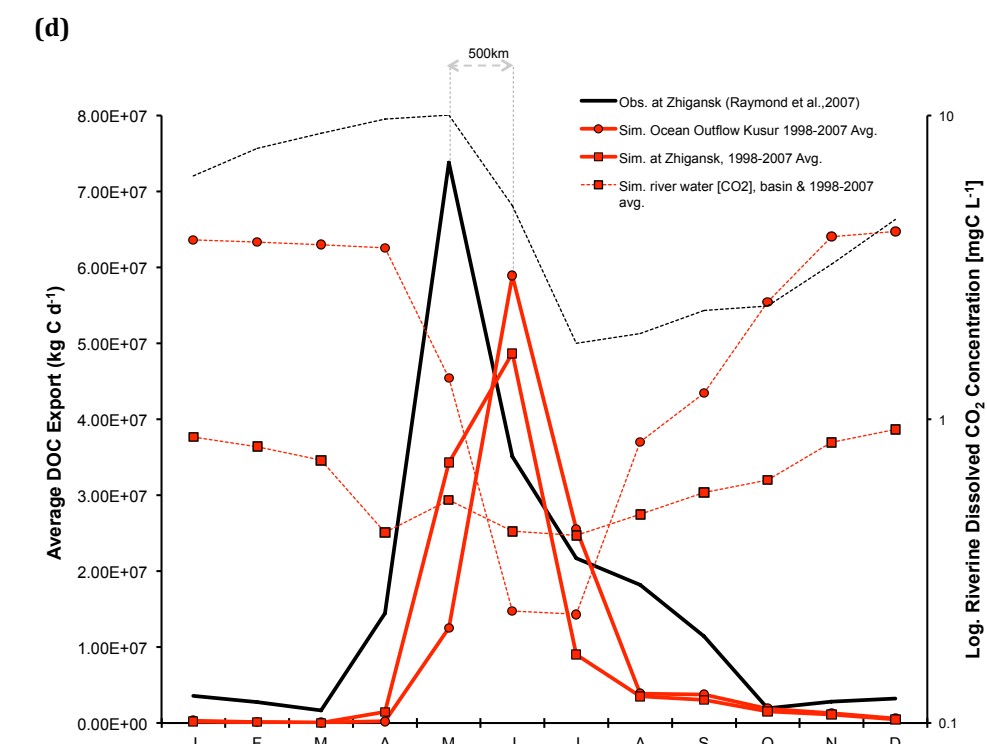

**Figure 4: (a)** Yearly DOC discharged from the Lena river into the Laptev sea is shown
here in tC yr$^{-1}$, over the entire simulation period (dashed red line), with the smoothed,
30-year running mean shown in asterisk. Observation based estimates for DOC
discharge from Lara et al. (1998), Raymond et al. (2007), Dolman et al. (2012) and
Holmes et al. (2012) are shown by the horizontal black, green triangle, blue diamond
and yellow circle line colours and symbols, respectively, and are to be compared against
the simulated mean over the last decade of simulation (1998-2007, horizontal red line),
with error bars added in grey displaying the standard deviation of simulated values over
that period. The range of estimates for total organic carbon discharged as shown in Lara
et al. (1998) are shown by the blue bounded region, where TOC here refers to DOC+POC.
**(b)** Average monthly DOC discharge (solid red, tC month$^{-1}$) and water discharge (dashed
red, m$^3$ s$^{-1}$) to the Laptev Sea over the period averaged for 1901-1910 (circles) and
1997-2007 (squares) are compared, with modern maxima closely tracking observed
values. Observed water discharge over 1936-2000 from R-ArcticNet v.4 (Lammers et al.,
2001) and published in Ye et al. (2009) are shown by the dashed black line. (c) **(d)**
Observed (black) and simulated (red) seasonal DOC fluxes (solid lines) and CO$_2$
discharge concentrations (dashed lines). Observed DOC discharge as published in
Raymond et al. (2007) from 2004-2005 observations at Zhigansk, a site ~500km
upstream of the Lena delta. This is plotted against simulated discharge for: (i) the Lena
delta at Kusur (red circles) and (ii) the approximate grid pixel corresponding to the
Zhigansk site (red squares) averaged over 1998-2008. Observed CO$_2$ discharge from a
downstream site (Cauwet & Sidorov, 1996; dashed black), and simulated from the





outflow site (dashed circle) and the basin average (dashed square) are shown on the
log-scale right-hand axis for 1998-2008.

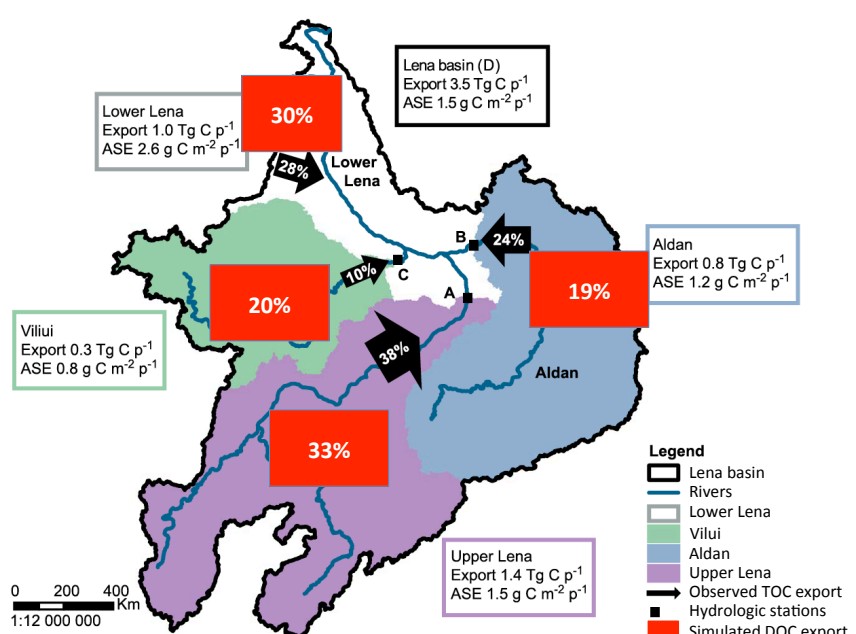

**Figure 5:** Map adapted from Fig. 2 in Kutscher et al. (2017) showing proportional sub-
basin contributions of TOC outflow to total TOC discharge in 2012-2013 as observed in
Kutscher et al., 2017 (black arrows), and DOC export contributions as simulated over the
period 1998-2007 by ORCHIDEE MICT-L (red boxes). Simulation pixels used in the
calculation are correlates of the real-world sampling locations unless the site
coordinates deviated from a mainstem hydrographic flowpath pixel –in which case a
nearest 'next-best' pixel was used.  Here the percentages are out of the summed mean
bulk DOC flow of each tributary, not the mean DOC discharge from the river mouth,
because doing so would negate the in-stream loss of DOC via degradation to $CO_2$ while
in-stream.




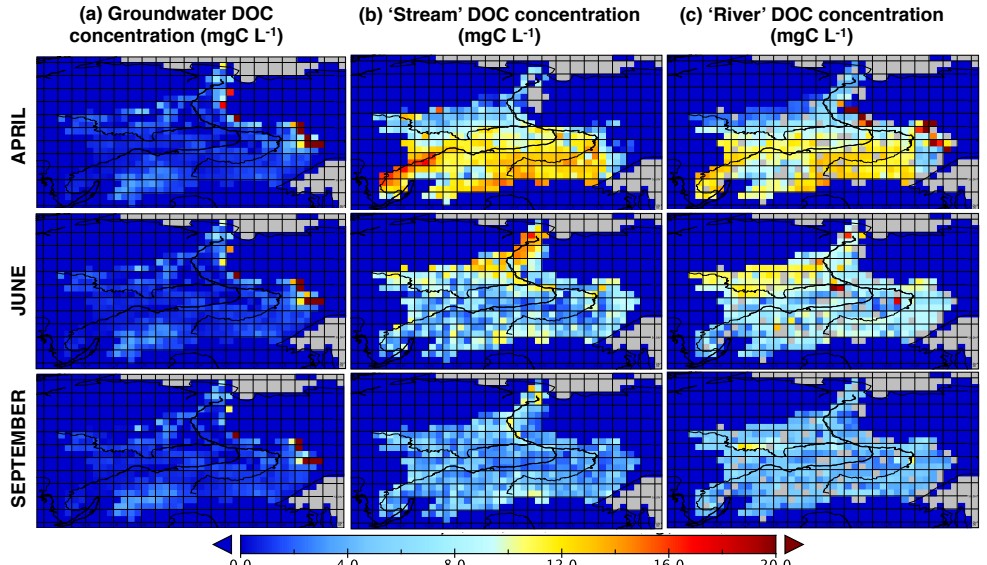

**Figure 6:** Maps of **(a)** DOC concentrations (mgC L$^{-1}$) in groundwater ('slow' water pool),
**(b)** stream water pool, **(c)** river water pool in April, June and September (first to third
rows, respectively), averaged over the period 1998-2007. The coastal boundary and a
water body overlay have been applied to the graphic in black, and the same scale applies
to all diagrams.
**(a)**

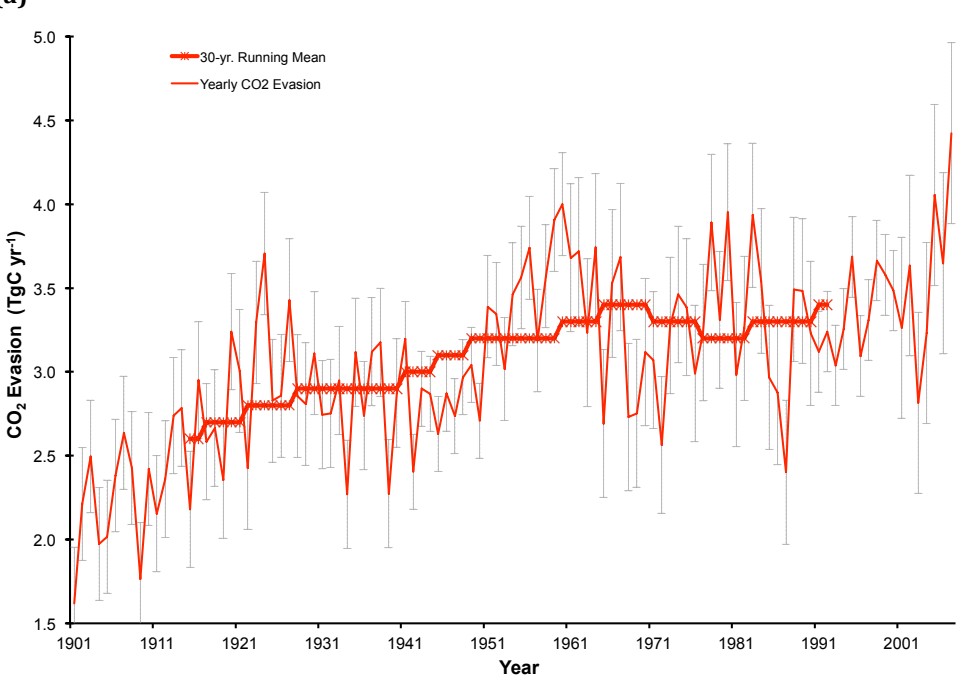




**(b)**

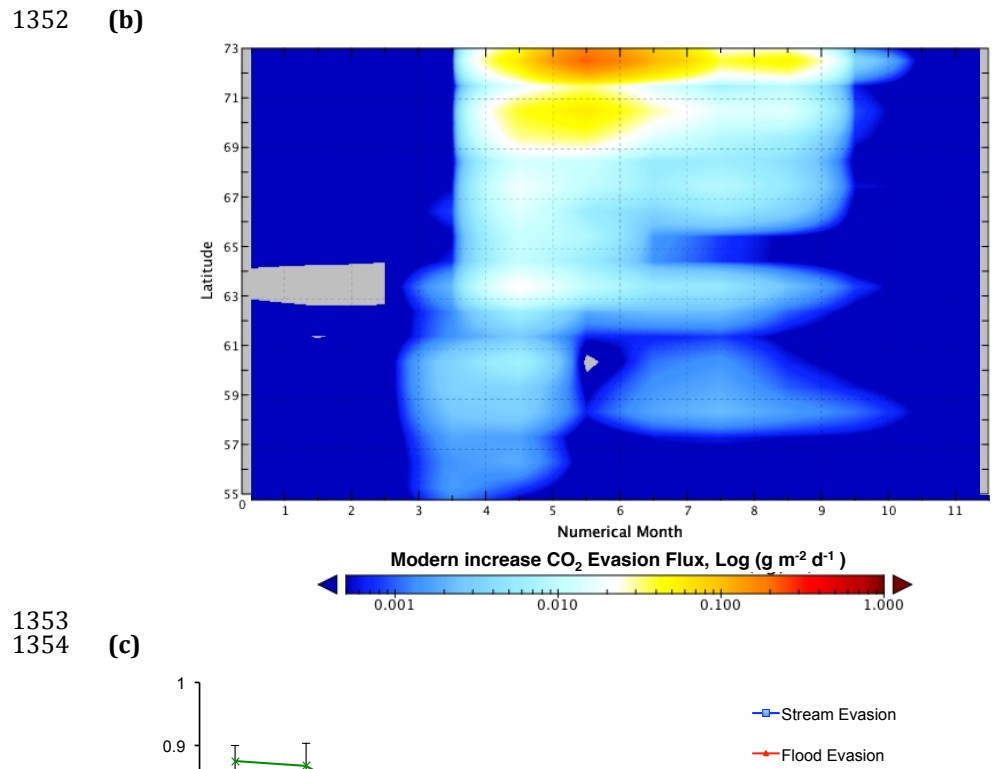

**(c)**

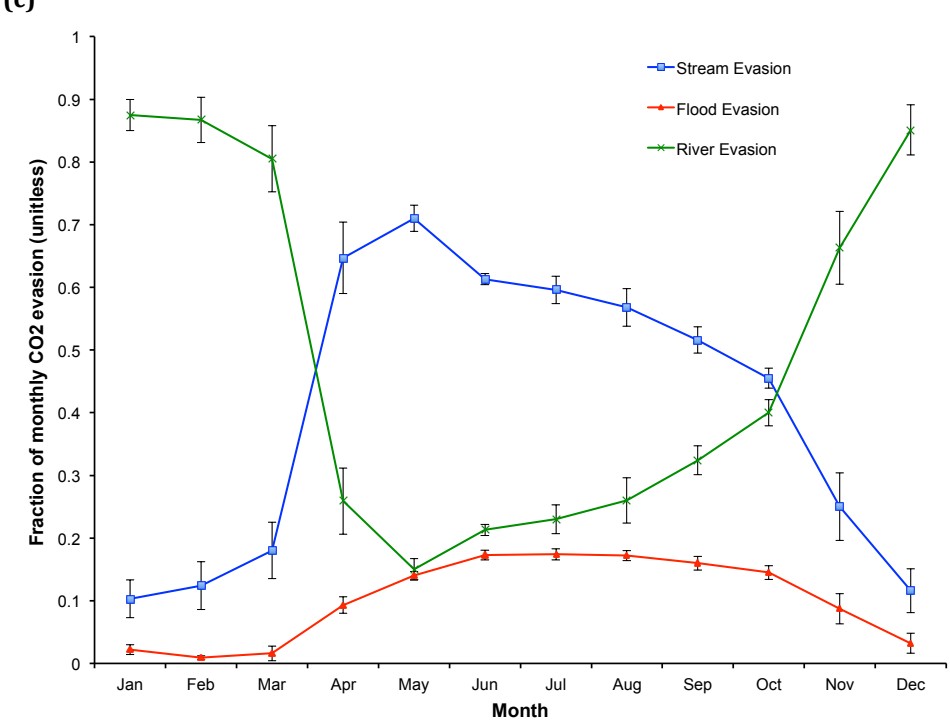

**(d)**



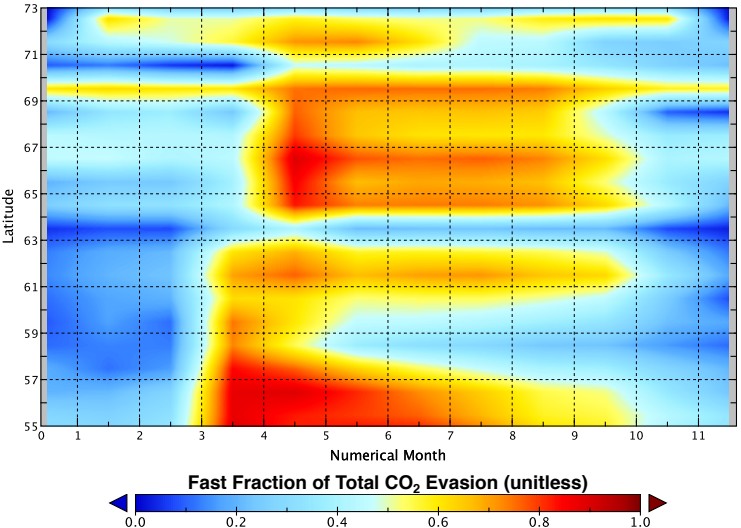

**(e)**

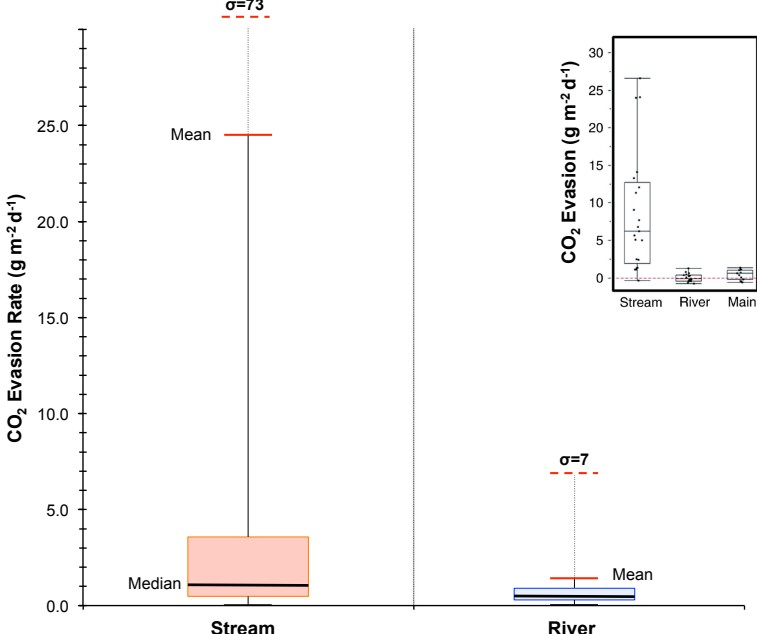

**Figure 7:** $CO_2$ evasion from stream, river, flood reservoirs. **(a)** Timeseries of total
yearly $CO_2$ evasion (tC $yr^{-1}$) summed over the three hydrological pools (red line) with





the 30-year running mean of the same variable overlain in thick red (askterisk). Error
bars give the standard deviation of each decade (e.g. 1901-1910) for each data point in
that decade. **(b)** Log-scale Hovmöller diagram plotting the longitudinally-averaged
difference (increase) in total $CO_2$ evaded from the Lena River basin between the average
of the periods 1998-2007 and 1901-1910, over each montly timestep, in (log) gC m$^{-2}$ d$^{-1}$.
Thus as the river drains northward the month-on-month difference in water-body $CO_2$
flux, between the beginning and end of the 20$^{th}$ Century is shown; **(c)** The fraction of
total $CO_2$ evasion emitted from each of the hydrological pools for the average of each
month over the period 1998-2007 is shown for river, flood and stream pools (blue,
green and red lines, respectively), with error bars depicting the standard deviation of
data values for each month displayed. **(d)** Hovmöller diagram showing the monthly
evolution of the stream pool fraction (range 0-1) per month and per latitudinal band,
averaged over the period 1998-2007. **(e)** Boxplot for approximate (see text) simulated
$CO_2$ evasion (gC m$^{-2}$ d$^{-1}$) from the streamwater reservoir and river water reservoir
averaged over 1998-2007. Coloured boxes denote the first and third quartiles of the
data range, internal black bars the median. Whiskers give the mean (solid red bar) and
standard deviation (dashed red bar) of the respective data. Empirical data on these
quantities using the same scale for rivers, streams and mainstem of the Kolyma river
from Denfeld et al., 2013 are shown inset.

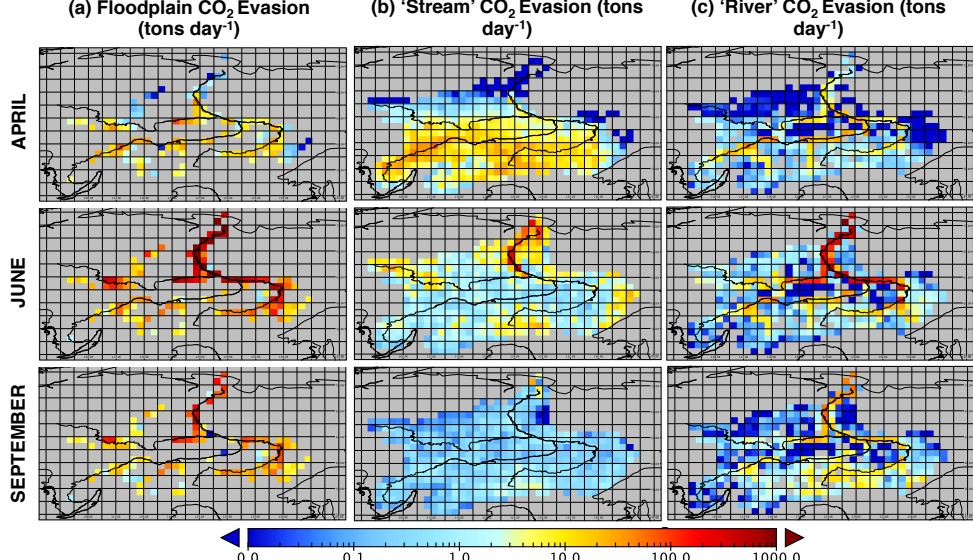

**Figure 8:** Maps of $CO_2$ evasion from the surface of the three surface hydrological pools,
(a) the floodplains, (b) streams and (c) rivers in April, June and September. All maps use
the same (log) scale in units of (tons pixel$^{-1}$ d$^{-1}$).



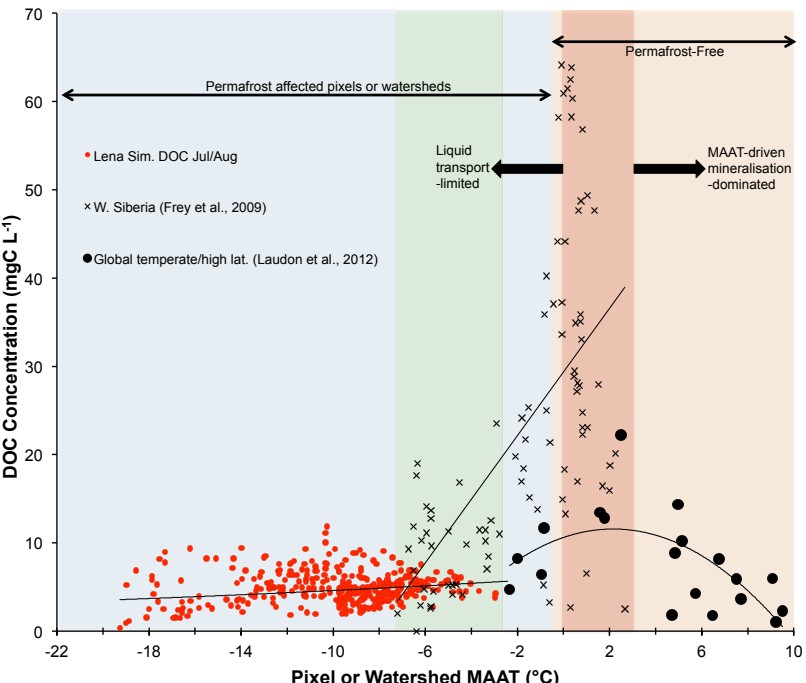

**Figure 9:** Mean summertime DOC concentrations (mgC L⁻¹) plotted against mean annual air temperature (MAAT, °Celsius) for simulated pixels over the Lena river basin (red circles), and observations for largely peat-influenced areas in western Siberia as reported in Frey et al., 2009 (black crosses), and observations from a global non-peat temperate and high latitude meta-analysis (black circles) reported in Laudon et al. (2012). The blue region represents permafrost-affected areas, while the orange region represents permafrost-free areas. The green region bounds the area of overlap in MAAT between the observed and simulated datasets. The dark red shaded area corresponds to the MAAT 'zone of optimality' for DOC production and transport proposed by Laudon et al. (2012). Regression curves of DOC against MAAT for each of the separate datasets are shown for each individual dataset.





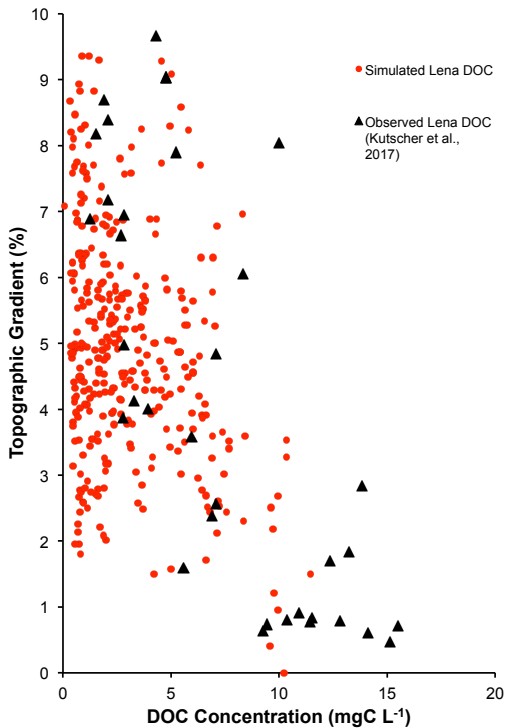

**Figure 10:** Variation of DOC concentrations versus topographic slope in Kutscher et al.,
2017 (black triangles) and (red dots) as simulated and averaged for the summer months
(JJA) over 1998-2007; observed values were measured during June and July 2012-2013.

**(a)**



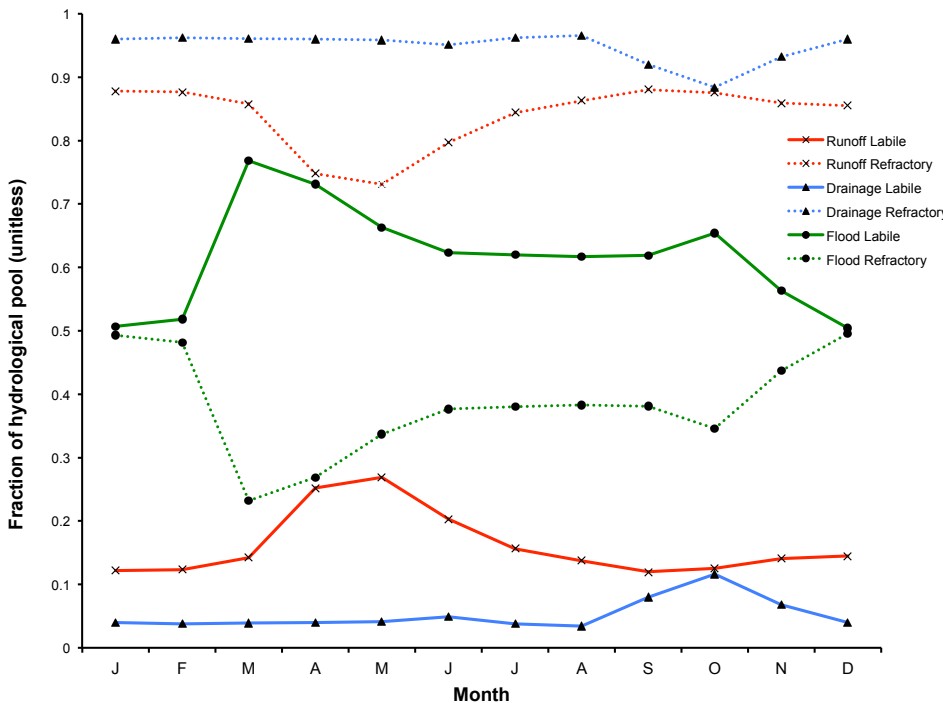

**(b)**

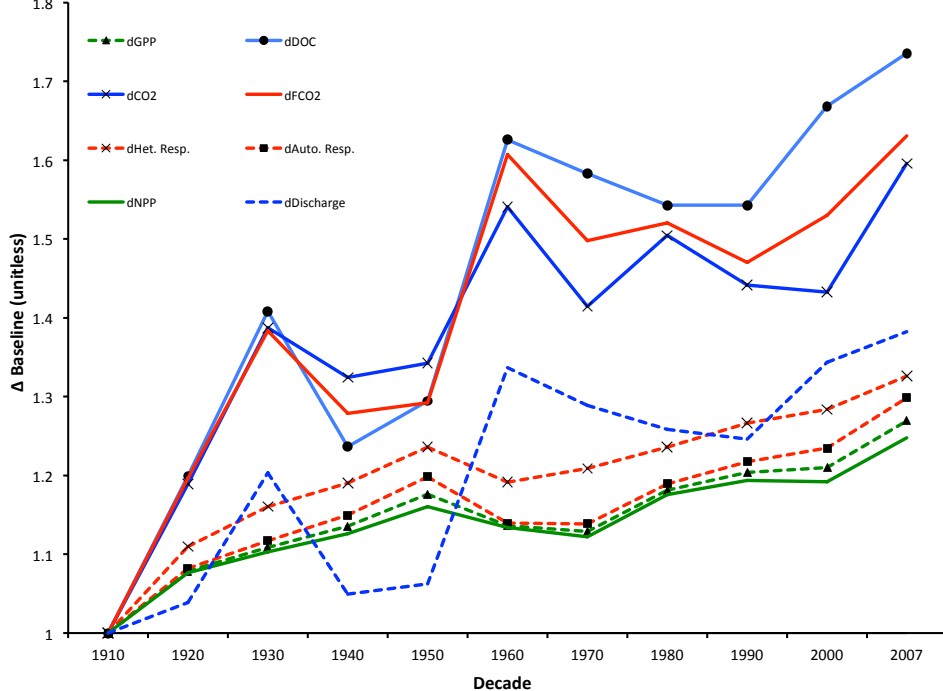






**(c)**

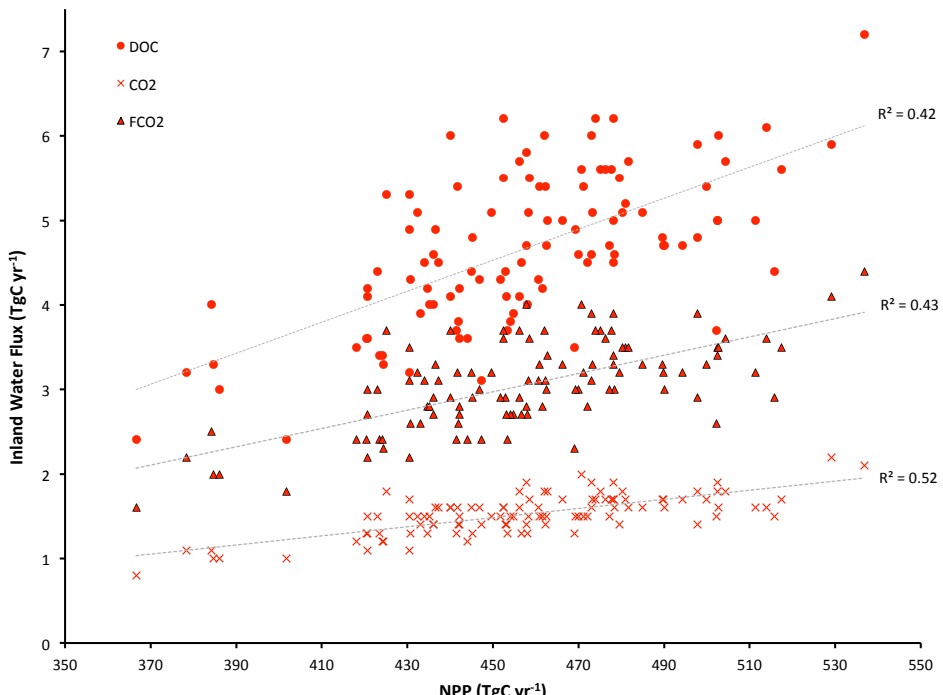

**Figure 11:** (a) The mean monthly fraction of each hydrological pool's (runoff, drainage,
floodplains) carbon reactivity constituents (labile and refractory) averaged across the
simulation area over 1998-2008. **(b)** Time series showing the decadal-mean fractional
change in carbon fluxes normalised to a 1901-1910 average baseline (=1 on the y-axis)
for NPP, GPP, autotrophic and heterotrophic respiration, DOC inputs to the water
column, $CO_2$ inputs to the water column, $CO_2$ evasion from the water surface (FCO2), and
discharge. **(c)** Summed yearly lateral flux versus NPP values for DOC discharge, $CO_2$
discharge and $CO_2$ evasion (FCO2) over the entire simulation period, with linear
regression lines shown.



**(a)**

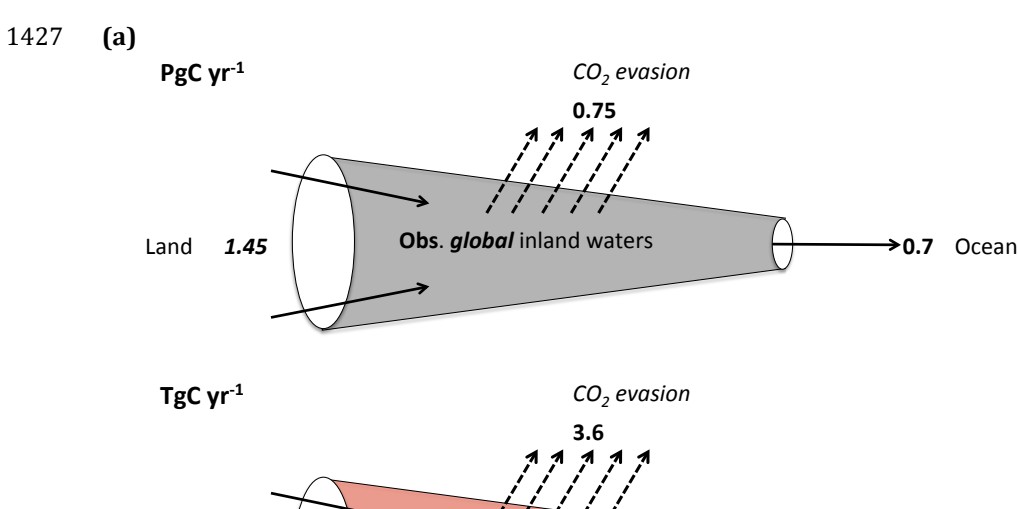

**(b)**

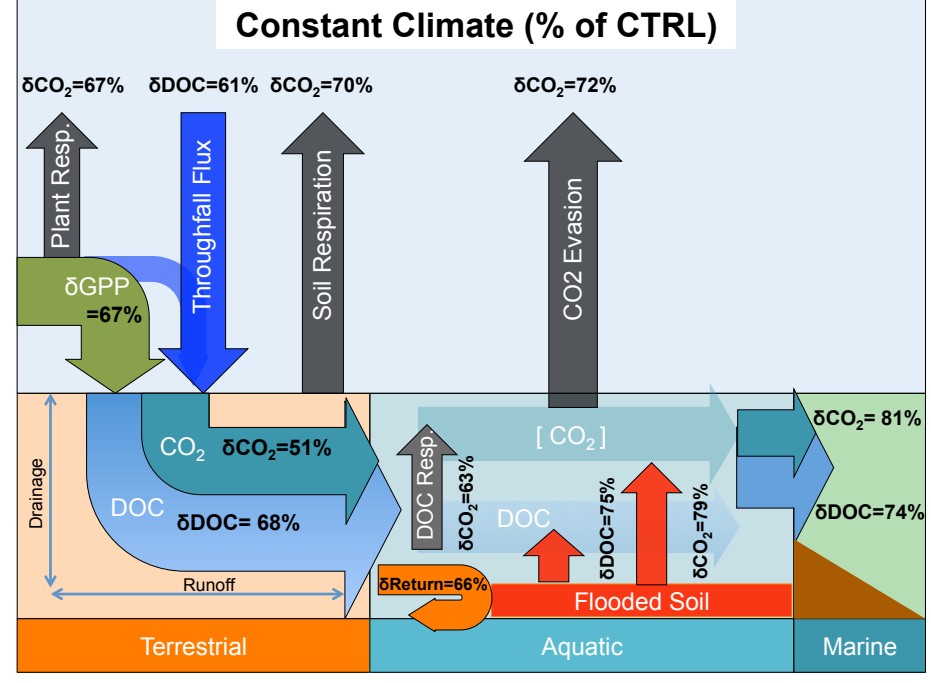

**(c)**





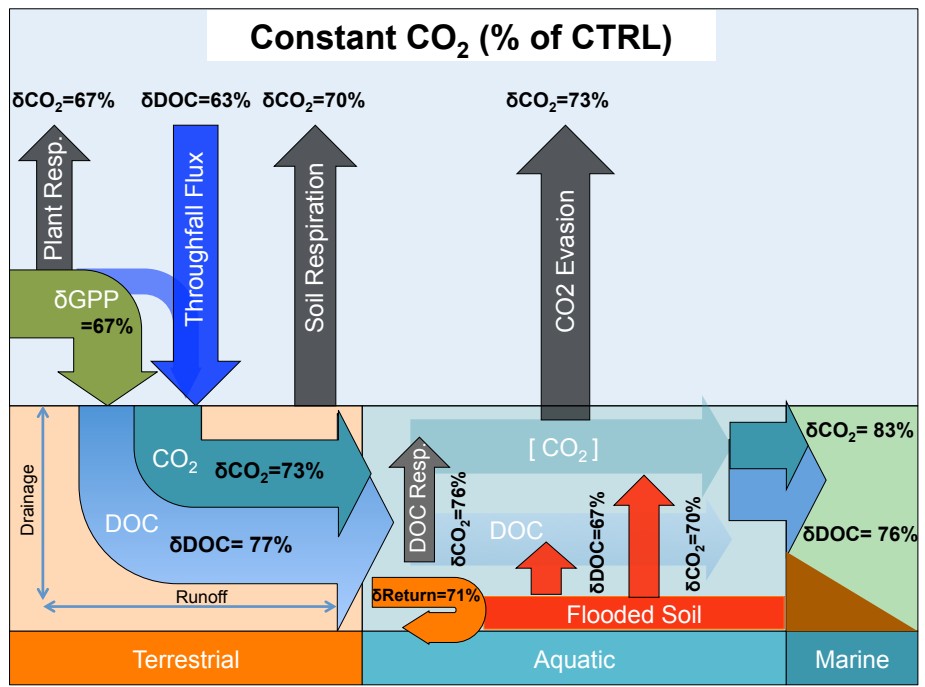

**Figure 12**: **(a)** Simplified 'leaky pipe' diagram representing the transport and
processing of DOC within the land-ocean hydrologic continuum.  The scheme template is
taken from Cole et al. (2007), where we reproduce their global estimate of DOC and non-
groundwater discharge portion of this flow in the top panel (PgC yr$^{-1}$), and the
equivalent flows from our Lena basin simulations in TgC yr$^{-1}$ in the bottom panel.  Thus
easy comparison would look at the relative fluxes within each system and compare them
to the other. **(b-c)**: Schematic diagrams detailing the major yearly carbon flux outputs
from simulations averaged over the period 1998-2007 as they are transformed and
transported across the land-aquatic continuum. Figures **(b)** and **(c)** give the same fluxes
as a percentage difference from the Control (CTRL-Simulation), for the constant climate
and CO$_2$ simulations, respectively.





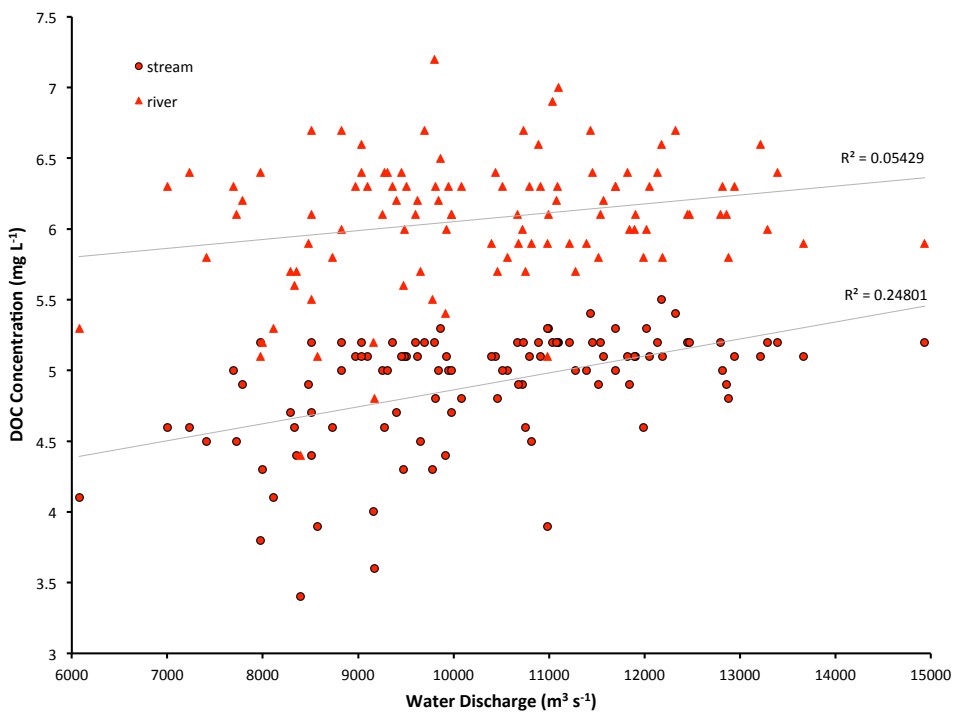

**Figure 13:** Simulated basin-mean annual DOC concentrations (mg L$^{-1}$) for the stream
and river water pools regressed against mean annual simulated discharge rates (m$^3$ s$^{-1}$)
at Kusur over 1901-2007. Linear regression plots with corresponding R$^2$ values are
shown.

1452
1453
1454