# Peer review of "ORCHIDEE MICT-LEAK (r5459), a global model for the production, transport and transformation of dissolved organic carbon from Arctic permafrost regions, Part 2: Model evaluation over the Lena River basin."

_Geoscientific Model Development, 2018_

## Referee Comment (RC1) · Anonymous Referee #1 · 2 May 2019

This portion of the study present the results of the model evaluation in the Lena River Delta. The first result that one notices is that there are some fairly significant problems with the simulated discharge, both in the timing and the magnitude. As the authors note, there is too much spring flow and the timing is a bit off, not enough growing season/fall baseflow. Plus it seems as though the model underestimates discharge by 50-100%. This is addressed in the manuscript directly, but little analysis is given about it. The failure of the model to accurately model the annual discharge results

in really substantial uncertainties in the budget for DOC exports. However, this is not at all quantified in the current analysis, which undermines any substantial conclusions on DOC export from the study. Until the issues with discharged are corrected, these numbers on DOC export and processing are most likely inaccurate. As an absolute minimum, the uncertainties resulting from the discrepancies with discharge need to be quantified.

The manuscript text in this model evaluation section is quite long and a bit disorganized. It could really use a major re-working to streamline and refine the points.

Throughout the main text & supplement: Maps all require lat/long labels (grid labels). lat/long grids necessary. Really hard to read with blue background; can't tell the watershed outline, can't differentiate terrestrial vs. Arctic Ocean.

Specific comments:

33: continuing the numbered list doesn't seem to make sense

91: check figure order

125: does is make sense to call the transient model the control?

813: I don't think Svalbard has forests

834-843: doesn't fit in this section

Figure 2: what do the CO2 numbers at the top mean?

Fig. 3: too much snowmelt, too little baseflow.

Fig 4a: legend order confusing, figure isn't super useful.

Fig 4d: extra dotted lines are confusing

Fig 5. Doesn't contain much new info, what does p-1 mean?

Fig 8: units of per pixel make not much sense.

[Figure]

Table S1: is it Tootchi et al., 2018 or Tootchi et al., 2019? Text says one, Table says another.

[Figure]

---

## Referee Comment (RC2) · Anonymous Referee #2 · 5 Jun 2019

General points

In this manuscript, using the ORCHIDEE MICT-LEAK described by the first part in accompanying paper, the authors assessed production, concentration, CO2 evasion, and riverine transport of dissolved organic carbon (DOC) over the Lena River Basin. They conducted long-term simulations and made attempts to factor out driving factors in DOC change in the study area. The research topic is potentially interesting in terms

of large-scale carbon budget, land-ocean linkage, and carbon-climate interactions. For example, the long-term increase of DOC discharge (e.g., Fig. 4a) looks intriguing, because this can affect biogeochemistry in the Arctic Ocean.

On the other hand, I have two major concerns on this manuscript. First, the simulated results were compared only with several literature data: e.g., Raymond et al. (2007), Kutscher et al. (2017), and Denfeld et al. (2013). The comparisons were not adequately quantitative, and so I could not figure out whether the model well captured observations. The low performance in simulating river discharge may indicate that the model hydrology should be improved before conducting DOC-related analyses. Second, the model simulations were conducted at a spatial resolution of 1 degree (about 100 km), but it looks too coarse to capture the spatial heterogeneity in this region. As the authors stated (Line 535), the model could not include small streams because of the coarse-scale river-routing scheme.

The manuscript provides numerous figures and text is a bit lengthy. In contrast, Simulation Rationale and Setup sections are brief and I felt inadequate. Data for comparison were described in Results and Discussion sections (e.g., Line 365–368, Line 615–621). I recommend moving these data descriptions to a section in Methods and Data. Therefore, the manuscript can be largely truncated and should be thoroughly reorganized.

Specific points

Line 45–46: In main text, no part discussed about '1.8°C warming' and '+85.6 ppm CO2 rise'. Why did you mention these values in Abstract?

Line 81: Did you examine the accuracy of GSWP3 in the study region? Especially for precipitation, some climate datasets may have serious biases.

Linen 580: Remove (g C m–2 d–1).

Line 787: Why did you discuss about NPP and soil respiration of Siberian forests in

this position of the manuscript? I lost context here.

Line 869: As long as I know, a version of ORCHIDEE (e.g., Naipal et al., 2018, Biogeosciences, 15, 4459–4480) includes POC erosion module.

Line 924: The ratio of DOC export relative to NPP, $\sim$1.5%, would be an important result but does not appears in Abstract.

---

## Author Comment (AC2) · 1 Aug 2019

Dear Anonymous Reviewer #1,

Thank you for your concise, informative and constructive assessment of this paper. In what follows, we will respond first to your general comments, followed by specific comments.

[Figure]

Response to General Comments:

General Comment 1

This portion of the study present the results of the model evaluation in the Lena River Delta. The first result that one notices is that there are some fairly significant problems with the simulated discharge, both in the timing and the magnitude. As the authors note, there is too much spring flow and the timing is a bit off, not enough growing season/fall baseflow. Plus it seems as though the model underestimates discharge by 50-100%. This is addressed in the manuscript directly, but little analysis is given about it. The failure of the model to accurately model the annual discharge results in really substantial uncertainties in the budget for DOC exports. However, this is not at all quantified in the current analysis, which undermines any substantial conclusions on DOC export from the study. Until the issues with discharged are corrected, these numbers on DOC export and processing are most likely inaccurate. As an absolute minimum, the uncertainties resulting from the discrepancies with discharge need to be quantified.

Thank you for pointing out what is clearly an issue with interpreting model results at face value. In the first part of this two-part paper, we showed that this model development version of ORCHIDEE, ORCHIDEE MICT-LEAK, was devoted to the development and inclusion of a permafrost-specific DOC and dissolved CO2 generation, transport and evasion module to the high-latitude version (ORCHIDEE-MICT) of the land surface model ORCHIDEE. The hydrology scheme in this model version is almost entirely unchanged from that latter version (ORCHIDEE-MICT), which is itself one of two parent model versions leading to this particular model instantiation. ORCHIDEE-MICT has itself already been subjected to a lengthy evaluation paper at the Pan-Arctic scale in Geoscientific Model Development (Guimberteau et al., 2018). Indeed, the sole substantial addition to the hydrology module in this model version (ORCHIDEE MICT-LEAK) is that floodplain inundation is now represented, with some significant but not cumulatively substantial impacts on water discharge. Otherwise, this work is focused

only on the production, metabolisation and transport of DOC from plant and soil matter in the Arctic. In this particular sense, our improvement of the model does not directly involve the representation of the hydrological cycle, which is itself dependent on how the surface energy balance, vegetative uptake and soil flow dynamics and, perhaps most importantly, the specific set of climatological data used as model input, are represented and read by the model, respectively. However, model output is clearly strongly impacted by these factors, both individually and in sum, and we agree that stronger quantification and explanation of the inadequacy of the hydrological module, and its effects on the DOC-generating module's results, is necessary. We also feel that a more hydrology-independent metric for model evaluation should have been used. These we address in what follows by providing further identificaiton for the factors causing low hydrological discharge, quantification of the dependency of DOC discharge error on water discharge error (DOC error (%) dependent on hydrological error), followed by summary of the remaining possible causes of error, including substantial error introduced by choice of climatological forcing dataset and, finally, introduce a new evaluation metric to evaluate DOC representation in the model that is quasi- but not fully-independent, from modelled hydrological discharge.

First, we more concretely address the causes of the model-observation mismatch in hydrological discharge, by adding the following new text to the manuscript:

[revised manuscript text omitted]

This is done to clarify that DOC discharge is indeed strongly contingent on water discharge.

Next, we compare how (obs. vs. model) DOC discharge differential (%) compares to the (obs. vs. model) river discharge differential. Then by applying the regression slope of the relationship between DOC and river discharge to the mean river discharge discrepancy of 36%, we find that 84% of the differential between observed and simulated DOC discharge can be explained by the underperformance of the hydrology module. We then go through the various other modelled modules (NPP, radiative balance, etc.)

that can affect how end-result hydrological outflows, with this largely new text:

"The observed vs. simulated mean annual water discharge differential hovers at 36% (Figs. 3d, 4c), close to the 43% differential between observed and simulated DOC discharge, giving some indication that, given the linear relationship between water and DOC discharge, most of the DOC discrepancy can be explained by the performance of the hydrology and not the DOC module, the latter of which was the subject of developments added in ORCHIDEE M-L. Applying the regression slope of the relationship in Fig. 3d (9E-06 mgC per m3s-1) to the mean river discharge discrepancy of 36%, we find that 84% of the differential between observed and simulated discharge can be explained by the underperformance of the hydrology module.

Further sources of error are process exclusion and representation/forcing limitations. Indeed, separate test runs carried out using a different set of climatological input forcing show that changing from the GSWP3 input dataset to bias-corrected projected output from the IPSL Earth System Model under the second Inter-Sectoral Impact Model Intercomparison Project (ISIMIP2b (Frieler et al., 2017; Lange, 2016, 2018)) increases DOC discharge to the ocean to 4.14 TgC yr-1 (+37%), largely due to somewhat higher precipitation rates in that forcing dataset (see Table S3). Thus, the choice of input dataset itself introduces a significant degree of uncertainty to model output.

In addition, this model does not include explicit peatland formation and related dynamics, which is the subject of further model developments (Qiu et al., 2018) yet to be included in this iteration. With peatlands thought to cover ∼17% of the Arctic land surface (Tarnocai et al., 2009), and with substantially higher leaching concentrations, this may be a significant omission from our model. The remaining biases likely arise from errors in the interaction of simulated NPP, respiration and DOC production and decomposition, which will impact on the net in and out -flow of dissolved carbon to the fluvial system. However, the DOC relationship with these variables is less clear-cut than with river discharge. Indeed, regressions (Fig. 3e) of annual DOC versus NPP (TgC yr-1) show that DOC is highly sensitive to increases in NPP, but is less coupled to it (more scattered, R2=0.42) than other simulated fluvial carbon variables shown, CO2 evasion and soil CO2 export. Thus low biases in simulated NPP can potentially strongly or weakly influence DOC export production. The differences in correlation and slope of the variables in Fig. 3e are expected: CO2 evasion is least sensitive yet most tightly coupled to NPP (R2=0.52), while CO2 export is intermediate between the two for both (R2=0.43) –CO2 export is the intermediate state between DOC export and CO2 evasion. The greater correlation (R2) with NPP of DOC compared to evasion is understandable, given that DOC leaching is a covariate of both GPP and runoff, whereas evasion flux is largely dependent on organic inputs (production) and temperature (see Part 1). "

Finally, we evaluate the seasonal DOC discharge in terms of DOC concentration, which was not done in the first draft of this manuscript. The reasoning for this is that DOC concentrations are less dependent on hydrological discharge than bulk DOC fluxes, and thus offer a clearer means by which to evaluate the DOC module as a standalone product. This evaluation shows that indeed, DOC concentrations are reasonably well represented compared to observations for the majority of the year's bulk DOC discharge, but underestimates concentrations during wintertime. The latter deficiency is consistent with the observation from Guimberteau et al. (2018) that the model poorly simulates wintertime subsurface water flow in the soil, which, by exaggerating the soil vertical impermeability of permafrost, greatly reduces the amount of DOC leachate that can be transferred to the (warmer) subsoil and laterally transferred into the river. Thus we add the following subsection to the manuscript:

"While total DOC discharge captures the integral of processes leading to fluvial biogeochemical outflow, simulations of this are highly sensitive to the performance of modelled hydrology and climatological input data. A more precise measure for the performance of the newly-introduced DOC production and transport module, that is less sensitive to reproduction of river water discharge, is DOC concentration. This is because while the total amount of DOC entering river water depends on the amount of water available as a vehicle for this flux (hydrology), the concentration of DOC depends on the rate of soil carbon leaching, itself depending largely on the interaction of soil biogeochemistry with primary production and climatic factors. This we evaluate in Figure 5a, This shows that for the majority of the thaw period or growing season (April-September), which corresponds to the period during which over 90% of DOC production and transport occurs, the model largely tracks the observed seasonality of DOC concentrations in Arctic-GRO data averaged over 1999-2007. There is a large overestimate of the DOC concentration in May owing to inaccuracies in simulating the onset of the thaw period, while the months June-September underestimate concentrations by an average of 18%. On the other hand, frozen period (November-April) DOC concentrations are underestimated by between ∼30-500%. This is due to deficiencies in representing wintertime soil hydrological water flow in the model, which impedes water flow when the soil is frozen, as discussed in Section 4.2.1. Because of this deficiency, slow-moving groundwater flows that contain large amounts of DOC leachate are under-represented. This interpretation is supported by the fact that in both observations and simulations, at low discharge rates (corresponding to wintertime), DOC concentrations exhibit a strong positive correlation with river discharge, while this relationship becomes insignificant at higher levels of river discharge (Fig. 5b). Thus wintertime DOC concentrations suffer from the same deficiencies in model representation as those for water discharge. In other words, the standalone representation of DOC leaching is satisfactory, while when it is sensitive to river discharge, it suffers from the same shortfalls identified in Section 4.2.1 and 4.2.2."

The accompanying figures to this text are shown below:

General Comment 2:

The manuscript text in this model evaluation section is quite long and a bit disorganized. It could really use a major re-working to streamline and refine the points.

We agree that the manuscript lacked some concision and could have been shortened.

[Figure]

On the other hand, both reviewers asked for some additional material to be added into the introduction, evaluation and interpretation segments of the manuscript. As such, the manuscript has been entirely edited to account for these shortcomings. In doing so, we have focussed on text readability, reducing repetition and simplifying the nature of the text itself, substantially reducing the length of the original text body. The number of textual changes are too numerous and in some cases too lengthy to enumerate piecemeal here, so we ask that you refer to the 'track-changes' version of the new manuscript draft to evaluate these changes directly. In addition, we have moved one entire subsection (Evaluation of NPP and Soil Respiration) from the main body to the Supplement (Text S2), given that this has already been evaluated, albeit at a larger scale, in Guimberteau et al. (2018) and given that its evaluation here detracts somewhat from the central foci of our manuscript.

General Comment 3:

Throughout the main text & supplement: Maps all require lat/long labels (grid labels). lat/long grids necessary. Really hard to read with blue background; can't tell the watershed outline, can't differentiate terrestrial vs. Arctic Ocean.

Thank you for spotting this issue of legibility in the manuscript. All maps have been revised as follows: (i) lon/lat labels have been introduced and increased in their font size. (ii) The terrestrial continental boundary has been included in all maps in red, with inland water body boundaries given in grey. (iii) A spatial mask has been applied to in shaded blue or grey, as shown in the following figure examples.

Response to Specific Comments:

33: continuing the numbered list doesn't seem to make sense Thank you for noticing this unnecessary notation. This has been corrected.

91: check figure order The figure order has now been totally revised.

125: does is make sense to call the transient model the control? We feel it makes sense to call the transient the control with respect to what are factorial model 'experiments' (CO2/CLIM) that hold one or another controlling climatological factor constant. We feel that for more general readership, this makes reading and understanding the results less burdensome and linguistically technocratic.

813: I don't think Svalbard has forests Thank you for spotting this. Indeed, the relevant study refers only shrubs and other small primary producers on Svalbard, and is therefore not representative of much of the vegetation overlying the Lena river basin. Reflecting this, we have removed this reference in its entirety from the text.

834-843: doesn't fit in this section Thank you for spotting this inconsistency. This has been moved to section 4.2.2 (lines 539-550) as part of the interpretation of DOC discharge dependence on NPP.

Figure 2: what do the CO2 numbers at the top mean? These refer to the fact that the carbon release from these sources or processes in model output is in gaseous form, in this case CO2. On the other hand, as noted in the figure caption, all values for carbon flux are carbon-equivalent (C) in units of Tg yr-1.

Fig. 3: too much snowmelt, too little baseflow. We assume you refer to Figure 3c, and indeed this observation is correct. We have identified, and tried to describe and explain in greater depth this part of the model output in this second draft of the manuscript (please see response to General Comment 1).

Fig 4a: legend order confusing, figure isn't super useful. Thank you for taking the time to note this. We have removed the 'total organic carbon' range from the original figure to streamline the number of sources used in this diagram. On the other hand, the general relevance of this diagram is, we feel justified, for a number of reasons. First, it lays out the modelled discharge of DOC over the 20th Century, both annually and for an annualised 30 year running mean, to show that the model outputs a long-term and unequivocal increase in DOC discharge from the Lena river over the 20th Century. This is of interest since there are no DOC discharge observations spanning this length of time, or, indeed the length of time necessary to construct a long-term observational trendline. Secondly, on the same diagram we include the average of the last ten years of simulated DOC discharge (horizontal lines) and also mark empirical estimates of the same quantity from various empirical studies within that approximate timeframe. The reasons for this are that (i) We can benchmark the trendline mentioned against any potential systematic 'gap' in observed versus simulated DOC discharge, which we show in the manuscript is indeed a systematic one derived from errors in the hydrological module. This would imply that even if modelled absolute values are inaccurate relative to observations, the trendline might still reflect a real tendency over the 20th Century. (ii) We discuss the difference in observational estimates and, despite coming to the conclusion that the latest estimates are likely the most accurate, include them all in the diagram to illustrate that the empirical numbers are at the end of the day also estimates.

Fig 4d: extra dotted lines are confusing Thank you for this observation. We agree that the diagram is not necessarily the easiest to follow, as is often the case for dual-axis figures, but it is our opinion that directly comparing the modelled and observed DOC and CO2 seasonality is useful for interpreting how and whether these two variables evolve and/or co-evolve over the course of the year. For this reason we don't feel that separating these two variables is in the interest of the manuscript.

Fig 5. Doesn't contain much new info, what does p-1 mean? We agree that this is perhaps not the most interesting facet of the model output, and have moved the figure to the Supplement (Fig. S2). The (p-1) is carried over from the 'p'eriod used as a temporal unit in Kutscher et al. (2017) from whom this figure is directly derived. The unit explained in the accompanying figure caption, by the sentence: "Map adapted from Fig. 2 in Kutscher et al. (2017) showing proportional sub-basin contributions of TOC outflow to total TOC discharge in June and July (designated as their sampling period 'p-1') of 2012-2013, as observed in Kutscher et al., 2017 (black arrows)".

Fig 8: units of per pixel make not much sense. This has been changed to units of mgC m-2d-1, and to increase readability, we have removed one of the sub-figures (floodplains) from the diagram.

Table S1: is it Tootchi et al., 2018 or Tootchi et al., 2019? Text says one, Table says another.

Thank you for your diligence, this error has been corrected.

Please also note the supplement to this comment:
https://www.geosci-model-dev-discuss.net/gmd-2018-322/gmd-2018-322-AC2-supplement.pdf
* * *
*Table S3: Observed versus simulated DOC discharge (1998-2007), where we compare the output of two separate climatological datasets used as input to the model (GSWP3 and ISIMIP 2b). Also shown are the simulated versus observed DOC discharge for the six largest Arctic rivers (the "Big Six") and for the Pan-Arctic as a whole.*

|  | Simulated DOC to Ocean GSWP3 | Simulated DOC to Ocean ISIMIP 2b | Observations (Holmes et al., 2012) PARTNERS/Arctic-GRO |
|---|---|---|---|
| **Lena** | 3.16 | 4.14 | 5.68 |
| **Big 6** |  | 19.36 | 18.11 |
| **Pan-Arctic** |  | 32.06 | 34.04 |

**Fig. 1.**

[Figure]

*Figure 5: (a) Simulated and observed (Arctic-GRO/Holmes et al., 2012) DOC concentration seasonality for the Lena basin over the period 1999-2007. (b) Plots of DOC concentration versus river discharge as in observations (Raymond et al., 2007) and simulations, where simulations data points are monthly averages taken over the period 1999-2007*

**Fig. 2.**

**(a) 'Stream' CO₂ Evasion (mgC m⁻² d⁻¹)**   **(b) 'River' CO₂ Evasion (mgC m⁻² d⁻¹)**

**Figure 8:** Maps of $CO_2$ evasion from the surface of the two fluvial hydrological pools in the model, (a) streams and (b) rivers in April, June and September. All maps use the same (log) scale in units of (mgC m⁻² d⁻¹).

**Fig. 3.**

**(a) Groundwater DOC concentration (mgC L⁻¹)** **(b) 'Stream' DOC concentration (mgC L⁻¹)** **(c) 'River' DOC concentration (mgC L⁻¹)**

APRIL

JUNE

SEPTEMBER

[revised manuscript text omitted]

| | Empirical Evaluation Sources |
|---|---|
| DOC Discharge | Cauwet and Sidorov (1996); Dolman et al. (2012); Holmes et al. (2012); Lara et al. (1998); Raymond et al. (2007); Semiletov et al. (2011); Kutscher et al. (2017). |
| Water Discharge | Ye et al. (2009); Lammers et al. (2001) |
| DOC concentration | Shvartsev (2008); Denfeld et al. (2013); Mann et al. (2015); Raymond et al. (2007); Semiletov et al. (2011); Arctic-GRO/PARTNERS (Holmes et al., 2012) |
| NPP | Beer et al. (2006); Lloyd et al. (2002); Roser et al. (2002); Schulze et al. (1999); Shvidenko and Nilsson, (2003) |
| Soil Respiration | Elberling (2007); Sawamoto et al. (2000); Sommerkorn (2008). |
| CO2 Evasion | Denfeld et al. (2013); Serikova et al. (2018). |

**Table S3**: Observed versus simulated DOC discharge (1998-2007), where we compare
the output of two separate climatological datasets used as input to the model (GSWP3
and ISIMIP 2b). Also shown are the simulated versus observed DOC discharge for the six
largest Arctic rivers (the "Big Six") and for the Pan-Arctic as a whole.

| | Simulated DOC to Ocean | Simulated DOC to Ocean | Observations (Holmes et al., 2012) |
|---|---|---|---|
| | **GSWP3** | **ISIMIP 2b** | **PARTNERS/Arctic-GRO** |
| **Lena** | 3.16 | 4.14 | 5.68 |
| **Big 6** | | 19.36 | 18.11 |
| **Pan-Arctic** | | 32.06 | 34.04 |

**(a)**

[Figure]

**(c)**

[Figure]

**Figure S1: (a-b)** Carbon and water flux map for core DOC elements in model structure
relating to DOC transport and transformation, first published in Part 1 of this study. **(a)**
Summary of the differing extent of vertical discretisation of soil and snow for different
processes calculated in the model. Discretisation occurs along 32 layers whose thickness
increases geometrically from 0-38m. N refers to the number of layers, SWE=snow water
equivalent, $S_n$ = Snow layer n. Orange layers indicate the depth to which diffusive carbon
(turbation) fluxes occur. **(b)** Conceptual map of the production, transfer and
transformation of carbon in its vertical and lateral (i.e., hydrological) flux as calculated
in the model. Red boxes indicate meta-reservoirs of carbon, black boxes the actual pools
as they exist in the model. Black arrows indicate carbon fluxes between pools, dashed
red arrows give carbon loss as $CO_2$, green arrows highlight the fractional distribution of
DOC to SOC (no carbon loss incurred in this transfer), a feature of this model. For a given
temperature (5°C) and soil clay fraction, the fractional fluxes between pools are given
for each flux, while residence times for each pool ( $\tau$ ) are in each box. The association of
carbon dynamics with the hydrological module are shown by the blue arrows. Blue
coloured boxes illustrate the statistical sequence which activates the boolean
floodplains module. Note that for readability, the generation and lateral flux of
dissolved $CO_2$ is omitted from this diagram, but is described at length in the Methods
section. **(c)** (Left) Soil carbon concentrations per depth level for each soil carbon
reactivity pool at the end of the spinup period. (Right) Evolution of each soil carbon pool
over the course of the 400-year spinup quasi-eqliuibration period.

[Figure]

**Figure S2:** Map adapted from Fig. 2 in Kutscher et al. (2017) showing proportional sub-
basin contributions of TOC outflow to total TOC discharge in June and July (designated
as their sampling period 'p$^{-1}$') of 2012-2013, as observed in Kutscher et al., 2017 (black
arrows), and DOC export contributions as simulated over the period 1998-2007 by
ORCHIDEE MICT-L (red boxes). Simulation pixels used in the calculation are correlates
of the real-world sampling locations unless the site coordinates deviated from a
mainstem hydrographic flowpath pixel –in which case a nearest 'next-best' pixel was
used. Here the percentages are out of the summed mean bulk DOC flow of each
tributary, not the mean DOC discharge from the river mouth, because doing so would
negate the in-stream loss of DOC via degradation to $CO_2$ while in-stream.

(a)

[Figure]

(b)

[Figure]

**Figure S3:** (a) Maximum floodable fraction of grid cells for the Lena basin per the input
map from Tootchi et al. (2018). (b) Podzol and Arenosol map (Nachtergaele, Freddy,
Harrij van Velthuizen, Luc Verelst, N. H. Batjes, Koos Dijkshoorn, V. W. P. van Engelen,
Guenther Fischer, Arwyn Jones, 2010) used as input to the 'poor soils' module, basin
mask in the background.

[Figure]

**Figure S4:** Groundwater DOC concentrations over the Lena basin for April, June and
September averaged over 1998-2007, with mean observed concentrations for
permafrost groundwater inset.

(a)

[Figure]

(b)

[Figure]

(c)

(d)

[Figure]

**Figure S5**: (a) Absolute yearly gross primary productivity (GPP, TgC yr-1) for the four
relevant PFT groups over the Lena basin, averaged over 1998-2007.  (b) Mean July and
August soil heterotrophic respiration rates (g m² d-1) for the same PFT groups as in (a),
during the period 1998-2007.  (c) Average yearly NPP (gC m² yr-1) averaged over the
period 1998-2007.  All maps have the Lena basin area shaded in the background. (d)
Mean monthly carbon uptake (GPP) versus its heterotrophic respiration from the soil
(Het_Resp) in TgC per month, over the period 1998-2007.

[Figure]

**Figure S6:** Simulated basin-mean annual DOC concentrations (mg L$^{-1}$) for the floodplain
water pool regressed against mean annual simulated discharge rates at Kusur (m$^3$ s$^{-1}$)
over 1901-2007.  A linear regression with R$^2$ is plotted.

[Figure]

**Figure S7:** Time series showing the decadal-mean fractional change in carbon fluxes
normalised to a 1901-1910 average baseline (=1 on the y-axis) for NPP, GPP,
autotrophic and heterotrophic respiration, DOC inputs to the water column, $CO_2$ inputs
to the water column, $CO_2$ evasion from the water surface (FCO2), and discharge.

---

## Author Response (AR1)

Dear Anonymous Reviewer #1,
Thank you for your concise, informative and constructive assessment of this paper. In
what follows, we will respond first to your general comments, followed by specific
comments.
**Response to General Comments:**
General Comment 1
**This portion of the study present the results of the model evaluation in the Lena**
**River Delta. The first result that one notices is that there are some fairly**
**significant problems with the simulated discharge, both in the timing and the**
**magnitude. As the authors note, there is too much spring flow and the timing is a**
**bit off, not enough growing season/fall baseflow. Plus it seems as though the**
**model underestimates discharge by 50-100%. This is addressed in the manuscript**
**directly, but little analysis is given about it. The failure of the model to accurately**
**model the annual discharge results in really substantial uncertainties in the**
**budget for DOC exports. However, this is not at all quantified in the current**
**analysis, which undermines any substantial conclusions on DOC export from the**
**study. Until the issues with discharged are corrected, these numbers on DOC**
**export and processing are most likely inaccurate. As an absolute minimum, the**
**uncertainties resulting from the discrepancies with discharge need to be**
**quantified.**
Thank you for pointing out what is clearly an issue with interpreting model results at
face value.  In the first part of this two-part paper, we showed that this model
development version of ORCHIDEE, ORCHIDEE MICT-LEAK, was devoted to the
development and inclusion of a permafrost-specific DOC and dissolved $CO_2$ generation,
transport and evasion module to the high-latitude version (ORCHIDEE-MICT) of the land
surface model ORCHIDEE.  The hydrology scheme in this model version is almost
entirely unchanged from that latter version (ORCHIDEE-MICT), which is itself one of two
parent model versions leading to this particular model instantiation.  ORCHIDEE-MICT
has itself already been subjected to a lengthy evaluation paper at the Pan-Arctic scale in
Geoscientific Model Development (Guimberteau et al., 2018).   Indeed, the sole
substantial addition to the hydrology module in this model version (ORCHIDEE MICT-
LEAK) is that floodplain inundation is now represented, with some significant but not
cumulatively substantial impacts on water discharge. Otherwise, this work is focused
only on the production, metabolisation and transport of DOC from plant and soil matter
in the Arctic. In this particular sense, our improvement of the model does not directly
involve the representation of the hydrological cycle, which is itself dependent on how
the surface energy balance, vegetative uptake and soil flow dynamics and, perhaps most
importantly, the specific set of climatological data used as model input, are represented and read by the model, respectively. However, model output is clearly strongly impacted by these factors, both individually and in sum, and we agree that stronger quantification and explanation of the inadequacy of the hydrological module, and its effects on the DOC-generating module's results, is necessary. We also feel that a more hydrology-independent metric for model evaluation should have been used. These we address in what follows by providing further identificaiton for the factors causing low hydrological discharge, quantification of the dependency of DOC discharge error on water discharge error (DOC error (%) dependent on hydrological error), followed by summary of the remaining possible causes of error, including substantial error introduced by choice of climatological forcing dataset and, finally, introduce a new evaluation metric to evaluate DOC representation in the model that is quasi- but not fully- independent, from modelled hydrological discharge.

First, we more concretely address the causes of the model-observation mismatch in hydrological discharge, by adding the following new text to the manuscript:

[revised manuscript text omitted]

This is done to clarify that DOC discharge is indeed strongly contingent on water discharge.

Next, we compare how (obs. vs. model) DOC discharge differential (%) compares to the (obs. vs. model) river discharge differential. Then by applying the regression slope of the relationship between DOC and river discharge to the mean river discharge discrepancy of 36%, we find that 84% of the differential between observed and simulated DOC discharge can be explained by the underperformance of the hydrology module. We then go through the various other modelled modules (NPP, radiative balance, etc.) that can affect how end-result hydrological outflows, with this largely new text:

*"The observed vs. simulated mean annual water discharge differential hovers at 36% (Figs. 3d, 4c), close to the 43% differential between observed and simulated*

*DOC discharge, giving some indication that, given the linear relationship between*
*water and DOC discharge, most of the DOC discrepancy can be explained by the*
*performance of the hydrology and not the DOC module, the latter of which was the*
*subject of developments added in ORCHIDEE M-L.   Applying the regression slope of*
*the relationship in Fig. 3d (9E-06 mgC per $m^3s^{-1}$) to the mean river discharge*
*discrepancy of 36%, we find that 84% of the differential between observed and*
*simulated discharge can be explained by the underperformance of the hydrology*
*module.*
*Further sources of error are process exclusion and representation/forcing*
*limitations. Indeed, separate test runs carried out using a different set of*
*climatological input forcing show that changing from the GSWP3 input dataset to*
*bias-corrected projected output from the IPSL Earth System Model under the second*
*Inter-Sectoral Impact Model Intercomparison Project (ISIMIP2b (Frieler et al., 2017;*
*Lange, 2016, 2018)) increases DOC discharge to the ocean to 4.14 TgC $yr^{-1}$ (+37%),*
*largely due to somewhat higher precipitation rates in that forcing dataset (see*
*Table S3).  Thus, the choice of input dataset itself introduces a significant  degree of*
*uncertainty to model output.*
*In addition, this model does not include explicit peatland formation and related*
*dynamics, which is the subject of further model developments (Qiu et al., 2018)  yet*
*to be included in this iteration. With peatlands thought to cover ~17% of the Arctic*
*land surface (Tarnocai et al., 2009), and with substantially higher leaching*
*concentrations, this may be a significant omission from our model. The remaining*
*biases likely arise from errors in the interaction of simulated NPP, respiration and*
*DOC production and decomposition, which will impact on the net in and out -flow of*
*dissolved carbon to the fluvial system.   However, the DOC relationship with these*
*variables is less clear-cut than with river discharge. Indeed, regressions (Fig. 3e) of*
*annual DOC versus NPP (TgC $yr^{-1}$) show that DOC is highly sensitive to increases in*
*NPP, but is less coupled to it (more scattered, $R^2=0.42$) than other simulated fluvial*
*carbon variables shown, $CO_2$ evasion and soil $CO_2$ export.   Thus low biases in*
*simulated NPP can potentially strongly or weakly influence DOC export production.*
*The  differences in correlation and slope of the variables in Fig. 3e are expected: $CO_2$*
*evasion is least sensitive yet most tightly coupled to NPP ($R^2=0.52$), while $CO_2$ export*
*is intermediate between the two for both ($R^2=0.43$) –$CO_2$ export is the intermediate*
*state between DOC export and $CO_2$ evasion. The greater correlation ($R^2$) with NPP of*
*DOC compared to evasion is understandable, given that DOC leaching is a covariate*
*of both GPP and runoff, whereas evasion flux is largely dependent on organic inputs*
*(production) and temperature (see Part 1).*
*"*
*Table S3: Observed versus simulated DOC discharge (1998-2007), where we*
*compare the output of two separate climatological datasets used as input to the*
*model (GSWP3 and ISIMIP 2b).  Also shown are the simulated versus observed DOC*
*discharge for the six largest Arctic rivers (the "Big Six") and for the Pan-Arctic as a*
*whole.*

| | Simulated DOC to Ocean | Simulated DOC to Ocean | Observations (Holmes et al., 2012) |
|---|---|---|---|
| | GSWP3 | ISIMIP 2b | PARTNERS/Arctic-GRO |
| Lena | 3.16 | 4.14 | 5.68 |
| Big 6 | | 19.36 | 18.11 |
| Pan-Arctic | | 32.06 | 34.04 |

Finally, we evaluate the seasonal DOC discharge in terms of DOC concentration, which was not done in the first draft of this manuscript. The reasoning for this is that DOC concentrations are less dependent on hydrological discharge than bulk DOC fluxes, and thus offer a clearer means by which to evaluate the DOC module as a standalone product. This evaluation shows that indeed, DOC concentrations are reasonably well represented compared to observations for the majority of the year's **bulk** DOC discharge, but underestimates concentrations during wintertime. The latter deficiency is consistent with the observation from Guimberteau et al. (2018) that the model poorly simulates wintertime subsurface water flow in the soil, which, by exaggerating the soil vertical impermeability of permafrost, greatly reduces the amount of DOC leachate that can be transferred to the (warmer) subsoil and laterally transferred into the river. Thus we add the following subsection to the manuscript:

*"While total DOC discharge captures the integral of processes leading to fluvial biogeochemical outflow, simulations of this are highly sensitive to the performance of modelled hydrology and climatological input data. A more precise measure for the performance of the newly-introduced DOC production and transport module, that is less sensitive to reproduction of river water discharge, is DOC concentration. This is because while the total amount of DOC entering river water depends on the amount of water available as a vehicle for this flux (hydrology), the concentration of DOC depends on the rate of soil carbon leaching, itself depending largely on the interaction of soil biogeochemistry with primary production and climatic factors. This we evaluate in Figure 5a, This shows that for the majority of the thaw period or growing season (April-September), which corresponds to the period during which over 90% of DOC production and transport occurs, the model largely tracks the observed seasonality of DOC concentrations in Arctic-GRO data averaged over 1999-2007. There is a large overestimate of the DOC concentration in May owing to inaccuracies in simulating the onset of the thaw period, while the months June-September underestimate concentrations by an average of 18%. On the other hand, frozen period (November-April) DOC concentrations are underestimated by between ~30-500%. This is due to deficiencies in representing wintertime soil hydrological water flow in the model, which impedes water flow when the soil is frozen, as discussed in Section 4.2.1. Because of this deficiency, slow-moving groundwater flows that contain large amounts of DOC leachate are under-represented. This interpretation is supported by the fact that in both observations and simulations, at low discharge rates (corresponding to wintertime), DOC concentrations exhibit a strong positive correlation with river discharge, while this relationship becomes insignificant at higher levels of river discharge (Fig. 5b). Thus wintertime DOC concentrations suffer from the same deficiencies in model representation as those for water discharge. In other words, the standalone representation of DOC leaching is satisfactory, while when it is sensitive to river discharge, it suffers from the same*

*shortfalls identified in Section 4.2.1 and 4.2.2."*
The accompanying figures to this text are shown below:

[Figure]

*Figure 5: (a) Simulated and observed (Arctic-GRO/Holmes et al., 2012) DOC*
*concentration seasonality for the Lena basin over the period 1999-2007. (b) Plots of*
*DOC concentration versus river discharge as in observations (Raymond et al., 2007)*
*and simulations, where simulations data points are monthly averages taken over*
*the period 1999-2007*
General Comment 2:

**The manuscript text in this model evaluation section is quite long and a bit**
**disorganized. It could really use a major re-working to streamline and refine the**
**points.**
We agree that the manuscript lacked some concision and could have been shortened.  On
the other hand, both reviewers asked for some additional material to be added into the
introduction, evaluation and interpretation segments of the manuscript.  As such, the
manuscript has been entirely edited to account for these shortcomings.  In doing so, we
have focussed on text readability, reducing repetition and simplifying the nature of the
text itself, substantially reducing the length of the original text body. The number of
textual changes are too numerous and in some cases too lengthy to enumerate
piecemeal here, so we ask that you refer to the 'track-changes' version of the new
manuscript draft to evaluate these changes directly.  In addition, we have moved one
entire subsection (Evaluation of NPP and Soil Respiration) from the main body to the
Supplement (Text S2), given that this has already been evaluated, albeit at a larger scale,
in Guimberteau et al. (2018) and given that its evaluation here detracts somewhat from
the central foci of our manuscript.
General Comment 3:
**Throughout the main text & supplement: Maps all require lat/long labels (grid**
**labels). lat/long grids necessary. Really hard to read with blue background; can't**
**tell the watershed outline, can't differentiate terrestrial vs. Arctic Ocean.**
Thank you for spotting this issue of legibility in the manuscript.  All maps have been
revised as follows: (i) lon/lat labels have been introduced and increased in their font
size.  (ii) The terrestrial continental boundary has been included in all maps in red, with
inland water body boundaries given in grey.  (iii) A spatial mask has been applied to in
shaded blue or grey, as shown in the following figure examples.

[Figure]

**Figure 6:** Maps of **(a)** DOC concentrations (mgC L$^{-1}$) in groundwater ('slow' water pool),
**(b)** stream water pool, **(c)** river water pool in April, June and September (first to third
rows, respectively), averaged over the period 1998-2007. The coastal boundary and a
water body overlay have been applied to the graphic in red and black, respectively, and
the same scale applies to all diagrams. All maps have the Lena basin area shaded in the
background.

[Figure]

**Figure 8:** Maps of $CO_2$ evasion from the surface of the two fluvial hydrological pools in the model, (a) streams and (b) rivers in April, June and September. All maps use the same (log) scale in units of (mgC $m^{-2}$ $d^{-1}$).

[Figure]

[Figure]

**Figure S4:** Groundwater DOC concentrations over the Lena basin for April, June and
September averaged over 1998-2007, with mean observed concentrations for
permafrost groundwater inset.

[Figure]

[Figure]

**Figure S5**: (a) Absolute yearly gross primary productivity (GPP, TgC yr-1) for the four relevant PFT groups over the Lena basin, averaged over 1998-2007. (b) Mean July and August soil heterotrophic respiration rates (g m² d-1) for the same PFT groups as in (a), during the period 1998-2007. (c) Average yearly NPP (gC m² yr-1) averaged over the period 1998-2007. All maps have the Lena basin area shaded in the background.

**Response to Specific Comments:**

**33: continuing the numbered list doesn't seem to make sense**
Thank you for noticing this unnecessary notation. This has been corrected.
**91: check figure order**
The figure order has now been totally revised.
**125: does is make sense to call the transient model the control?**
We feel it makes sense to call the transient the control with respect to what are factorial
model 'experiments' (CO2/CLIM) that hold one or another controlling climatological
factor constant. We feel that for more general readership, this makes reading and
understanding the results less burdensome and linguistically technocratic.
**813: I don't think Svalbard has forests**
Thank you for spotting this. Indeed, the relevant study refers only shrubs and other
small primary producers on Svalbard, and is therefore not representative of much of the
vegetation overlying the Lena river basin. Reflecting this, we have removed this
reference in its entirety from the text.
**834-843: doesn't fit in this section**
Thank you for spotting this inconsistency. This has been moved to section 4.2.2 (lines
539-550) as part of the interpretation of DOC discharge dependence on NPP.
**Figure 2: what do the CO2 numbers at the top mean?**
These refer to the fact that the carbon release from these sources or processes in model
output is in gaseous form, in this case $CO_2$. On the other hand, as noted in the figure
caption, all values for carbon flux are carbon-equivalent (C) in units of Tg yr$^{-1}$.
**Fig. 3: too much snowmelt, too little baseflow.**
We assume you refer to Figure 3c, and indeed this observation is correct. We have
identified, and tried to describe and explain in greater depth this part of the model
output in this second draft of the manuscript (please see response to General Comment
1).
**Fig 4a: legend order confusing, figure isn't super useful.**
Thank you for taking the time to note this. We have removed the 'total organic carbon'
range from the original figure to streamline the number of sources used in this diagram.
On the other hand, the general relevance of this diagram is, we feel justified, for a
number of reasons. First, it lays out the modelled discharge of DOC over the 20th
Century, both annually and for an annualised 30 year running mean, to show that the
model outputs a long-term and unequivocal increase in DOC discharge from the Lena
river over the 20th Century. This is of interest since there are no DOC discharge
observations spanning this length of time, or, indeed the length of time necessary to
construct a long-term observational trendline. Secondly, on the same diagram we
include the average of the last ten years of simulated DOC discharge (horizontal lines)
and also mark empirical estimates of the same quantity from various empirical studies
within that approximate timeframe. The reasons for this are that (i) We can benchmark
the trendline mentioned against any potential systematic 'gap' in observed versus
simulated DOC discharge, which we show in the manuscript is indeed a systematic one
derived from errors in the hydrological module. This would imply that even if modelled absolute values are inaccurate relative to observations, the trendline might still reflect a
real tendency over the 20th Century. (ii) We discuss the difference in observational
estimates and, despite coming to the conclusion that the latest estimates are likely the
most accurate, include them all in the diagram to illustrate that the empirical numbers
are at the end of the day also estimates.

[Figure]

**Fig 4d: extra dotted lines are confusing**
Thank you for this observation.  We agree that the diagram is not necessarily the easiest
to follow, as is often the case for dual-axis figures, but it is our opinion that directly
comparing the modelled and observed DOC and CO2 seasonality is useful for
interpreting how and whether these two variables evolve and/or co-evolve over the
course of the year.  For this reason we don't feel that separating these two variables is in
the interest of the manuscript.
**Fig 5. Doesn't contain much new info, what does p-1 mean?**
We agree that this is perhaps not the most interesting facet of the model output, and
have moved the figure to the Supplement (Fig. S2). The $(p^{-1})$ is carried over from the
'p'eriod used as a temporal unit in Kutscher et al. (2017) from whom this figure is
directly derived.  The unit explained in the accompanying figure caption, by the
sentence: ***"Map adapted from Fig. 2 in Kutscher et al. (2017) showing proportional***
***sub-basin contributions of TOC outflow to total TOC discharge in June and July***
***(designated as their sampling period 'p$^{-1}$') of 2012-2013, as observed in Kutscher et***
***al., 2017 (black arrows)".***
**Fig 8: units of per pixel make not much sense.**

This has been changed to units of mgC m$^{-2}$d$^{-1}$, and to increase readability, we have removed one of the sub-figures (floodplains) from the diagram.

**Table S1: is it Tootchi et al., 2018 or Tootchi et al., 2019? Text says one, Table says another.**

Thank you for your diligence, this error has been corrected.

**Response to Reviewer 2 i**nteractive comment on "ORCHIDEE MICT-LEAK (r5459), a
global model for the production, transport and transformation of dissolved organic
carbon from Arctic permafrost regions, Part 2: Model evaluation over the Lena River
basin" by Simon P. K. Bowring et al.
Dear Anonymous Reviewer #2,
Thank you for your concise, informative and constructive assessment of this paper. In
what follows, we will respond first to your general comments, followed by specific
comments.
**Response to General Comments:**
General Comment 1
**In this manuscript, using the ORCHIDEE MICT-LEAK described by the first part in**
**accompanying paper, the authors assessed production, concentration, CO2**
**evasion, and riverine transport of dissolved organic carbon (DOC) over the Lena**
**River Basin. They conducted long-term simulations and made attempts to factor**
**out driving factors in DOC change in the study area. The research topic is**
**potentially interesting in terms of large-scale carbon budget, land-ocean linkage,**
**and carbon-climate interactions. For example, the long-term increase of DOC**
**discharge (e.g., Fig. 4a) looks intriguing, because this can affect biogeochemistry**
**in the Arctic Ocean. On the other hand, I have two major concerns on this**
**manuscript. First, the simulated results were compared only with several**
**literature data: e.g., Raymond et al. (2007), Kutscher et al. (2017), and Denfeld et**
**al. (2013). The comparisons were not adequately quantitative, and so I could not**
**figure out whether the model well captured observations. The low performance in**
**simulating river discharge may indicate that the model hydrology should be**
**improved before conducting DOC-related analyses.**
Thank you for your kind words. Here, we respond to your points sequentially.
We understand your concern regarding the relatively small number of studies referred
to for quantitative evaluation of model output. However, the fact remains that there are
very few observational studies specific to the Lena River basin whose sampling scale at
both spatial and temporal level are adequate for diagnosing output from a global-scale
land surface model. Indeed, this can be indirectly inferred from the fact that even
observed annual DOC discharge, which might in other world regions be considered a
relatively straightforward, first order diagnostic, carries estimates whose values differ
by a factor of over two. It is for this reason that we have, for this metric for example,
chosen to include empirical estimates from six different studies, if only to illustrate that
only one or two of these are likely to most closely approximate real-world DOC
discharge (e.g. Raymond et al. 2007, Holmes et al. 2012). Likewise, as we point out in
the manuscript, for some variables there simply do not exist observational estimates at any scale. This is true for example for river surface CO2 evasion from the Lena river, for which we had to resort to measurements from the Kolyma river for evaluation, or groundwater-sourced hydrological discharge. We reiterate that many studies that have been carried out over the Lena basin have an inadequate spatial or temporal sampling resolution for our scale of evaluation, which is that of the basin. In this sense, we are of the opinion that we have largely covered the spectrum of the relevant and appropriate observational literature, summarised in the new Supplementary table below, in evaluating ORCHIDEE M-L. In addition, we have added one more evaluation source for CO2 evasion from the Ob river in Western Siberia, for comparison, as is now included in the following text:

***"Likewise, mean annual evasion rates of <0.8 up to around 7 gC m$^{-2}$ d$^{-1}$ have been found for the Ob and Pur rivers in Western Siberia (Serikova et al., 2018)."***

| | Empirical Evaluation Sources |
|---|---|
| **DOC Discharge** | Cauwet and Sidorov (1996); Dolman et al. (2012); Holmes et al. (2012); Lara et al. (1998); Raymond et al. (2007); Semiletov et al. (2011); Kutscher et al. (2017). |
| **Water Discharge** | Ye et al. (2009); Lammers et al. (2001) |
| **DOC concentration** | Shvartsev (2008); Denfeld et al. (2013); Mann et al. (2015); Raymond et al. (2007); Semiletov et al. (2011); Arctic-GRO/PARTNERS (Holmes et al., 2012) |
| **NPP** | Beer et al. (2006); Lloyd et al. (2002); Roser et al. (2002); Schulze et al. (1999); Shvidenko and Nilsson, (2003) |
| **Soil Respiration** | Elberling (2007); Sawamoto et al. (2000); Sommerkorn (2008). |
| **CO2 Evasion** | Denfeld et al. (2013); Serikova et al. (2018). |

**Table S2:** Literature sources for empirical evaluation of model output.

Thank you for pointing out what is clearly an issue with interpreting model results at face value. ***Note to the Editor: The remainder of this response to General Comment 1 can be found in the Response to Reviewer #1, and is repeated verbatim in the following paragraphs.*** In the first part of this two-part paper, we showed that this model development version of ORCHIDEE, ORCHIDEE MICT-LEAK, was devoted to the development and inclusion of a permafrost-specific DOC and dissolved CO2 generation, transport and evasion module to the high-latitude version (ORCHIDEE-MICT) of the land surface model ORCHIDEE. The hydrology scheme in this model version is almost entirely unchanged from that latter version (ORCHIDEE-MICT), which is itself one of two parent model versions leading to this particular model instantiation. ORCHIDEE-MICT has itself already been subjected to a lengthy evaluation paper at the Pan-Arctic scale in Geoscientific Model Development (Guimberteau et al., 2018). Indeed, the sole substantial addition to the hydrology module in this model version (ORCHIDEE MICT-LEAK) is that floodplain inundation is now represented, with some significant but not cumulatively substantial impacts on water discharge. Otherwise, this work is focused only on the production, metabolisation and transport of DOC from plant and soil matter in the Arctic. In this particular sense, our improvement of the model does not directly involve the representation of the hydrological cycle, which is itself dependent on how the surface energy balance, vegetative uptake and soil flow dynamics and, perhaps most
importantly, the specific set of climatological data used as model input, are represented
and read by the model, respectively.   However, model output is clearly strongly
impacted by these factors, both individually and in sum, and we agree that stronger
quantification and explanation of the inadequacy of the hydrological module, and its
effects on the DOC-generating module's results, is necessary.   We also feel that a more
hydrology-independent metric for model evaluation should have been used.   These we
address in what follows by providing further identificaiton for the factors causing low
hydrological discharge, quantification of the dependency of DOC discharge error on
water discharge error (DOC error (%) dependent on hydrological error), followed by
summary of the remaining possible causes of error, including substantial error
introduced by choice of climatological forcing dataset and, finally, introduce a new
evaluation metric to evaluate DOC representation in the model that is quasi- but not
fully- independent, from modelled hydrological discharge.

First, we more concretely address the causes of the model-observation mismatch in
hydrological discharge, by adding the following new text to the manuscript:

[revised manuscript text omitted]

Finally, we evaluate the seasonal DOC discharge in terms of DOC concentration, which was not done in the first draft of this manuscript. The reasoning for this is that DOC concentrations are less dependent on hydrological discharge than bulk DOC fluxes, and thus offer a clearer means by which to evaluate the DOC module as a standalone product. This evaluation shows that indeed, DOC concentrations are reasonably well represented compared to observations for the majority of the year's **bulk** DOC discharge, but underestimates concentrations during wintertime. The latter deficiency is consistent with the observation from Guimberteau et al. (2018) that the model poorly simulates wintertime subsurface water flow in the soil, which, by exaggerating the soil vertical impermeability of permafrost, greatly reduces the amount of DOC leachate that can be transferred to the (warmer) subsoil and laterally transferred into the river. Thus we add the following subsection to the manuscript:

*"While total DOC discharge captures the integral of processes leading to fluvial biogeochemical outflow, simulations of this are highly sensitive to the performance of modelled hydrology and climatological input data. A more precise measure for the performance of the newly-introduced DOC production and transport module, that is less sensitive to reproduction of river water discharge, is DOC concentration. This is because while the total amount of DOC entering river water depends on the amount of water available as a vehicle for this flux (hydrology), the concentration of DOC depends on the rate of soil carbon leaching, itself depending largely on the interaction of soil biogeochemistry with primary production and climatic factors. This we evaluate in Figure 5a, This shows that for the majority of the thaw period or growing season (April-September), which corresponds to the period during which over 90% of DOC production and transport occurs, the model largely tracks the observed seasonality of DOC concentrations in Arctic-GRO data averaged over 1999-2007. There is a large overestimate of the DOC concentration in May owing to inaccuracies in simulating the onset of the thaw period, while the months June-September underestimate concentrations by an average of 18%. On the other hand, frozen period (November-April) DOC concentrations are underestimated by between ~30-500%. This is due to deficiencies in representing wintertime soil hydrological water flow in the model, which impedes water flow when the soil is frozen, as discussed in Section 4.2.1. Because of this deficiency, slow-moving groundwater flows that contain large amounts of DOC leachate are under-represented. This interpretation is supported by the fact that in both observations and simulations, at low discharge rates (corresponding to wintertime), DOC concentrations exhibit a strong positive correlation with river discharge, while this relationship becomes insignificant at higher levels of river discharge (Fig. 5b). Thus wintertime DOC concentrations suffer from the same deficiencies in model representation as those for water discharge. In other words, the standalone representation of DOC leaching*

*is satisfactory, while when it is sensitive to river discharge, it suffers from the same*
*shortfalls identified in Section 4.2.1 and 4.2.2."*
The accompanying figures to this text are shown below:

[Figure]

*Figure 5: (a) Simulated and observed (Arctic-GRO/Holmes et al., 2012) DOC*
*concentration seasonality for the Lena basin over the period 1999-2007. (b) Plots of*
*DOC concentration versus river discharge as in observations (Raymond et al., 2007)*
*and simulations, where simulations data points are monthly averages taken over*
*the period 1999-2007*
General Comment 2

**738** **Second, the model simulations were conducted at a spatial resolution of 1 degree**
**739** **(about 100 km), but it looks too coarse to capture the spatial heterogeneity in this**
**740** **region. As the authors stated (Line 535), the model could not include small**
**741** **streams because of the coarse-scale river-routing scheme.**
**742**
**743** Thank you for noting this poorly explained portion of the original text, which has also
**744** been rewritten for Part 1 of this study.  The smaller order streams of Strahler orders 1-3
**745** are actually implicitly represented, although their surface area is not calculated by the
**746** model.  To be clear, this is the overland flow of water calculated at the sub-grid scale,
**747** such that the movement from one quadrant of a grid cell to another quadrant of that
**748** same cell is represented by the 'fast' (or 'stream' as referred to in the manuscript)
**749** hydrological pool, which is then aggregated to the whole grid cell. We explain this in the
**750** following additional text:
**751**
**752** *"As noted in Part 1 of this study, although the model as a whole conducts*
**753** *simulations at the 1 degree scale, the routing of water and carbon, as well as the*
**754** *evasion of the latter, occurs at the sub-grid scale, such that we are able to simulate*
**755** *spatially explicit rivers whose size approximates Strahler order 4, and through the*
**756** *'fast' water pool in the model are able to simulate streams of Strahler order 1-3. "*
**757**
**758**
**759** General Comment 3
**760**
**761** **The manuscript provides numerous figures and text is a bit lengthy. In contrast,**
**762** **Simulation Rationale and Setup sections are brief and I felt inadequate. Data for**
**763** **comparison were described in Results and Discussion sections (e.g., Line 365–368,**
**764** **Line 615–621). I recommend moving these data descriptions to a section in**
**765** **Methods and Data. Therefore, the manuscript can be largely truncated and should**
**766** **be thoroughly reorganized.**
**767**
**768** We agree that the manuscript lacked some concision and could have been shortened.  On
**769** the other hand, both reviewers asked for some additional material to be added into the
**770** introduction, evaluation and interpretation segments of the manuscript.  As such, the
**771** manuscript has been entirely edited to account for these shortcomings.  In doing so, we
**772** have focussed on text readability, reducing repetition and simplifying the nature of the
**773** text itself, substantially reducing the length of the original text body. The number of
**774** textual changes are too numerous and in some cases too lengthy to enumerate
**775** piecemeal here, so we ask that you refer to the 'track-changes' version of the new
**776** manuscript draft to evaluate these changes directly.  In addition, we have moved one
**777** entire subsection (Evaluation of NPP and Soil Respiration) from the main body to the
**778** Supplement (Text S2), given that this has already been evaluated, albeit at a larger scale,
**779** in Guimberteau et al. (2018) and given that its evaluation here detracts somewhat from
**780** the central foci of our manuscript.  Figure 5 has now been moved to the Supplement
**781** (now Fig. S2), while Fig. 8 has been truncated by removing one of the evasion map suites
**782** (floodplains) to increase the size and readability of the overall image.
**783**
**784** The observational data compared, as addressed already in our Response to General
**785** Comment 1, is now summarised in Table S2 of the Supplement. In addition, we have
**786** substantially expanded segments of the Introduction/Methods/Data sections, to provide greater description and context to model functioning and the input data used. We have
also included a Figure directly drawn from Part 1 of this study (the model's carbon
module schematic), to provide greater understanding to the reader for how the model
functions (See Fig. S1) Descriptive changes in this vein are summarised in the following
additional texts:
*"In essence, photosynthetically fixed plant carbon is transformed by microbial*
*degradation to DOC and CO2; the DOC is itself either respired to CO2 or adsorbed, or*
*exchanged with particulate soil carbon. DOC can then be transferred by*
*precipitation-dependent water flow laterally across the terrestrial landmass, in*
*surface or subsurface flows to streams and rivers, whereupon it may either be*
*respired within the water column or exported to the marine realm. A flow diagram*
*depicting these flows and the residence times of the respective carbon pools,*
*reproduced from Part 1 of this study, is given in Figure S1a,b."*
*"Climatological forcing is input from the Global Soil Wetness Project  Phase 3*
*(GSWP3) v.0 data, based on 20th Century reanalysis using the NCEP land-*
*atmosphere model and downscaled to a 0.5°, 3-hourly resolution covering the*
*period 1901 to 2007 (Supplement, Table S1).  This is then upscaled to 1° resolution*
*and interpolated to a 30 minute timestep to comply with the timestep of ORCHIDEE's*
*surface water and energy balance calculation period. Precipitation was partitioned*
*into rainfall and snowfall, and a correction for wind-induced undercatch  was*
*applied separately.  These are described in greater detail in Guimberteau et al.*
*(2018) Over the simulation period under this climatological forcing dataset, the*
*Lena basin experiences a mean thaw period warming of 1.8°C, while atmospheric*
*$CO_2$ concentrations increase by 85.6ppm.  The GSWP3 dataset was chosen for its*
*prior suitability as input its relative performance in simulating the interannual*
*variability and seasonality of Pan-Arctic riverine discharge in ORCHIDEE-MICT*
*(Guimberteau et al., 2018), as compared to another data-driven climate forcing*
*product, CRUNCEP v7 (Kalnay et al., 1996; New et al., 1999)..  Indeed, under*
*CRUNCEP v7, ORCHIDEE-MICT was shown to underestimate river discharge by as*
*much as 83% over the Yukon basin.  An improved floodplains area input file for the*
*Lena basin (Tootchi et al., 2019) was used to drive the simulation of floodplain*
*dynamics (Supplement, Table S1).  The model structure is described in Part 1 of this*
*study, however we describe how the fluxes are generated with respect to the results*
*obtained by this study in some detail in the initial description of the results, below*
*(Section 4.1). "*
*"Simulations were run over the Lena river basin  (Fig. 3a) for the climate, $CO_2$ and*
*vegetation input forcing data (Supplement, Table S1) over 1901-2007 at a 1 degree*
*resolution (Fig. 1), to evaluate the simulated output of relevant carbon fluxes and*
*hydrologic variables against their observed values, as well as those of emergent*
*phenomena arising from their interplay (Fig. 1). We evaluate at the basin scale*
*because the isolation of a single geographic unit allows for a more refined analysis*
*of simulated variables than doing the same over the global Pan-Arctic, much of*
*which remains poorly accounted for in empirical databases and literature.   The*
*literature studies used in this evaluation are summarised in Table S2. "*

**Response to Specific Comments:**

**Specific Comments:**

Specific Comment 1:

**Line 45–46: In main text, no part discussed about '1.8_C warming' and '+85.6 ppm CO2 rise'. Why did you mention these values in Abstract?**

Indeed, thank you for spotting this omission. These have now been included in the main text body ('Simulation Rationale') with the following text:

*"Over the simulation period under this climatological forcing dataset, the Lena basin experiences a mean thaw period warming of 1.8°C, while atmospheric $CO_2$ concentrations increase by 85.6ppm."*

Specific Comment 2:

**Line 81: Did you examine the accuracy of GSWP3 in the study region? Especially for precipitation, some climate datasets may have serious biases.**

When using historical data -generated climatological datasets (as opposed to those generated by climate models), it has been shown by Guimberteau et al. (2018) that for the Pan-Arctic in general and for the Lena in particular, GSWP-3 already performs substantially better than the CRU-NCEP dataset (a widely used climatological data suite) with respect to timing and magnitude of simulated hydrological discharge. Our own decadal-scale preliminary test runs using the 'Princeton' (PGMF) dataset comes to the same conclusion, that GSWP3 results in comparatively better simulated river discharge. Thus there may indeed be some precipitation bias in the input datasets. As noted in the response to General Comment 2, we have also compared the modelled hydrographs when using GSWP3 and ISIMIP2b (see Table S2), which gives a substantial rise in both river and DOC discharge in the latter compared to GSWP3. We did not choose to run with the ISIMIP dataset because it is itself model-generated, while for the land surface model as a whole, we feel it is preferable to make use of the existing historically-generated data.

Specific Comment 3:

**Linen 580: Remove (g C m–2 d–1).**

This has now been removed.

Specific Comment 4:

**Line 787: Why did you discuss about NPP and soil respiration of Siberian forests in this position of the manuscript? I lost context here.**

Thank you for spotting this inconsistency. This has been moved to section 4.2.2 (lines
539-550) as part of the interpretation of DOC discharge dependence on NPP.
Specific Comment 5:
**Line 869: As long as I know, a version of ORCHIDEE (e.g., Naipal et al., 2018,**
**Biogeosciences, 15, 4459–4480) includes POC erosion module.**
It is correct that Naipal et al. (2018) introduced an erosion emulator to the default
version of ORCHIDEE. However, as is the case with many such model developments that
are made roughly simultaneously, the erosion module is not yet compatible with the
high latitude version of ORCHIDEE, and thus the DOC module here is neither compatible
with the erosion module. Of course in principle this should be addressed immediately
for a more 'complete' model, however in practice such code merges are extremely time-
consuming and thus beyond the temporal scope of this evaluation paper.
Specific Comment 6:
**Line 924: The ratio of DOC export relative to NPP, _1.5%, would be an important**
**result but does not appears in Abstract**
These have now been included in the Abstract with the following text:

[revised manuscript text omitted]

Simon Bowring 18/7/y 12:54

Simon Bowring 27/6/y 09:24

Simon Bowring 27/6/y 09:34

Simon Bowring 18/7/y 12:54

Simon Bowring 27/6/y 09:36

Simon Bowring 18/7/y 12:54

Simon Bowring 27/6/y 08:05

Simon Bowring 18/7/y 12:54

Simon Bowring 27/6/y 08:05

Simon Bowring 27/6/y 08:05

Simon Bowring 27/6/y 08:06

Simon Bowring 10/6/y 17:03

Lauerwald, Ronny 30/7/y 18:11

Simon Bowring 10/6/y 17:04

Lauerwald, Ronny 30/7/y 18:11

Simon Bowring 27/6/y 08:03

Simon Bowring 10/6/y 17:05

[revised manuscript text omitted]

Simon Bowring 14/7/y 15:05

Simon Bowring 18/7/y 12:54

Simon Bowring 14/7/y 15:05

Simon Bowring 7/6/y 16:45

Simon Bowring 14/7/y 15:09

Simon Bowring 14/7/y 15:09

Simon Bowring 14/7/y 15:09

Simon Bowring 18/7/y 12:54

Simon Bowring 14/7/y 14:48

Simon Bowring 14/7/y 15:09

Simon Bowring 18/7/y 12:54

Simon Bowring 18/7/y 12:54

summarised in Figs. 12(b-c), in which we show the same 1998-2007 –averaged yearly
variable fluxes as in the CTRL simulation, expressed as percentages of the CTRL values
given in Fig. 2. A number of conclusions can be drawn from these diagrams.
First, all fluxes are lower in the factorial simulations, which can be expected due to
lower carbon input to vegetation from the atmosphere (constant $CO_2$) and colder
temperatures (constant climate) inhibiting more vigorous growth and carbon cycling.
Second, broadly speaking, both climate and $CO_2$ appear to have similar effects on all
fluxes, at least within the range of climatic and $CO_2$ values to which they have subjected
the model in these historical runs. With regard to lateral export fluxes in isolation,
variable climate (temperature increase) is a more powerful driver than $CO_2$ increase
(see below). Third, the greatest difference between the constant climate and $CO_2$
simulation carbon fluxes appear to be those associated with terrestrial inflow of
dissolved matter to the aquatic network, these being more sensitive to climatic than $CO_2$
variability. This is evidenced by a 49% and 32% decline in $CO_2$ and DOC export,
respectively, from the land to rivers in the constant climate simulation, versus a 27%
and 23% decline in these same variables in the constant $CO_2$ simulation. Given that the
decline in primary production and respiration in both factorial simulations was roughly
the same, this difference in terrestrial dissolved input is attributable to the effect of
climate (increased temperatures) on the hydrological cycle, driving changes in lateral
export fluxes.
This would imply that at these carbon dioxide and climatic ranges, the modelled DOC
inputs are slightly more sensitive to changes in the climate rather than to changes in
atmospheric carbon dioxide concentration and the first order biospheric response to
this. However, while the model biospheric response to carbon dioxide concentration
may be linear, thresholds in environmental variables such as MAAT may prove to be
tipping points in the system's emergent response to change, as implied by Fig. 9,
meaning that the Lena, as with the Arctic in general, may soon become much more
temperature-dominated with regard to the drivers of its own change.
**5.1.3 LOAC export flux considerations**
Despite our simulations' agreement with observations regarding the proportional fate of
terrestrial DOC inputs as evasion and marine export (Fig. 12a), our results suggest
substantial and meaningful differences in the magnitude of those fluxes relative to NPP
in the Lena, compared to those estimated by other studies in temperate or tropical
biomes. Our simulations' cumulative DOC and $CO_2$ export from the terrestrial realm into
inland waters is equivalent to ~1.5 % of NPP.
This is considerably lower than Cole et al. (2007) and Regnier et al. (2013) who find
lateral transfer to approximate ~5% (1.9PgC yr$^{-1}$) of NPP at the global scale, while
Lauerwald et al. (2017) found similar rates for the Amazon. The cause of this
discrepancy with our results is beyond the scope of this study to definitively address,
given the lack of tracers for carbon source and age in our model. Nonetheless, our
analysis leads us to hypothesise the following.
Temperature limitation of soil microbial respiration at the end of the growing season
(approaching zero by October, SI Fig. S5d) makes this flux neglible from November

Simon Bowring 14/7/y 14:56

Simon Bowring 14/7/y 15:09

Simon Bowring 18/7/y 12:54

Simon Bowring 14/7/y 14:56

Simon Bowring 18/7/y 12:54

Simon Bowring 18/7/y 12:54

Simon Bowring 18/7/y 12:45

Simon Bowring 18/7/y 12:54

through May (SI Fig. S5d). In late spring, mobilisation of organic carbon is performed by
both microbial respiration and leaching of DOC via runoff and drainage water fluxes.
However, because the latter are controlled by the initial spring meltwater flux period,
which occurs before the growing season has had time to produce litter or new soil
carbon (May-June, Fig. 4b), aggregate yearly DOC transport reactivity is characterised by
the available plant matter from the previous year, which is overwhelmingly derived
from recalcitrant soil matter (Fig. 11a) and is itself less available for leaching based on
soil carbon residence times.

This causes relatively low leaching rates and riverine DOC concentrations (e.g. Fig. 9), as
compared to the case of leaching from the same year's biological production.
Highlighting this point is floodplain domination by labile carbon sourced from that
year's production with a mean DOC concentration of 12.4 mgC L$^{-1}$ (1998-2007 average),
with mean riverine DOC concentrations around half that value (6.9 mgC L$^{-1}$).
Nonetheless the May-June meltwater pulse period dominates aggregate DOC discharge.
As this pulse rapidly subsides by late July, so does the leaching and transport of organic
matter. Warmer temperatures come in conjunction with increased primary production
and the temperature driven soil heterotrophic degradation of contemporary and older
matter (via active layer deepening). These all indicate that transported dissolved matter
in rivers, at least at peak outflow, is dominated by sources originating in the previous
year's primary production, that was literally 'frozen out' of more complete
decomposition by soil heterotrophs.

Further, we infer from the fact that all of our simulation grid cells fall within areas of low
(<-2°C) MAAT, far below the threshold MAAT (>3°C) proposed by Laudon et al. (2012)
for soil respiration-dominated carbon cycling systems (Fig. 9), that the Lena is
hydrologically-limited with respect to DOC concentration and its lateral flux. Indeed, the
seasonal discharge trend of the Lena –massive snowmelt-driven hydrological and
absolute DOC flux, coupled with relatively low DOC concentrations at the river mouth
(Fig. 4b, simulation data of Fig. 9), are in line with the Laudon et al. (2012) typology.

We therefore suggest that relatively low lateral transport relative to primary production
rates (e.g. as a percentage of net primary production, (%NPP)) in our simulations
compared to the lateral transport : NPP percentages reported from the literature in
other biomes is driven by meltwater (vs. precipitation) dominated DOC mobilisation,
which occurs during a largely pre-litter deposition period of the growing season. DOC is
then less readily mobilised by being sourced from recalcitrant matter, leading to low
leaching concentrations relative to those from labile material. As discharge rates
decline, the growing season reaches its peak, leaving carbon mobilisation of fresh
organic matter to be overwhelmingly driven by in situ heterotrophic respiration.

While we have shown that bulk DOC fluxes scale linearly to bulk discharge flows (Fig.
3d), DOC concentrations (mgC L$^{-1}$) hold a more complex and weaker positive
relationship with discharge rates, with correlation coefficients ($R^2$) of 0.05 and 0.25 for
river and stream DOC concentrations, respectively (Fig. 13). This implies that while
increasing discharge reflects increasing runoff and an increasing vector for DOC
leaching, particularly in smaller tributary streams, by the time this higher input of
carbon reaches the river main stem there is a confounding effect of dilution by increased
water fluxes which reduces DOC concentrations, explaining the difference between

Simon Bowring 18/7/y 12:45

Simon Bowring 18/7/y 12:54

Simon Bowring 18/7/y 12:54

Simon Bowring 18/7/y 12:54

Simon Bowring 14/7/y 15:07

Simon Bowring 18/7/y 12:54

Simon Bowring 14/7/y 15:07

Simon Bowring 18/7/y 12:54

Simon Bowring 14/7/y 15:07

Simon Bowring 14/7/y 15:00

Simon Bowring 18/7/y 12:54

Simon Bowring 18/7/y 12:54

Simon Bowring 14/7/y 15:00

Simon Bowring 18/7/y 12:54

Simon Bowring 18/7/y 12:54

stream and river discharge vs. DOC concentration regressions in the Figure. Thus, and as a broad generalisation, with increasing discharge rates we can also expect somewhat higher concentrations of terrestrial DOC input to streams and rivers. Over the floodplains, DOC concentrations hold no linear relationship with discharge rates ($R^2$=0.003 , SI Fig. S6), largely reflecting the fact that DOC leaching is here limited by terrestrial primary production rates more than by hydrology. To the extent that floodplains fundamentally require flooding and hence do depend on floodwater inputs at a primary level, we hypothesise that DOC leaching rates are not limited by that water input, at least over the simulated Lena basin.

As discussed above simulated DOC and $CO_2$ export as a percentage of simulated NPP over the Lena basin was 1.5% over 1998-2007. However, this proportion appears to be highly dynamic at the decadal timescale. As shown in Fig. S7, all lateral flux components in our simulations increased their relative throughput at a rate double to triple that of NPP or respiration fluxes over the 20[th] century, also doing so at a rate substantially higher than the rate increase in discharge. In addition, differentials of these lateral flux rates with the rates of their drivers (discharge, primary production) have on average increased over the century (Fig. S7). This suggests that there are potential additive effects of the production and discharge drivers of lateral fluxes that could lead to non-linear reponses to changes in these drivers as the Arctic environment transforms, as suggested by the Laudon et al. (2012) data plotted in Fig. 4. Acceleration of the hydrological cycle compounded by temperature and $CO_2$ -driven increases in primary production could therefore increase the amount of matter available for leaching, increase the carbon concentration of leachate, and increase the aggregate generation of runoff to be used as a DOC transport vector. Given that these causal dynamics apply generally to permafrost regions, both low lateral flux as %NPP and the hypothesised response of those fluxes to future warming may be a feature particular to most high latitude river basins.

**6. Conclusion**

This study has shown that the new DOC-representing high latitude model version of ORCHIDEE, ORCHIDEE MICT-LEAK, is able to reproduce with reasonable accuracy modern concentrations, rates and absolute fluxes of carbon in dissolved form, as well as the relative seasonality of these quantities through the year. When combined with a reasonable reproduction of real-world stream, river and floodplain dynamics, we demonstrate that this model is a potentially powerful new tool for diagnosing and reproducing past, present and potentially future states of the Arctic carbon cycle. Our simulations show that of the 34 TgC yr$^{-1}$ remaining after GPP is respired autotrophically and heterotrophically in the Lena basin, over one-fifth of this captured carbon is removed into the aquatic system. Of this, over half is released to the atmosphere from the river surface during its period of transport to the ocean, in agreement with previous empirically-derived global-scale studies. Both this transport and its transformation are therefore non-trivial components of the carbon system at these latitudes that we have shown are sensitive to changes in temperature, precipitation and atmospheric $CO_2$ concentration. Our results, in combination with empirical data, further suggest that changes to these drivers –in particular climate –may provoke non-linear responses in the transport and transformation of carbon across the terrestrial-aquatic system's

Simon Bowring 14/7/y 15:00

Simon Bowring 18/7/y 12:46

Simon Bowring 18/7/y 12:54

Simon Bowring 18/7/y 12:40

Simon Bowring 18/7/y 12:54

Simon Bowring 18/7/y 12:40

Simon Bowring 18/7/y 12:54

Simon Bowring 18/7/y 12:54

Simon Bowring 14/7/y 15:08

Simon Bowring 18/7/y 12:54

[revised manuscript text omitted]

Simon Bowring 9/6/y 11:46

[Figure]

Simon Bowring 18/7/y 12:54

[Figure]

Simon Bowring 18/7/y 12:54

Simon Bowring 9/6/y 11:49

Simon Bowring 18/7/y 12:54

[Figure]

[Figure]

[Figure]

**Figure 3**: Map of the Lena **(a)** with the scale bar showing the mean grid cell topographic slope from the simulation, and the black line the satellite-derived overlay of the river main stem and sub-basins. Mountain ranges of the Lena basin are shown in orange. Green circles denote the outflow gridcell (Kusur) from which our simulation outflow data are derived, as well as the Zhigansk site, from which out evaluation against data from Raymond et al. (2007) are assessed. The regional capital (Yakutsk) is also included for geographic reference. Coastal outline and inland water bodies are shown as dashed red and solid black lines, respectively. **(b)** Maps of river water discharge ($\log(m^3\,s^{-1})$) in April, June and September, averaged over 1998-2007. **(c)** The mean monthly river discharge differential between observed discharge for the Lena (Ye et al., 2009) and simulated discharge averaged over 1998-2007, in absolute ($m^3\,s^{-1}$) and percentage terms. **(d)** Regression of simulated monthly DOC discharge versus simulated river discharge at the river mouth (Kusur) over the entire simulation period (1901-2007). **(e)** Summed yearly lateral flux versus NPP values for DOC discharge, $CO_2$ discharge and $CO_2$ evasion ($FCO_2$) over the entire simulation period, with linear regression lines shown.

**(a)**

[Figure]

**(b)**

Simon Bowring 18/7/y 12:54

Simon Bowring 18/7/y 12:54

**(c)**

[Figure]

**(d)**

[Figure]

Simon Bowring 18/7/y 12:54

Simon Bowring 10/6/y 17:26
Simon Bowring 18/7/y 12:54
Simon Bowring 18/7/y 12:54

[revised manuscript text omitted]

---

## Author Response (AR2)

**Response to Major Revision Comment by Topical Editor**

**The length of the text is way too much, and it was quite painful to read throughout the document. This problem became much worse after the revision, although previous reviewer pointed out this issue. Of cause, authors have reason to add new texts for addressing reviewer's comments as authors mentioned it in their reply letter. But, still, due to the absence of readability, it is indeed very painful to read through the document. Actually, both previous referees refused to review the revised manuscript, and this is the first case in my long experience as an editor.**

**I agree that authors addressed all issues raised by previous referees. It's OK. But, the manuscript should be re-organized to reduce the amount of main text substantially (like, less than half). I believe that majority of the text in the section 4 and some figures can be moved to the supplemental information.**

Thank you for taking the time to go through this iteration of this manuscript. We understand the issues that you and the referees have raised, and have accordingly reduced the main text body from 21 to 11 pages, and its total length (including Figures, etc) from 45 to 27 pages. These reductions have entailed a mix of deletion and the movement of large sections of the text and Figures into the Supplement, focusing largely on Section 4, as you have helpfully pointed out. On the other hand we were somewhat perplexed by the reported refusal to referee the previous version of the manuscript. All prior specific and general comments by referees were responded to, and indeed we had substantially reduced and restructured the text as per a non-specific suggestion to do so, while as a native English speaker the first author finds the charge of 'unreadability' rather puzzling. Since this manuscript has been in review for over seven months, we were surprised to learn that we should cut the text body by a factor of two or more at this late stage, given that this was never specifically raised in such a context previously. Again, we understand and agree that the length of the document posed problems of overburden for the reader in general and the referees specifically, however we felt this necessarily reflected the very broad content of such a study, given that it appeals to both the modelling and field work communities, and would need to satisfy their rigor and interest across a large number of domains (primary production, hydrology, data, soil science and thermodynamics, and the rather more niche specificities of the inland water continuum -DOC discharge, concentration, evasion, etc.) within the context of the model's structure and function, or lack thereof. We look forward to moving on to the next iteration of this manuscript's evolution, and thank you for your patience with and participation in that process.

[revised manuscript text omitted]
| Spinup Soil Carbon Stock | 20ky ORCHIDEE-MICT soil carbon spinup | (Guimberteau et al., 2018) |

**Table S2:** Literature sources for empirical evaluation of model output.

| | Empirical Evaluation Sources |
|---|---|
| **DOC Discharge** | Cauwet and Sidorov (1996); Dolman et al. (2012); Holmes et al. (2012); Lara et al. (1998); Raymond et al. (2007); Semiletov et al. (2011); Kutscher et al. (2017). |
| **Water Discharge** | Ye et al. (2009); Lammers et al. (2001) |
| **DOC concentration** | Shvartsev (2008); Denfeld et al. (2013); Mann et al. (2015); Raymond et al. (2007); Semiletov et al. (2011); Arctic-GRO/PARTNERS (Holmes et al., 2012) |
| **NPP** | Beer et al. (2006); Lloyd et al. (2002); Roser et al. (2002); Schulze et al. (1999); Shvidenko and Nilsson, (2003) |
| **Soil Respiration** | Elberling (2007); Sawamoto et al. (2000); Sommerkorn (2008). |
| **CO2 Evasion** | Denfeld et al. (2013); Serikova et al. (2018). |

**Table S3**: Observed versus simulated DOC discharge (1998-2007), where we compare the output of two separate climatological datasets used as input to the model (GSWP3 and ISIMIP 2b). Also shown are the simulated versus observed DOC discharge for the six largest Arctic rivers (the "Big Six") and for the Pan-Arctic as a whole.

Simon Bowring 8/10/y 16:34

Simon Bowring 8/10/y 16:53

[revised manuscript text omitted]

Andersson, P. S.: Spatial variation in concentration and sources of organic carbon in the
Lena River, Siberia, J. Geophys. Res. Biogeosciences, 122(8), 1999–2016,
doi:10.1002/2017JG003858, 2017.
Lammers, R. B., Shiklomanov, A. I., Vörösmarty, C. J., Fekete, B. M. and Peterson, B. J.:
Assessment of contemporary Arctic river runoff based on observational discharge
records, J. Geophys. Res. Atmos., doi:10.1029/2000JD900444, 2001.
Lange, S.: EartH2Observe, WFDEI and ERA-Interim data Merged and Bias-corrected for
ISIMIP (EWEMBI), GFZ Data Serv., doi:10.5880/pik.2016.004, 2016.
Lange, S.: Bias correction of surface downwelling longwave and shortwave radiation for
the EWEMBI dataset, Earth Syst. Dyn., doi:10.5194/esd-9-627-2018, 2018.
Laudon, H., Buttle, J., Carey, S. K., McDonnell, J., McGuire, K., Seibert, J., Shanley, J.,
Soulsby, C. and Tetzlaff, D.: Cross-regional prediction of long-term trajectory of stream
water DOC response to climate change, Geophys. Res. Lett., doi:10.1029/2012GL053033,
2012.
Lauerwald, R., Hartmann, J., Ludwig, W. and Moosdorf, N.: Assessing the nonconservative
fluvial fluxes of dissolved organic carbon in North America, J. Geophys. Res.
Biogeosciences, doi:10.1029/2011JG001820, 2012.
Lauerwald, R., Laruelle, G. G., Hartmann, J., Ciais, P. and Regnier, P. A. G.: Spatial patterns
in CO2evasion from the global river network, Global Biogeochem. Cycles,
doi:10.1002/2014GB004941, 2015.
Lloyd, J., Shibistova, O., Zolotoukhine, D., Kolle, O., Arneth, A., Wirth, C., Styles, J. M.,
Tchebakova, N. M. and Schulze, E. D.: Seasonal and annual variations in the
photosynthetic productivity and carbon balance of a central Siberian pine forest, Tellus,
Ser. B Chem. Phys. Meteorol., doi:10.1034/j.1600-0889.2002.01487.x, 2002.
Nachtergaele, Freddy, Harrij van Velthuizen, Luc Verelst, N. H. Batjes, Koos Dijkshoorn,
V. W. P. van Engelen, Guenther Fischer, Arwyn Jones, Luca Montanarella, Monica Petri,
Sylvia PrielerB, Xuezheng Shi, Edmar Teixera and David Wiberg: The harmonized world
soil database, Proc. 19th World Congr. Soil Sci. Soil Solut. a Chang. World, Brisbane, Aust.
1-6 August 2010, pp. 34-37., 34–37 [online] Available from:
https://library.wur.nl/WebQuery/wurpubs/fulltext/154132, 2010.
Qiu, C., Zhu, D., Ciais, P., Guenet, B., Krinner, G., Peng, S., Aurela, M., Bernhofer, C.,
Brümmer, C., Bret-Harte, S., Chu, H., Chen, J., Desai, A. R., Dušek, J., Euskirchen, E. S.,
Fortuniak, K., Flanagan, L. B., Friborg, T., Grygoruk, M., Gogo, S., Grünwald, T., Hansen, B.
U., Holl, D., Humphreys, E., Hurkuck, M., Kiely, G., Klatt, J., Kutzbach, L., Largeron, C.,
Laggoun-Défarge, F., Lund, M., Lafleur, P. M., Li, X., Mammarella, I., Merbold, L., Nilsson,
M. B., Olejnik, J., Ottosson-Löfvenius, M., Oechel, W., Parmentier, F. J. W., Peichl, M., Pirk,
N., Peltola, O., Pawlak, W., Rasse, D., Rinne, J., Shaver, G., Peter Schmid, H., Sottocornola,
M., Steinbrecher, R., Sachs, T., Urbaniak, M., Zona, D. and Ziemblinska, K.: ORCHIDEE-
PEAT (revision 4596), a model for northern peatland CO2, water, and energy fluxes on
daily to annual scales, Geosci. Model Dev., 11(2), 497–497, doi:10.5194/gmd-11-497-
2018, 2018.
Rawlins, M. A., Fahnestock, M., Frolking, S. and Vörösmarty, C. J.: On the evaluation of
snow water equivalent estimates over the terrestrial Arctic drainage basin, in
Hydrological Processes., 2007.
Reynolds, C. A., Jackson, T. J. and Rawls, W. J.: Estimating soil water-holding capacities by
linking the Food and Agriculture Organization soil map of the world with global pedon
databases and continuous pedotransfer functions, Water Resour. Res., 36(12), 3653–
3662, doi:10.1029/2000WR900130, 1999.
Roser, C., Montagnani, L., Schulze, E.-D., Mollicone, D., Kolle, O., Meroni, M., Papale, D.,

Marchesini, L. B., Federici, S. and Valentini, R.: Net CO2 exchange rates in three different successional stages of the "Dark Taiga" of central Siberia, Tellus B, doi:10.1034/j.1600-0889.2002.01351.x, 2002.

Sawamoto, T., Hatano, R., Yajima, T., Takahashi, K. and Isaev, A. P.: Soil respiration in siberian taiga ecosystems with different histories of forest fire, Soil Sci. Plant Nutr., doi:10.1080/00380768.2000.10408759, 2000.

Schulze, E. D., Lloyd, J., Kelliher, F. M., Wirth, C., Rebmann, C., Luhker, B., Mund, M., Knohl, A., Milyukova, I. M., Schulze, W., Ziegler, W., Varlagin, A. B., Sogachev, A. F., Valentini, R., Dore, S., Grigoriev, S., Kolle, O., Panfyorov, M. I., Tchebakova, N. and Vygodskaya, N. N.: Productivity of forests in the eurosiberian boreal region and their potential to act as a carbon sink - a synthesis, Glob. Chang. Biol., doi:10.1046/j.1365-2486.1999.00266.x, 1999.

Serikova, S., Pokrovsky, O. S., Ala-Aho, P., Kazantsev, V., Kirpotin, S. N., Kopysov, S. G., Krickov, I. V., Laudon, H., Manasypov, R. M., Shirokova, L. S., Soulsby, C., Tetzlaff, D. and Karlsson, J.: High riverine CO2 emissions at the permafrost boundary of Western Siberia, Nat. Geosci., doi:10.1038/s41561-018-0218-1, 2018.

Shvartsev, S. L.: Geochemistry of fresh groundwater in the main landscape zones of the Earth, Geochemistry Int., doi:10.1134/S0016702908130016, 2008.

Shvidenko, A. and Nilsson, S.: A synthesis of the impact of Russian forests on the global carbon budget for 1961-1998, Tellus, Ser. B Chem. Phys. Meteorol., doi:10.1034/j.1600-0889.2003.00046.x, 2003.

Sommerkorn, M.: Micro-topographic patterns unravel controls of soil water and temperature on soil respiration in three Siberian tundra systems, Soil Biol. Biochem., doi:10.1016/j.soilbio.2008.03.002, 2008.

Tarnocai, C., Canadell, J. G., Schuur, E. A. G., Kuhry, P., Mazhitova, G. and Zimov, S.: Soil organic carbon pools in the northern circumpolar permafrost region, Global Biogeochem. Cycles, 23(2), doi:Gb2023\n10.1029/2008gb003327, 2009.

Tootchi, A., Jost, A. and Ducharne, A.: Multi-source global wetland maps combining surface water imagery and groundwater constraints, Earth Syst. Sci. Data, 11, 189–220, doi:10.5194/essd-11-189-2019, 2019.

Vonk, J. E., Mann, P. J., Davydov, S., Davydova, A., Spencer, R. G. M., Schade, J., Sobczak, W. V., Zimov, N., Zimov, S., Bulygina, E., Eglinton, T. I. and Holmes, R. M.: High biolability of ancient permafrost carbon upon thaw, Geophys. Res. Lett., 40(11), 2689–2693, doi:10.1002/grl.50348, 2013.

Vonk, J. E., Tank, S. E., Mann, P. J., Spencer, R. G. M., Treat, C. C., Striegl, R. G., Abbott, B. W. and Wickland, K. P.: Biodegradability of dissolved organic carbon in permafrost soils and aquatic systems: A meta-analysis, Biogeosciences, 12(23), 6915–6930, doi:10.5194/bg-12-6915-2015, 2015a.

Vonk, J. E., Tank, S. E., Bowden, W. B., Laurion, I., Vincent, W. F., Alekseychik, P., Amyot, M., Billet, M. F., Canário, J., Cory, R. M., Deshpande, B. N., Helbig, M., Jammet, M., Karlsson, J., Larouche, J., MacMillan, G., Rautio, M., Walter Anthony, K. M. and Wickland, K. P.: Reviews and Syntheses: Effects of permafrost thaw on arctic aquatic ecosystems, Biogeosciences Discuss., 12(23), 7129–7167, doi:10.5194/bgd-12-10719-2015, 2015b.

Vorosmarty, C. J., Fekete, B. M., Meybeck, M. and Lammers, R. B.: Global system of rivers: Its role in organizing continental land mass and defining land-To-Ocean linkages, Global Biogeochem. Cycles, 14(2), 599–621, doi:10.1029/1999GB900092, 2000.

[revised manuscript text omitted]

---

## Author Response (AR3)

Dear Topical Editor,

Thank you for your response.

To the following suggested changes made to the manuscript:

 (1) Figures 11 and 12 are missing.
(2) "Figs. 2 and 11" at line 129 would to be "Figs. 2 and 10".

We have altered them accordingly.

Kind regards,

Simon Bowring